# Modelling post-implantation human development to yolk sac blood emergence

Joshua Hislop[1,2,3], Qi Song[4,5,11], Kamyar Keshavarz F.[1,2,3,11], Amir Alavi[4,5,11], Rayna Schoenberger[1,2,3], Ryan LeGraw[2,3], Jeremy J. Velazquez[2,3], Tahere Mokhtari[1,2,3], Mohammad Naser Taheri[1,2,3], Matthew Rytel[2,3], Susana M. Chuva de Sousa Lopes[6], Simon Watkins[7,8], Donna Stolz[7,8], Samira Kiani[1,2,3,9], Berna Sozen[10], Ziv Bar-Joseph[4,5] & Mo R. Ebrahimkhani[1,2,3,9] ✉

Implantation of the human embryo begins a critical developmental stage that comprises profound events including axis formation, gastrulation and the emergence of haematopoietic system[1,2]. Our mechanistic knowledge of this window of human life remains limited due to restricted access to in vivo samples for both technical and ethical reasons[3–5]. Stem cell models of human embryo have emerged to help unlock the mysteries of this stage[6–16]. Here we present a genetically inducible stem cell-derived embryoid model of early post-implantation human embryogenesis that captures the reciprocal codevelopment of embryonic tissue and the extra-embryonic endoderm and mesoderm niche with early haematopoiesis. This model is produced from induced pluripotent stem cells and shows unanticipated self-organizing cellular programmes similar to those that occur in embryogenesis, including the formation of amniotic cavity and bilaminar disc morphologies as well as the generation of an anterior hypoblast pole and posterior domain. The extra-embryonic layer in these embryoids lacks trophoblast and shows advanced multilineage yolk sac tissue-like morphogenesis that harbours a process similar to distinct waves of haematopoiesis, including the emergence of erythroid-, megakaryocyte-, myeloid- and lymphoid-like cells. This model presents an easy-to-use, high-throughput, reproducible and scalable platform to probe multifaceted aspects of human development and blood formation at the early post-implantation stage. It will provide a tractable human-based model for drug testing and disease modelling.

Understanding human embryogenesis is central to treating congenital diseases and infertility as well as creating functional human cells and organs for transplantation. Immediately after implantation, the embryo and codeveloping extra-embryonic tissues are profoundly remodelled and initiate morphological changes central to the success of pregnancy, including the formation of the amniotic cavity and emergence of yolk sac haematopoiesis[1,2]. However, early post-implantation stages of human development are difficult to study due to limited access, technical complexities and ethical concerns[3–5]. Furthermore, with its flat bilaminar disc structure, the early post-implantation stage human embryo morphologically differs from its mouse counterpart. In vitro human stem cell-based embryo models or human embryoids have recently emerged, opening tremendous biomedical opportunities to study this stage of human life in detail[6–16]. However, current models each have their own drawbacks, such as instability in post-implantation stage progression, lack of crucial extra-embryonic layers, low efficiency

and technical challenges, including reliance on complex culture conditions that may not effectively support a wide range of cell types with optimal efficiency. Furthermore, none of the models have been able to reconstitute the intricate process of early human haematopoiesis, a critical event affecting both embryo viability and adult human health[17].

To overcome some of these challenges, we present a human embryoid model containing extra-embryonic niche and yolk sac haematopoiesis dubbed 'heX-embryoid'. This platform leverages an approach that genetically engineers a population of human-induced pluripotent stem (hiPS) cells. On induction, an extra-embryonic hypoblast-like niche is formed that triggers three-dimensional (3D) self-organization of the epiblast-like cells. heX-embryoid is similar to a flattened yolk sac cavity surrounding an epiblast–hypoblast interface and amniotic cavity (Extended Data Fig. 1a). It shows the development of a human bilaminar disc structure, specification of anterior hypoblast-like cells, lumenogenesis, formation of amnion-like tissue and symmetry breaking

[1]Department of Bioengineering, Swanson School of Engineering, University of Pittsburgh, Pittsburgh, PA, USA. [2]Department of Pathology, Division of Experimental Pathology, School of Medicine, University of Pittsburgh, Pittsburgh, PA, USA. [3]Pittsburgh Liver Research Center, University of Pittsburgh, Pittsburgh, PA, USA. [4]Computational Biology Department, School of Computer Science, Carnegie Mellon University, Pittsburgh, PA, USA. [5]Machine Learning Department, School of Computer Science, Carnegie Mellon University, Pittsburgh, PA, USA. [6]Department of Anatomy and Embryology, Leiden University Medical Center, Einthovenweg, Leiden, The Netherlands. [7]Center for Biologic Imaging, University of Pittsburgh, Pittsburgh, PA, USA. [8]Department of Cell Biology and Molecular Physiology, University of Pittsburgh, Pittsburgh, PA, USA. [9]McGowan Institute for Regenerative Medicine, University of Pittsburgh, Pittsburgh, PA, USA. [10]Department of Genetics, Yale School of Medicine, Yale University, New Haven, CT, USA. [11]These authors contributed equally: Qi Song, Kamyar Keshavarz F., Amir Alavi. ✉e-mail: mo.ebr@pitt.edu

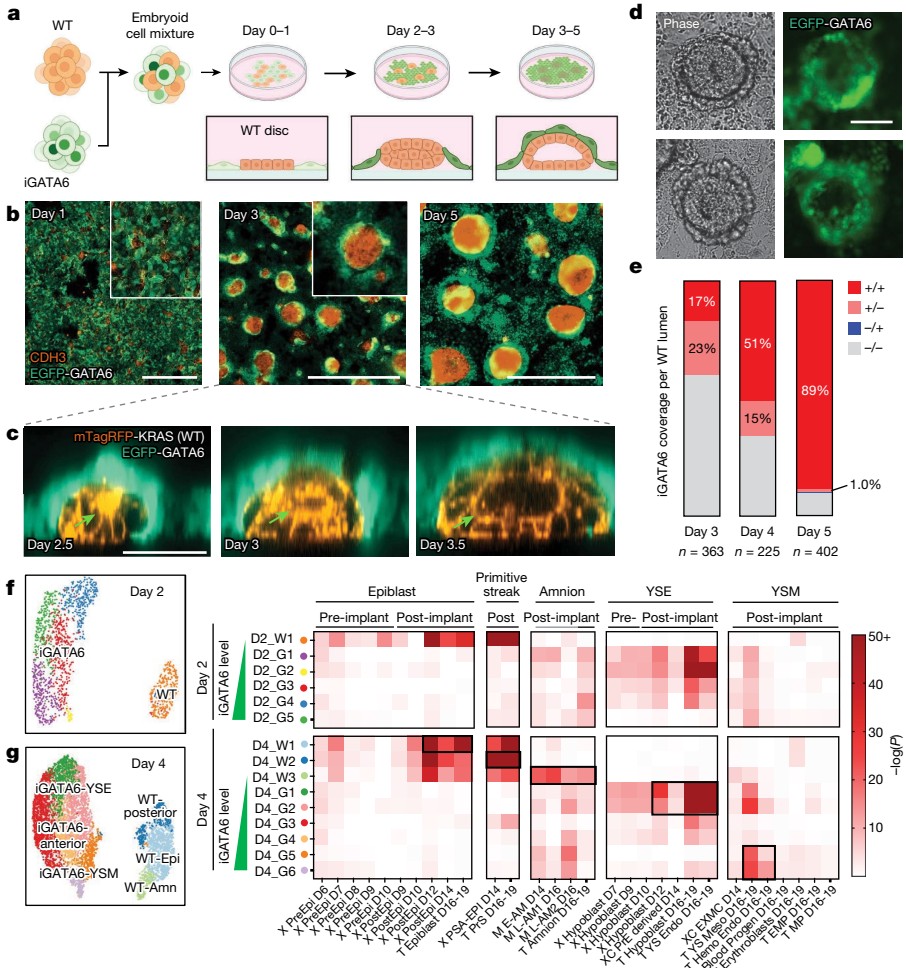

**Fig. 1 | Engineering codevelopment of embryonic and extra-embryonic endoderm tissues. a**, Schematic demonstrating cell mixing and subsequent organization of iGATA6 (green) and WT (orange) hiPS cells after GATA6 induction by Dox. **b**, IF staining demonstrating self-organization of heX-embryoids in the cultures. **c**, Live time-lapse images of one embryoid showing growth of a central lumen from a rosette indicated by an arrow. **d**, Phase and fluorescence images from live day 4 cultures showing developed heX-embryoid morphologies with EGFP-expressing iGATA6 around a WT cluster. **e**, Quantification of characteristics of WT clusters possessing different iGATA6 coverage and lumen formation characteristics. No islands without coverage were observed possessing a lumen (±0%). *n* represents the number of heX-embryoid assessed from at least two biological replicates. **f**, Single-cell UMAP and hypergeometric statistical comparisons of differentially expressed

gene (DEG) lists between day 2 embryoids and human and cynomolgus monkey embryo single-cell datasets. Labels above heatmaps indicate pre-implantation versus post-implantation embryo sample comparisons; grey bars under labels indicate cynomolgus monkey comparisons. Leading letters indicate the source dataset: T from ref. 23, X from ref. 24, M from ref. 26 and XC from refs. 24,25. **g**, Single-cell UMAP and hypergeometric statistical comparisons of differentially expressed gene lists between day 4 embryoids and human and cynomolgus monkey embryo single-cell datasets. Black boxes highlight high DEG similarity (low *P* values) to relevant populations of interest. YSE, yolk sac endoderm; YSM, yolk sac mesoderm. For hypergeometric tests, we performed Benjamini–Hochberg correction for multiple tests in all comparisons. Tests were performed as one-sided tests. Scale bar, 50 μm (**c**), 100 μm (**d**), 1,000 μm (**b**). Illustration in **a** was created using BioRender (https://biorender.com).

leading to posterior pole specification. Its extra-embryonic-like tissue shows maturation towards yolk sac tissue identity and the emergence of possible haematopoietic niches together with different waves of blood-like cells. Our platform lacks any trophoblast layer or analogue; hence, it offers a 'non-integrated' model of human embryogenesis with extra-embryonic haematopoiesis, at an early implantation stage when the embryo in vivo is at its most inaccessible state.

## Epiblast and hypoblast codevelopment

We engineered an hiPS cell line with an inducible transgene that expresses human GATA6 (iGATA6), a key transcription factor for extra-embryonic endodermal fate decision (Fig. 1a and Extended Data Fig. 1b). These iGATA6 cells express different amounts of GATA6 on the addition of a small molecule, doxycycline (Dox), distinguishable by expression of the reporter enhanced green fluorescent protein (EGFP)

(Extended Data Fig. 1c). When iGATA6 and wild-type (WT) hiPS cells were co-assembled in 3D, they yielded aggregates in which the entire EGFP+ extra-embryonic endoderm-like cells (outer layer) were in close contact with WT epiblast-like cells (inner layer), missing parts analogous to yolk sac tissue that was formed away from the epiblast cells (parietal). Furthermore, we found that WT cells could not consistently induce symmetry breaking to amnion- and epiblast-like domains and the overall organization did not recapitulate the structural format of the bilaminar disc (Extended Data Fig. 1d,e). Moreover, in vitro reattachment of 3D embryo-like structures to a dish to mimic implantation has shown organizational instability that limits developmental potential[6].

To address these challenges, we aimed to develop a strategy to generate structures off of the cell culture plate by igniting two-dimensional (2D)-to-3D self-organization. The iGATA6-hiPS cells were mixed with WT hiPS cells, and the mixed population of cells was seeded onto standard culture plates at a defined cell ratio and density (Fig. 1a). After treatment

with Dox, the iGATA6 cells upregulate GATA6 and EGFP as expected, and show loss of the pluripotency marker NANOG (Extended Data Fig. 1f). During the first 48 h, from an initial state of random distribution, induced iGATA6 cells and WT cells proliferate, and WT cells are organized into disc-shaped clusters confined by iGATA6 cells (Supplementary Video 1, Fig. 1b and Extended Data Fig. 1a,g,h). Subsequently, the migration of the iGATA6 cells over the upper surface of these WT clusters occurs in conjunction with 3D growth within the WT disc (Extended Data Fig. 1i and Supplementary Video 2). This behaviour was observed when the system was maintained on mTeSR (Extended Data Fig. 1j). These events result in a membrane of extra-embryonic endoderm-like cells that express GATA4 and SOX17 but lack OCT4 over a multilayer WT structure (Fig. 1a,c,d and Extended Data Fig. 1i). The process of upwards iGATA6 migration occurs in parallel with the deposition of a laminin membrane surrounding the WT clusters (Extended Data Fig. 2a,b). The laminin deposition by the iGATA6 cells can subsequently trigger cell polarization within cells of the WT cluster, which promotes the formation of cellular rosettes central to the WT clusters, resembling embryonic morphology around Carnegie Stage (CS)5a (Fig. 1c and Extended Data Fig. 2a). We observe the conversion of these rosettes into lumens, mimicking the expansion of the pro-amniotic cavity around CS5b–CS5c of development (Fig. 1c, Extended Data Fig. 2a and Supplementary Videos 3 and 4)[18]. This is facilitated by expression of PODXL, an apically expressed antiadhesive surface protein implicated in embryonic lumen formation, and ZO-1, an apical tight junction protein (Extended Data Fig. 2d,e)[19]. This laminin deposition and lumen formation is not observed in WT-only culture in equivalent conditions (Extended Data Fig. 2f). We noticed that when an iGATA6 layer did not surround the WT clusters, lumen formation did not occur (Fig. 1e). The number of clusters with no coverage by an iGATA6 layer progressively decreases by day 5 as the migration of iGATA6 cells moves forward (Fig. 1e). Ultimately, following expansion of the lumen, the two cell monolayers, composed of epiblast-like WT (NANOG⁺) and endoderm-like GATA6⁺ cells, are positioned on either side of a laminin membrane (Fig. 1a,c and Extended Data Fig. 2d), forming a cellular arrangement similar to the bilaminar disc in the early post-implantation human embryo[20]. Analysis of the culture media showed high amounts of secreted AFP and APOA1, two proteins known to be produced by primitive endoderm and yolk sac tissues (Extended Data Fig. 2g).

During our initial analysis, we noticed that at day 5, in WT clusters with higher area and often lower circularity, cavity formation occurred by several lumens (Extended Data Fig. 2h,i). To optimize cluster size and number, we tested a range of seeding ratio and densities for iGATA6 and WT cells, to achieve cluster area most consistently within a size range corresponding to the E9–E17 human bilaminar disc (Extended Data Fig. 2j) (Carnegie nos. 8004 and 7700 in ref. 21, Carnegie no. 7801 in ref. 22). To this end, we selected a seeding ratio of 81/5 iGATA6/WT at a density of 54,000 cells per cm² (Extended Data Fig. 2j, dotted boxes). When we applied this optimized seeding density, we could detect 74% of embryoids in the expected physiological size range (Extended Data Fig. 2k). We also observed most (70.9%) of our embryoids in this size range produced a single lumen (Extended Data Fig. 2k). Thus, by adjusting these initial parameters, we could gain control over size, number, circularity and lumen formation.

## Intra- and/or extra-embryonic scRNA trajectories

To uncover the cell types that develop during the observed morphogenetic events, we conducted single-cell RNA sequencing (scRNA-seq) from day 0 to day 5 of cultures (Extended Data Figs. 3–5). Performing an unsupervised comparison of the transcriptomes with the E16–19 human embryo[23], E6–E14 human embryo samples[24,25] and the E18–20 cynomolgus embryo[26] revealed the acquisition of distinct post-implantation-like cellular fates in our identified subpopulations (Fig. 1f,g and Extended Data Fig. 4a–c). As early as 24 h (D1) after induction, cultures showed the emergence of cell populations (D1_G1, G2, G3) with transcriptomic similarity to human hypoblast, alongside a corresponding epiblast-like population (D1_W1) (Extended Data Fig. 4a–c). By 36 h (D1.5), iGATA6 cells become statistically similar to the human E12 hypoblast lineage and this similarity strengthens through to day 5 of culture (Extended Data Fig. 4b)[24]. Comparison with the E16–19 human and cynomolgus embryo shows significant similarity to yolk sac endoderm and hypoblast, appearing at D1 and strengthening to day 5 ($P = 6.00 \times 10^{-54}$ to $2.84 \times 10^{-93}$ similarity significance to yolk sac (YS) Endoderm at day 5) (Fig. 1f and Extended Data Fig. 4a,c)[23,26].

On day 2, there is significant statistical similarity to yolk sac endoderm within the iGATA6 population (clusters D2_G1, D2_G2, D2_G3, D2_G4; $P = 1.3 \times 10^{-17}$ to $9.5 \times 10^{-69}$), with the highest *GATA6*-expressing cells (cluster D2_G5) diverging from this fate towards a more yolk sac mesoderm-like state ($P = 6.8 \times 10^{-11}$) (Fig. 1f and Extended Data Fig. 4a–c). By day 4, we observed the strengthening of the identities and of the observed fate divergence. The lower *GATA6* clusters (D4_G1 and D4_G2) showed increased expression of key yolk sac marker genes (*GATA4*, *PDGFRA*, *LAMA1*, *CUBN*, *AMN*, *NODAL*) and statistical similarity to human yolk sac endoderm (Fig. 1g and Extended Data Figs. 3b and 4a–c, $P_{G1} = 1.3 \times 10^{-92}$, $P_{G2} = 1.8 \times 10^{-63}$)[23,27]. Each of the day 4 putative iGATA6 endoderm was also statistically compared to the differentially expressed gene lists of yolk sac endoderm and proliferating definitive endoderm from the E16–19 human embryo. The lower *GATA6* clusters D4_G1 and D4_G2 show a higher number of genes in common with yolk sac, including apolipoprotein genes, *PDGFRA* and *CUBN* (Extended Data Fig. 3c)[23], which is corroborated with the differential Jaccard similarity to yolk sac endoderm between these populations (Extended Data Fig. 3d).

We next assessed other developed populations in the iGATA6 layer. We observed that a 'medium' *GATA6* population (D4_G3 and D4_G4) had a strong anterior hypoblast-like identity (homologous to mouse anterior visceral endoderm, with specific upregulation of anterior hypoblast markers (*LHX1*, *HHEX*, *FZD5*, *CER1*, *LEFTY1*) (Extended Data Fig. 3b)[23,28–30]. It showed lower statistical similarity to hypoblast and/or yolk sac endoderm compared with low expressing *GATA6* populations (Fig. 1g and Extended Data Fig. 4). This population shows a higher number of differentially upregulated genes in common with the definitive endoderm, which corroborates a recent study aligning human anterior endoderm more closely to a definitive endoderm trajectory (Extended Data Fig. 3c,d)[31]. We observed that the highest *GATA6* clusters, D4_G5 and D4_G6, demonstrated statistical similarity to human yolk sac mesoderm with very subtle to no equivalent similarity to emergent, nascent and axial mesoderm (Fig. 1g and Extended Data Fig. 4a,b, $P_{G5} = 1.7 \times 10^{-22}$, $P_{G6} = 2.5 \times 10^{-25}$). Our analysis revealed that the signature for extra-embryonic mesoderm emerged as early as 12 h after the system's induction in the highest *GATA6* population (Extended Data Fig. 4a). Taken together, our data demonstrate a divergence in tissue fates similar to those in the yolk sac on the basis of *GATA6* expression amounts, with yolk sac endoderm-, anterior hypoblast- and yolk sac mesoderm-like cells stratifying between populations expressing low, medium and high average amounts of *GATA6* (Extended Data Fig. 4 top row).

Recently, observations in the cynomolgus monkey by Zhai et al. indicate a similar stratification of GATA6 concentrations in a variety subset of tissues, notably between yolk sac endoderm (lower GATA6) and mesoderm (higher GATA6) populations[32]. Within the WT clusters, identified by a lack of *GATA6* or *EGFP* transcripts, we observe the divergence of three separate populations from the epiblast-like compartment. The first and largest cluster (D4_W1) maintained the similarity to human epiblast, with expression of the pluripotency markers *POU5F1* (OCT4), *SOX2* and *NANOG* (Extended Data Figs. 3b and 4a, $P_{W1} = 1.15 \times 10^{-49}$). The second (D4_W2) WT cluster had lower statistical similarity to human epiblast ($P_{W2} = 3.4 \times 10^{-20}$ to D16–19 Epi), but stronger similarity to human primitive streak ($P_{W2} = 3.6 \times 10^{-95}$ to D16 PrS) (Fig. 1g and Extended Data

Figs. 3b and 4a). Our assessment of the third WT cluster at this time point (D4_W3) showed the occurrence of a unique transcriptomic similarity to human and cynomolgus amnion ($P_{W3}$ = 8.0×10$^{-15}$ and 1.6 × 10$^{-25}$, respectively) (Fig. 1g and Extended Data Figs. 3b and 4a,c)[33]. We did not observe a similarity to human trophoblast lineages (Extended Data Fig. 4b). When we integrated all clusters across different days, iGATA6 and WT lineages occupied distinct compartments within the projection space. However, we did observe spatial positioning of populations aligned with their time of sampling (Extended Data Fig. 5a). Populations sampled at different times can show similar fates (that is, day 3 versus day 5 iGATA6 yolk sac endoderm-like cells) (Extended Data Fig. 5b). The top-upregulated genes show distinct patterns across different cell fates, with overlapping genes present across similar fates sampled at different time points (Extended Data Fig. 5c).

## Specification of amniotic ectoderm

Primate amniogenesis differs structurally and temporally from mice, emphasizing the need for human model systems[33]. During the early post-implantation phase, following lumenogenesis in the epiblast, the epiblast cells undergo dorsal–ventral patterning to separate the epiblast from amniotic ectoderm[34]. Our single-cell transcriptomic analysis in WT cells showed a group of cells with transcriptomic similarity to human and cynomolgus amnion that emerged around day 3 (Fig. 2a,b and Extended Data Figs. 4a and 5b). These cells express amnion markers, including *ISL1*, *TFAP2A* (AP-2α) and *GATA3*, maintaining expression of OCT4 while substantially lacking expression of *NANOG* (Fig. 2a,b)[16,33]. Our analysis through immunofluorescent (IF) staining corroborated this finding, demonstrating a layer of ISL1$^+$AP-2α$^+$ in heX-embryoids (Fig. 2c–e and Extended Data Fig. 6a). We observed that these cells were spatially segregated within the cavitated sac-like structure, with the amnion-like lineage positioning away from the bilaminar iGATA6/WT disc, forming a membrane against the tissue culture dish. WT cells above, within the bilaminar iGATA6/WT disc, showed expression of NANOG (Fig. 2c–e and Supplementary Video 5). We found that all embryoids containing cavitated structures (100%, $n$ = 350) expressed ISL1 at a level high above the background signal (Fig. 2f).

Further IF staining for phosphorylated (p)SMAD1, pSMAD5 and pSMAD8/9 (BMP4 effectors) revealed BMP4 signal transduction in a ring pattern concentrated at the edges of the embryoid (Fig. 2g and Extended Data Fig. 6b). scRNA-seq at day 4 also showed specific upregulation of BMP4 and its targets within the amnion-like population (Extended Data Fig. 6c). Recently, the active role of primate amniotic tissue to promote BMP4 signalling has been suggested, acting downstream of ISL1 (ref. 33). It was also reported that BMP4 can promote the formation of ISL1$^+$ amniotic tissue in a human embryo model[16,33]. Hence, to probe the link between BMP4 signalling and amnion fate in heX-embryoid, we aimed to inhibit BMP signalling by treatment with Noggin. We observed effective suppression of SMAD1/5/8 phosphorylation (Fig. 2h and Extended Data Fig. 6d), and a marked decrease in amnion fate acquisition assessed by ISL1 expression (Fig. 2i and Extended Data Fig. 6e,f). Collectively, we conclude that our platform shows robust amniogenesis with formation of a dorsal–ventral axis that is dependent on BMP4 signalling, supporting the use of these embryoids for studying this key developmental event in humans.

## Anterior hypoblast and posterior domain

In vivo evidence shows a group of cells with anterior visceral endoderm characteristics in human and non-human primates that have been termed anterior hypoblast[29]. In mice, this tissue influences the posterior axis specification through the expression of signalling inhibitors, including CER1. Our scRNA-seq analysis of iGATA6 cells as early as day 2 revealed a cluster of cells that show markers associated with anterior hypoblast. These cells are positive for *CER1*, *LHX1*, *HHEX*, *LEFTY1* and

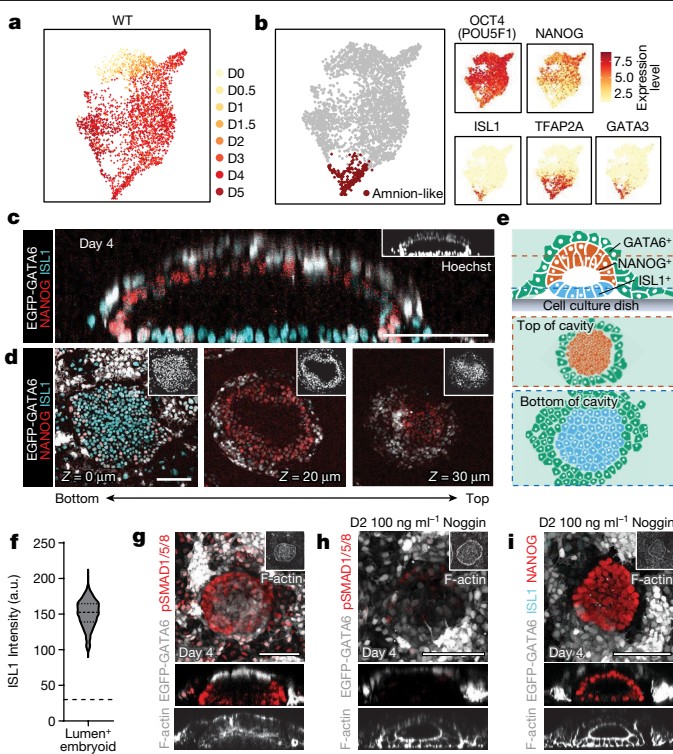

**Fig. 2 | Amniotic cavity formation and expansion. a**, Merged UMAP of all WT lineages from the embryoids labelled by day of development. **b**, The merged WT population showing the compartment expressing markers of amnion. *ISL1*, *TFAP2A* (AP-2α) and *GATA3* are expressed in this area, whereas *NANOG* is negative and OCT4 is low. **c**, An orthogonal slice of an individual embryoid showing top to bottom compartmentalization of ISL1$^+$ and NANOG$^+$ cells. iGATA6 (white) cells cover the top. **d**, Horizontal z-slices at the indicated distance from the dish of the WT cluster from **c**. **e**, Schematic showing the position of each population within a single heX-embryoid. Dotted lines indicate the area of slices shown. **f**, Violin plot showing the expression of ISL1 in day 4 embryoids containing lumens ($n$ = 350). The dotted line indicates a negative threshold for ISL1. **g**, Expression patterns of BMP4 effectors (phosphorylated SMAD1, SMAD5 and SMAD8/9). Lower images show a lateral slice of the WT disc. **h**, IF staining for the BMP4 effectors (phosphorylated SMAD1, SMAD5 and SMAD8/9) at day 4 of cultures after application of the inhibitor Noggin at day 2 of development. Lower images show a lateral slice of the WT disc. **i**, IF staining for ISL1 and NANOG at day 4 after application of Noggin at day 2 of development. Lower images show a lateral slice of the WT disc. Scale bars, 100 μm. $n$ represents embryoid structures harvested from three independent experiments. a.u., arbitrary units.

*LEFTY2* (Fig. 3a)[29]. A population expressing these markers continued to be present in the following days (days 3 to 5) (Fig. 3a) but showed a distinct cluster from day 2, primarily related to cell cycle-related pathways (Extended Data Fig. 5d). Moreover, the analysis detected the emergence of a distinct cluster of cells at day 3 and 4 in WT cells presenting *TBXT* (Brachyury), *MIXL1*, *GDF3* and *EOMES*, key markers associated with development of the posterior pole in the embryo (Fig. 3b)[35].

Following up with IF staining, we identified domains of cells expressing anterior hypoblast-related proteins (LHX1, HHEX) in a subset of extra-embryonic cells surrounding the WT clusters as early as day 2 (Extended Data Fig. 6g). On day 4, CER1 is observed to be expressed in a polar manner within the LHX1$^+$/HHEX$^+$ domains (Fig. 3c). Out of 396 embryoids assessed, we identified 42.4 ± 4.1% (mean ± s.d. 169) embryoids with polar CER1-expressing domains on day 4, showing a similar efficiency to existing ex vivo human blastocyst cultures[11,29]. In parallel, we showed development of TBXT$^+$ domains in 46.4 ± 9.0% (323 out of 746) of WT clusters on day 4. 30.8 ± 5.0% (mean ± s.d.; 215) of these 746 clusters showed asymmetric (polar) TBXT expression (Fig. 3d,e and

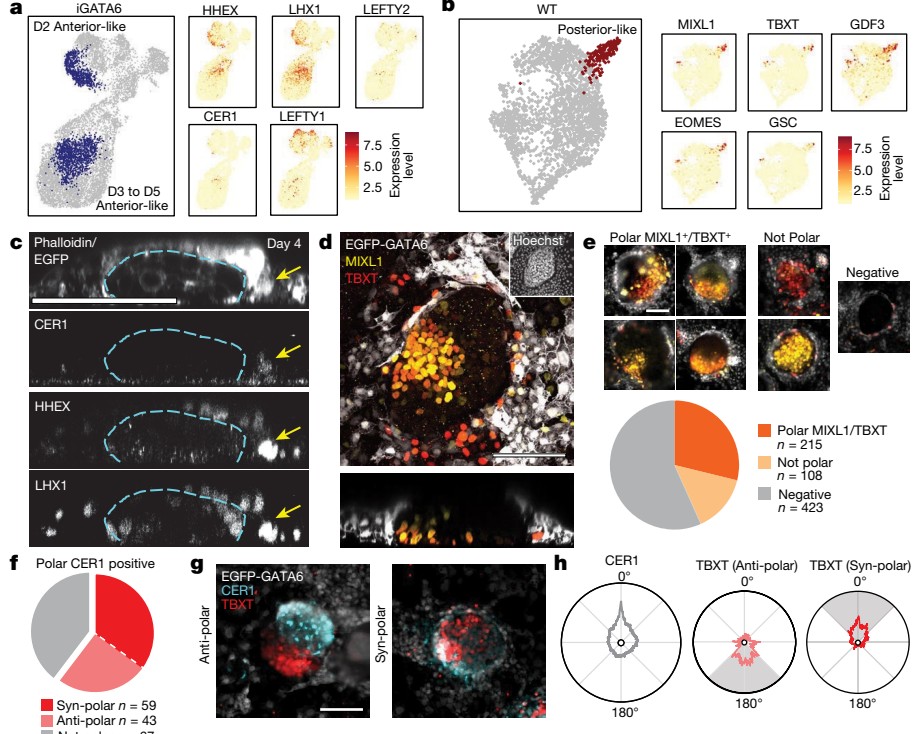

**Fig. 3 | Anterior hypoblast domain and posterior pole in heX-embryoids.**
**a**, Merged UMAP of all iGATA6 lineages showing the compartments expressing markers of anterior hypoblast. Two separate domains of these markers were observed, one within day 2 and one within day 3–5 of the embryoid cells. (UMAP labelled by day can be seen in Extended Data Fig. 5). **b**, Merged UMAP of all WT lineages showing the compartment expressing markers of the posterior pole and primitive streak. **c**, IF staining showing a *z*-slice of a heX-embryoid with a HHEX/LHX1/CER1 copositive domain adjacent to a WT cluster. The dotted line indicates the boundaries of the WT cluster in each image, and the arrow indicates the copositive domain. **d**, IF staining showing a TBXT/MIXL1 copositive domain within the WT clusters of the embryoid. **e**, Examples of WT clusters with polarized TBXT/MIXL1 domains (top) or without co-expression of markers (bottom). The pie chart shows proportions of cluster types observed

in the embryoid cultures, *n* = 746 total. **f**, A pie chart showing the proportion of clusters with a particular TBXT/CER1 polarity type when CER1 confined to one pole is present. *n* = 169 total. **g**, Representative WT clusters showing TBXT and CER1 expression domains at opposing poles of the cluster (anti-polar, top) or at the same pole of the cluster (syn-polar, bottom). **h**, Diagrams indicating the average radial expression patterns of TBXT in each polarity pattern from WT clusters with a polarized CER1-expressing domain indicated in **f**. All diagrams are scaled to the same expression intensity value (1 a.u.). Degrees indicate radial distance around the circularized perimeter of a WT cluster from the CER1 peak shown. Shaded areas indicate region of highest average TBXT polarity corresponding to the polarity types. Scale bars, 100 µm. *n* represents the embryoid structures from three to four separate experiments harvested on day 4 after induction.

Supplementary Video 6). Treatment with Noggin could eliminate TBXT⁺ poles, whereas we still observed the specification of CER1-expressing cells (Extended Data Fig. 6h,i). Although this observation warrants further mechanistic investigation in the future, it highlights the role of BMP4 from amnion-like structures in posterior domain specification in this model, corroborating with past reports[36].

We then analysed our embryoids to evaluate anterior–posterior axis positionings. On day 4, we observed that 61.2 ± 8.6% (mean ± s.d.; 102 of 169) of embryoids with polar configuration in CER1-expressing cells possessed a TBXT-expressing pole (Fig. 3f). Within embryoids with polarity in both CER1 and TBXT we observed two distinct patterns (Fig. 3f,g and Extended Data Fig. 6j,k). 41.2 ± 8.5% show an anti-polar configuration, in which the TBXT⁺ pole occupies the opposite radial region as the CER1⁺ domain, similar to anterior–posterior axis positioning in mouse at E6.5 (Fig. 3f–h and Extended Data Fig. 6k). In a second set of polar embryoids, we saw high expression of CER1 in the proximity of TBXT⁺ domains (Fig. 3f–h and Extended Data Fig. 6j). This set of embryoids shows CER1⁺ and TBXT⁺ poles with a same radial region (syn-polar), in contrast to past reported literature.

We also noted expression of CER1 in a subset of TBXT⁺ cells in the posterior domains in some cases (Extended Data Fig. 6j,k) and subsequent scRNA-seq analysis showed the presence of a subpopulation of *CER1⁺TBXT⁺* cells in the cluster that represents posterior identity (Extended Data Fig. 6l). Co-expression of CER1 and TBXT may be

reflective of a more advanced stage post-E14 in which CER1 expression acts either to prevent spreading of posterior domain[37] or represent early mesendoderm commitment derived from a primitive streak[32]. Further research is needed to explore the in vivo relevance of co-expressing cells and the observed non-canonical polarization types (syn-polar) in the human context. In summary, we propose heX-embryoids offer a model to study anterior–posterior axis biology, the development of anterior hypoblast and posterior poles, and their respective cellular progeny in a human context.

## Yolk sac mesoderm and blood progenitors

During human development, yolk sac serves dual roles as a nutritional source and a site for early blood production, beginning around E16–18 (refs. 1,38,39). We detected developmental and functional maturation of cells in the iGATA6 layer aligned with human yolk sac endoderm and mesoderm fates (Fig. 1g and Extended Data Figs. 2f and 4a). Hypergeometric statistical analysis of the clusters with highest expression of GATA6 on day 4 (D4_G6) and 5 (D5_G3) reveals significant similarity to human yolk sac mesoderm, haematopoietic lineage progenitors and haemogenic endothelial cells using the annotation of the E16–19 human embryo. However, the haemogenic endothelial nature of this cluster cannot be determined without further analysis. (Extended Data Fig. 7a,b)[23]. Examination of the D5_G3 cluster reveals higher expression

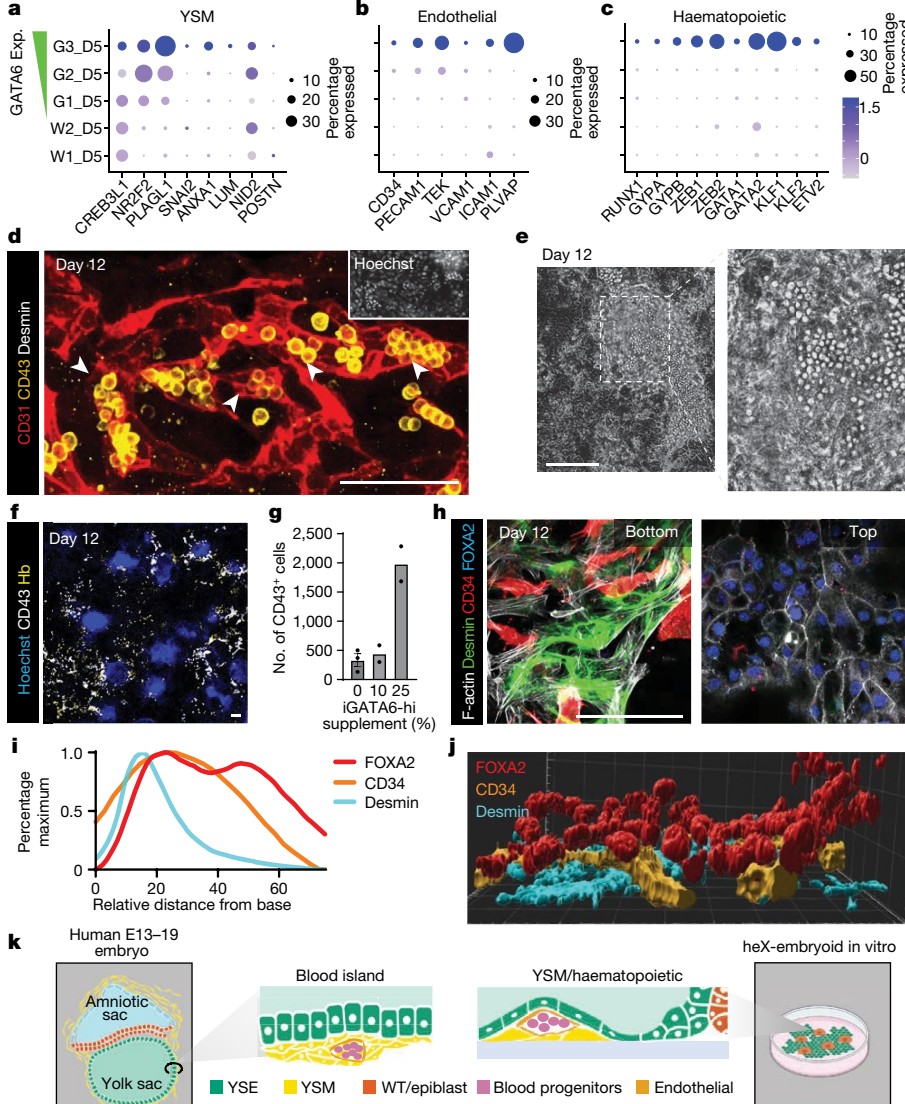

**Fig. 4 | Haematopoietic lineages and haematopoietic foci structures in the heX-embryoids. a**, Dot plot showing the expression pattern of yolk sac mesoderm markers in day 5 embryoid scRNA-seq populations. **b**, Dot plot showing the expression pattern of endothelial markers in day 5 embryoid scRNA-seq populations. **c**, Dot plot showing the expression pattern of haematopoietic markers in day 5 embryoid scRNA-seq populations. **d**, IF image of day 12 embryoid showing the generation of CD43⁺ spherical cells within CD31⁺ endothelial cells. **e**, Live phase image taken on day 12 showing cells spherical cells. The dotted box indicates area of the inset. **f**, IF image of a day 12 culture after GATA6-hi cells were added into the starting cell mix. **g**, Bar plot showing the number of CD43⁺ cells detected on day 12 after the given percentage of GATA6-hi cells were supplemented. $n$(0%) = 4 replicates, $n$(10%) = 2 replicates, $n$(25%) = 2 replicates. Error bars represent mean ± s.e.m.

**h**, Two slices from a single representative structure demonstrating Desmin⁺ mesoderm-like cells and CD34⁺ endothelial-like cells localized underneath a FOXA2⁺ endoderm-like layer, reminiscent of yolk sac blood island morphology. **i**, Histogram showing the $z$-distribution of the indicated markers between the bottom of the dish and the top of the culture. Bimodal distribution of FOXA2 indicates areas of expression outside the haematopoietic foci. Distributions are averaged from nine foci from three technical replicates. **j**, 3D reconstruction of one haematopoietic focus showing the positioning of endoderm (FOXA2), endothelial (CD34) and mesoderm (Desmin) marker expression. **k**, Schematic depicting in vivo embryonic yolk sac blood islands and in vitro haematopoietic foci structures. Scale bars, 100 μm. Illustration in **k** was created using BioRender (https://biorender.com).

of the extra-embryonic mesoderm cell surface marker BST2, as well as of ECM proteins (Extended Data Fig. 7a)[39,40]. Subpopulations of cells within this cluster also show a higher average expression markers of human yolk sac mesoderm (*CREB3L1*, *NR2F2*, *PLAGL1*, *ANXA1*, *NID2*)[39,40], markers of endothelial cells (*CD34*, *PECAM1* (CD31), *TEK*, *ICAM1*, *PLVAP*)[23,41] and markers of human haematopoiesis (*GATA1*, *GATA2*, *KLF1*, *KLF2*, *ZEB1*, *ZEB2*, *GYPA* (CD235a) and *GYPB* (CD235b)) (Fig. 4a–c)[23,39,41].

IF analysis on day 5 initially reveals spindle-shaped cells positive for TAL1 (scl), a key regulator of blood development. Further analysis shows a subset of these cells were copositive for CD34 and ERG, markers associated with haematopoietic and endothelial fates; we

also identified a small population copositive for TAL1 and RUNX1, itself another master transcription factor in haematopoiesis (Extended Data Fig. 7c–e)[41]. Image analysis shows ERG⁺ endothelial-like cells consistently emerged underneath an iGATA6 layer (Extended Data Fig. 7f), resembling the initial in vivo localization of human yolk sac mesoderm, where it is positioned against ECM protein in the area underneath the yolk sac endoderm[39]. The CD34⁺TAL1⁺ cells were also EGFP⁺, indicating their development from an iGATA6-derived parental population (Extended Data Fig. 7d). Staining for the endothelial markers KDR (VEGFR2) and CDH5 confirmed similar localization in day 5 embryoids (Extended Data Fig. 7g). To further characterize the development of

these endothelial-like cells with haematopoietic characteristics, we followed these cultures beyond day 5. We observed minimal expansion or differentiation of the CD34+ endothelium in mTeSR media (Extended Data Fig. 7h). Switching to Iscove's modified Dulbecco's medium (IMDM) basal media without Dox after day 5 resulted in a notable expansion of the CD34+ endothelial-like cells by day 12 (Extended Data Fig. 7h,i). We also observed the emergence of condensed areas containing spherical cells in the yolk sac tissue-like compartment by day 12 (Fig. 4d,e). IF staining of these areas revealed the presence of a CD31+ endothelial layer surrounding spherical cells expressing Leukosialin (CD43), a marker of haematopoietic progenitors (Fig. 4d and Supplementary Video 7)[42]. Having noted that yolk sac mesodermal-like and haemogenic progenitor-like cells were predominantly associated with the GATA6-hi cluster in scRNA-seq analysis, we further enriched heX-embryoid cultures at the time of seeding with cells with a high copy number of the inducible GATA6 circuit. This supplementation resulted in a notable increase in the generation of CD43+ haematopoietic-like foci (Fig. 4f,g and Extended Data Fig. 8a,b), demonstrating the ability to predictably and controllably enhance the final culture phenotype at the time of seeding.

## Yolk sac-like haematopoiesis

Haematopoietic processes emerge in vivo in the form of blood islands within yolk sac tissues. Structural image analysis of CD43+ foci in heX-embryoids revealed an intricate 3D hierarchical organization. CD34+ endothelial vessel-like structures were packed between FOXA2+ endoderm-like cells occupying the top compartment, while Desmin+ mesoderm-like cells formed the basal layer (Fig. 4h–j, Extended Data Fig. 8c,d and Supplementary Video 8). This cellular arrangement resembles the reported morphology of blood islands in the developing human yolk sac at roughly E19–23 (Fig. 4k)[39].

Further analysis showed the specification of cells expressing markers for erythroid (CD235a and haemoglobin), myeloid and/or macrophage (CD33 and CX₃CR1) and megakaryocyte lineages (CD42b and CD41) (Fig. 5a–c). These cells are specified in multilineage or uni-lineage foci within the yolk sac-like tissue. We show foci containing both erythroid-like and megakaryocyte-like cells or foci with myeloid-like and megakaryocyte-like cells. Most haematopoietic foci observed were multilineage, with a smaller fraction being uni-lineage (for example, CD42b+ uni-lineage foci; roughly 87.2–95.0% multi- versus 8.8–5.0% uni-lineage among all foci) (Fig. 5d,e and Extended Data Fig. 9a–d). Next, we examined the haematopoietic-like lineages that emerge beyond the second week of culture. A colony-forming unit (CFU) assay initiated from day 21 embryoids confirmed haematopoietic potential and showed generation of both erythroid-like and myeloid-like colonies (Fig. 5f). We then performed scRNA-seq on day 21 of the embryoids and compared these data against the E16–19 human embryo dataset[23]. We were able to identify cell populations with significant similarity to endothelial cells, erythroblasts, myeloid and/or erythro-myeloid progenitors and blood progenitors within this dataset (Fig. 5g and Extended Data Fig. 9e). To further interrogate the presence of a yolk sac-like haematopoiesis, we also investigated transcript expression for HOX genes. The HOXA family gene expression is not found in the haematopoietic wave preceding the emergence of aorta–gonad–mesonephros haematopoietic stem cells[43]. By contrast, *HOXB7* and *HOXB9* can be detected in haematopoietic cells during this phase. We show that endothelial-like and haematopoietic-like progenitors in heX-embryoid lack medial HOXA assessed by *HOXA5*, *HOXA7* and *HOXA9* while expressing posterior *HOXB5*, *HOXB7* and *HOXB9* genes (Extended Data Fig. 9f)[43,44]. These cells also show expression of embryonic genes *LIN28A*, *GAD1* and *FGF23* that are shown to be correlated with early waves of haematopoiesis and prehaematopoietic stem cell haematoendothelial programmes (before CS10-11) (Extended Data Fig. 9g)[44]. Furthermore, the expression of *HBE1* and *HBZ* in our

erythroid-like cells demonstrate an expression pattern unique to yolk sac erythropoiesis[45]. Collectively, we conclude that the de novo haematopoiesis-like process in heX-embryoids demonstrates yolk sac characteristics.

## Cell composition of haematopoietic waves

Yolk sac haematopoiesis comprises two waves: a primitive wave begins at CS7 (week 2.5 postfertilization), generating early erythroid, myeloid and megakaryocyte progenitors, and a subsequent definitive wave starts at roughly CS8–9 (week 3.25 postfertilization) comprising erythro-myeloid (EMP) and lymphoid-primed multipotent progenitors programmes[46,47]. We investigated these haematopoietic waves in our platform.

Primitive erythroid progenitors show high expression of embryonic globin (Hb ε) and low expression of fetal globin (Hb γ). As the embryonic erythroid programme shifts to produce EMP-derived erythroid cells, the change results in an inversion of this pattern. This inversion is reflected in a decline in the Hb ε to Hb γ ratio, resulting in a nearly even average ratio of expression by 4 weeks of gestation[45,48]. IF staining of heX-embryoid cultures revealed that Hb-expressing cells in week 2 embryoids (days 12 and 15) show high Hb ε and little to no expression of Hb γ. However, Hb-expressing cells at day 21 show an increase in Hb γ concentration (Fig. 5h). We assayed the culture at different time points using both image analysis of the individual cells and quantitative PCR with reverse transcription (RT–qPCR) of whole cultures. Both analyses showed a decline in the ratio of *HBE* to *HBG* from day 12 to 21 (Fig. 5i, Extended Data Fig. 10a–c and Supplementary Table 1). The temporal switch in the erythroid-like cells potentially implies a primitive-to-EMP-like transition in erythropoiesis. However, the maturation of primitive erythrocytes alone may also involve a globin switch[49], which warrants further investigation.

As EMP haematopoiesis results from the transition of haemogenic endothelial cells to haematopoietic progenitors, we next investigated the presence of a haemogenic endothelial signature within the endothelial population. Examination of the endothelial-like cell cluster in scRNA-seq identified a subcluster expressing genes matching those reported in haemogenic endothelial cells[50]. This population showed canonical endothelial markers (*CD34*, *CDH5*, *ERG*), lacked the expression of *NT5E* (CD73), a marker of non-haemogenic endothelium, and expressed *GATA2*, an indispensable factor for the endothelial-haematopoietic transition (Extended Data Fig. 10d)[51]. Whereas *RUNX1* transcripts are present in the haematopoietic clusters, we did not identify its expression in the endothelial cluster, probably because of its transient nature of upregulation in endothelial cells during the haematopoietic transition. We identified a small number of *CDH5^low+^RUNX1+ITGA2B+* cells in haematopoietic-like clusters that are probably the immediate descendants of the haematopoietic transition from endothelial-like cells (Extended Data Fig. 10e)[52]. We also identified cells in the haematopoietic-like cluster co-expressing *RUNX1* and *CD44*, a marker of definitive yolk sac haematopoietic progenitors (Extended Data Fig. 12f)[53,54]. Subsequent IF staining for RUNX1 on day 21 confirmed the specification of these VE-cad+RUNX1+ cells near VE-cad+ endothelium (Extended Data Fig. 10g). Hence, a subset of endothelial-like cells in our platform showed characteristics similar to the haemogenic endothelium.

Further analysis of our day 21 scRNA-seq revealed presence of erythroid- (*GYPA* (CD235a), *HBZ*, *HBE1*), megakaryocyte- (that is, *GP1BA* (CD42b), *CD226*, *PLEK*), macrophage-like (that is, *CD33*, *CD68*, *CX3CR1*) cells, confirming the cell types identified by CFU examination (Extended Data Fig. 10h). Flow cytometry analysis confirmed the presence of erythroid- (CD235ab+), myeloid- (CD33+) and megakaryocyte-like (CD42b+) cells within a CD43+ cell population during weeks 2 and 3. However, these lineages can arise from both primitive and EMP haematopoietic waves (Fig. 5j and Supplementary Fig. 1).

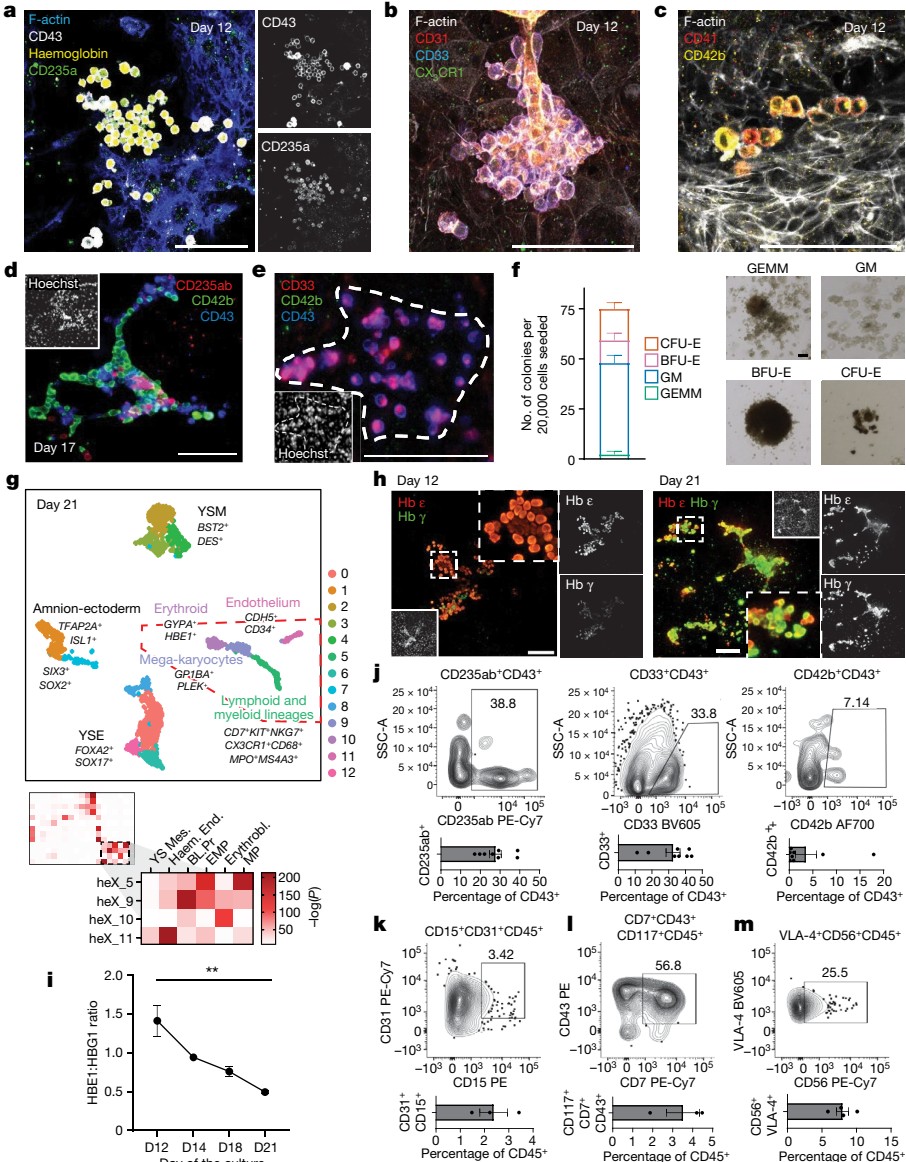

**Fig. 5 | Haematopoietic programme characterization in heX-embryoids.**
**a**, IF of day 12 erythroid-like progenitors. **b**, IF of day 12 myeloid-like progenitors.
**c**, IF of day 12 megakaryocyte-like progenitors. **d**, IF of day 17 multilineage
haematopoietic foci. **e**, IF of day 17 uni-lineage haematopoietic foci. **f**, Results
of a CFU methylcellulose assay seeded from day 21 cultures. Counts from $n = 3$
biological replicates shown. Example colonies are shown. CFU-E, colony
forming unit-erythroid; BFU-E, burst forming unit-erythroid; GM, colony
forming unit-granulocyte, macrophage; GEMM, colony forming unit-
granulocyte, erythrocyte, macrophage, megakaryocyte. **g**, UMAP of day 21
populations. Boxed area shows populations with similarity to in vivo
haematopoietic lineages. Hypergeometric statistical comparison to ref. 23
for these clusters is shown. For hypergeometric tests, we performed the
Benjamini–Hochberg correction for multiple comparisons in all comparisons.
Tests were one-sided. **h**, IF of day 12 and 21 cultures, showing representative
erythroid-like colonies with primitive (high Hb ε, low Hb γ) characteristics on
day 12 and definitive (high Hb γ, high Hb ε) characteristics on day 21. **i**, qPCR

showing change HBE and HBG expression ratio across the indicated days.
**$P = 0.0012$ (confidence interval 95%). $P$ was calculated using one-way ANOVA
with Tukey's multiple comparisons test between days 12 and 21. $n = 3$ biological
replicates sampled per day. **j**, Flow cytometry scatterplots showing erythroid-
(CD235ab[+]), myeloid- (CD33[+]) and megakaryocyte-like (CD42b[+]) populations
within the CD43[+] population of day 21 cultures. Bar plots show cells identified
within the CD43[+] population in eight biological replicates. **k**, Flow cytometry
analysis of week 3 culture for neutrophil-like cells, pregated for CD45[+] cells. The
bar plot shows the percentage of CD31[+]CD15[+] cells in three biological replicates.
**l**, Flow cytometry analysis of week 2 culture for lymphoid-like progeny, pregated
for CD45[+]CD117[+] cells. The bar plot shows the percentage of CD117[+]CD43[+]CD7[+]
cells in three biological replicates. **m**, Flow cytometry analysis of week 3 culture
for natural killer-like cells, pregated for CD45[+]VLA-4[+] cells. The bar plot shows
the percentage of CD56[+]VLA-4[+] cells in four biological replicates. Error bars are
±s.e.m. Scale bars, 100 μm.

We then probed our heX-embryoids for cells similar to those gener-
ated during the definitive phase of yolk sac haematopoiesis but not
the primitive. Transcriptomic analysis of the haematopoietic-like
cluster shows cells co-expressing function-specific genes of neutro-
phils such as *FUT4* (CD15), *MPO* and *PRTN3* (Extended Data Fig. 10h).
The neutrophil-like identity was confirmed in flow cytometry by

identification of a population of CD15[+]CD31[+]CD45[+] cells that have speci-
fied by the third week of culture (Fig. 5k and Supplementary Fig. 2)[54–56].
Although the capacity of the human yolk sac to produce EMP line-
ages, with potency for erythroid, megakaryocyte, macrophage and
granulocyte cell types has been shown, its ability to produce lymphoid
progenitors from within the human yolk sac has only recently started to

be suggested[54,57,58]. CD117 (*KIT*) and CD7 co-expression is prominent in human yolk sac lymphoid lineages[54]. We could identify markers associated with lymphoid progenies and innate lymphoid cell types (that is, *CD7, IL7R, CD3D, NCAM1* (CD56), *NKG7, CTSW, IL2RG*). (Extended Data Fig. 10h). We detected cells expressing CD7+CD43+CD117+CD45+ as early as day 16 and VLA-4+CD56+CD45+ by the third week of culture by using flow cytometry analysis. These cells represent a putative early lymphoid signature and natural killer-like cells, respectively (Fig. 5l,m and Supplementary Fig. 2)[56]. Hence, extra-embryonic-like tissue of heX-embryoids show EMP- and lymphoid-primed multipotent progenitor-like haematopoietic programmes.

## Discussion

heX-embryoid, an hiPS cell-based model of human embryogenesis, contains embryonic epiblast-like and extra-embryonic endoderm- and mesoderm-like components. The presence of these three tissues was sufficient to trigger complex cellular organization. In the first 4 days, heX-embryoids mimic the segregation of amnion from pluripotent epiblast cells, formation of an amniotic-like cavity, development of bilaminar disc-like structures, generation of anterior hypoblast-like cells and a posterior-like pole resembling events associated with CS5-7 of human development (Extended Data Fig. 11). heX-embryoids lack trophoblast, a prerequisite tissue for the formation of placenta. As such, they do not mirror the biology of the full integrated embryo and can provide a possibility for extra in vitro follow-ups with limited ethical concerns. In our analysis, WT clusters after day 5, showed characteristics mainly aligned with ectodermal fates, which is a subject for future studies and optimization (Extended Data Fig. 12a,b).

The emergence of early haematopoietic programmes has not been studied in fine detail in primates and is widely unexplored in the human embryo and its models. At 2–3 weeks, the de novo haematopoietic activity in heX-embryoids recapitulates the emergence of haematopoiesis in the yolk sac of the human embryo at CS7–9. The possibility to offer a multifaceted model for early blood-like development creates significant research prospects to produce hard-to-access populations of cells applicable to human cell therapies. The pattern of haemoglobin expression, the presence of yolk sac haemogenic endothelial-like progenitors and lineages derived from EMP haematopoiesis such as natural killer-like cells[59], together with the existence of granulocyte- and lymphoid-like lineages, indicate that our platform can model primitive and definitive blood formation of the yolk sac. The presence of uni-lineage foci, albeit not the majority, hints at specialized niches directing the differentiation of haematopoietic progenitors. Bona fide yolk sac blood islands are formed by means of the emergence of a lumen resulting from the disappearance of mesodermal cells within areas bordered by early endothelial cells[60]. Although we have not been able to confirm the occurrence of this process in our cultures, the extra-embryonic-like tissue of heX-embryoids develops morphologically relevant haematopoietic foci that show in vivo-like organization of yolk sac tissue layers with extra-embryonic endoderm- and mesoderm-like components.

From an engineering perspective, our platform highlights the power of tissue niche engineering and codifferentiation. Its creation leverages self-organization from 2D to 3D that is directed by establishing tissue boundaries and geometric confinement, vertical tissue growth, cell migration and polarization and tissue cavitation. Hence, it supports a new way to build 3D structures anchored to a 2D surface, leading to improved robustness, efficiency and control. Engineering multilineage fates by the expression of transcription factors enables self-produced environmental cues (for example, laminin, BMP4) and alleviates the need for complex and often supraphysiologic regimes of growth factors.

The extra-embryonic-like differentiation of heX-embryoids is pre-programmed into undifferentiated hiPS cells by means of an inducible genetic switch and mixed with WT hiPS cells; hence, the undifferentiated cell mix can be expanded, cryostored and shipped for use on demand, requiring only 2D culture plates, commercially available medium and addition of a small molecule, Dox (Extended Data Fig. 12c–e). The compatibility with live imaging, the high-throughput format, efficient generation and an easy-to-implement protocol enable establishment and use across different laboratories. These characteristics, coupled with the potential to be produced from hiPS cells with diverse genetic backgrounds (Extended Data Fig. 12f–i), facilitate previously hard-to-execute studies on the early stages of post-implantation and emergence of the human haematopoietic programmes. As such, this model will provide new routes for drug testing, developmental toxicology, tractable disease modelling and the generation of cells for regenerative therapies in a human context.

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

# Methods

## Ethics about the development of heX-embryoids

This research was approved and performed under the oversight of the University of Pittsburgh Human Stem Cell Research Oversight Committee to generate human embryo models from human iPS cell lines (approval number PROTO202300020). The work that has been reported in this study followed 2016 Guidelines for Stem Cell Research and Clinical Translation as well as 2021 ISSCR Guidelines on Ethical Standards for Stem Cell Embryo Model[3,5,61]. heX-embryoids reported in this study are generated from human iPS cells that are derived from human somatic cells such as fibroblasts. The PGP1 and PGP9 hiPS cells were a kind gift from the Weiss Laboratory (Massachusetts Institute of Technology, USA). Consent was obtained within the Personal Genome Project (https://www.personalgenomes.org/). Details related to authentication and mycoplasma testing are denoted in the reporting summary. heX-embryoids are attached to a cell culture dish, lacking an extra-embryonic trophectoderm tissue critical for full integrated embryo development and implantation in the uterine cavity. The yolk sac-like cavity in these embryoids is not closed, and the tissues cannot be harvested for any implantation without substantial disruption of their structures. TBXT[+] posterior-like domains were observed during the study but were not sustained during the development of our system. These features collectively restrict the ability of this model to undergo the full integrated development of a human embryo in vitro and/or implant to support further development of a conceptus in vivo. This study does not use human blastocysts, does not involve the derivation of any human embryonic stem cell lines and does not use samples obtained from fetal abortions. This study does not involve in utero transfer of any human cells or structures into any other species.

## Cell culture

All cells and tissues were cultured in a humidified incubator at 37 °C and 5% $CO_2$. Our hiPS cell lines were cultivated under sterile conditions in mTeSR-1 (StemCell Technologies), changed daily. Tissue culture plates were coated for 1 h at room temperature with BD ES-qualified Matrigel (BD Biosciences) diluted according to the manufacturer's instructions in ice cold DMEM/F-12. Routine passaging was performed by incubating hiPS cell colonies for 5 min in Accutase (Sigma) at 37 °C, collecting the suspension and adding 5 ml of DMEM/F-12 medium containing 10 µM Y-27632, centrifuging at 300g for 5 min and resuspending in DMEM/F-12 supplemented with 10 µM Y-27632 for counting. Cells were seeded at a cell density of 25,000 cells per $cm^2$ for routine maintenance.

## GATA6-engineered cell line generation

The previously generated rtTA expressing PGP1 hiPS cells[62] were transfected using Lipofectamine 3000 (Thermo Fisher Scientific) with Super PiggyBac Transposase (System Biosciences) and the PiggyBac transposon vector with human GATA6-2A-EGFP under control of the tetracycline responsive element promoter. Transfected cells were selected by adding 0.5 mg ml[−1] puromycin to the mTeSR-1 maintenance medium. PGP9 iGATA6-hiPS cells were engineered as explained previously[62]. For the generation of high GATA6-expressing cells (GATA6-hi cell line), the iGATA6 cell line was sorted by using fluorescence-activated cell sorting (one cell per each well of a 96-well plate) in mTeSR-1 supplemented with 10 µM Y-27632 and 2 µM Thiazovivin. The media was replaced on the day after sorting with mTeSR-1 supplemented with 10 µM Y-27632 and 2 µM Thiazovivin. On day 3 after sorting, 125 µl of mTeSR-1 was added to each well. The wells were monitored afterwards for colony formation. On day 6, the wells with a considerable colony were passaged and the amount of GATA6-EGFP was characterized by Dox induction. The GATA6 concentrations were screened on the basis of a high level of EGFP reporter expression. For generating the supplemented cultures, 25% of iGATA6 cells were substituted for GATA6-hi cells at the time of seeding. The same ratios to WT cells were used as before.

## Generation of heX-embryoids

The GATA6-engineered hiPS cells were seeded at either a ratio of 81/5 or 4/1 with rtTA expressing PGP1 hiPS cells either containing or lacking an mKate reporter gene at a total density of 54,000 cells per $cm^2$ in mTeSR-1 supplemented with 10 µM Y-27632. The next day, the medium was changed to mTeSR-1 with 1 µg ml[−1] Dox to induce expression of the GATA6 transgene, and this medium was used for daily replacement for up to 5 days. For experiments that continued beyond 5 days, medium was switched to IMDM on day 6. The following presented experiments were seeded at a 4/1 ratio before optimization: Figs. 1b,c, 2c,d and 3d,e, Extended Data Figs. 1f, 2b–d,g,h, 6a, 7d,e, 8d,e and 12c,d,f–i. The remaining experiments shown were seeded at 81/5. We did not predetermine sample sizes via statistical methods. The number of samples used in each experiment was selected to ensure data consistency and reproducibility, and was based on the available resources. Embryoid samples developed in each culture for each experiment were allocated to control and experimental groups in a random manner at seeding when different conditions (that is, pathway inhibition) were present. We did not conduct blinding.

Seeding of 3D heX-embryoids was performed in AggreWell 400 plates at 24,000 cells in total, targeting 12–20 cells per microwell. Media changes were performed as described above, with the following modification: after the initial media change to mTeSR containing Dox, media changes were performed in accordance with the protocol recommended by the manufacturer, in which 75% of the media is removed and replaced to prevent displacement of aggregates.

## Cryostorage of heX-embryoids

Uninduced cells were incubated with Dispase at 70% confluence for 10 min, or until there was visible lifting of colony edges. Cells were washed twice with DMEM/F-12, then colonies were manually scraped from the plate. Colonies were centrifuged at 300g for 5 min, and then were resuspended in Cryostor 10. Cells were cooled at −80 °C for 24 h before transfer to liquid nitrogen for long-term storage. Cells were stored in liquid nitrogen for at least 24 h before defrosting.

## Signalling pathway inhibition

BMP4 signalling was inhibited by application of 100 ng ml[−1] Noggin into the normal media at day 2 of heX-embryoid culture. This concentration of Noggin was supplemented into the media changed on subsequent days.

## Staining on glass coverslips

Cells were grown on Matrigel-coated 8 mm diameter circular glass coverslips, on 14 mm coverslip bottom dishes (Mattek Corporation) or on eight-well µ-Slide (ibidi). Cultures were fixed for 10 min in 4% paraformaldehyde (Electron Microscopy Sciences) at room temperature. Coverslips were then washed three times with PBS followed by 15 min permeabilization with 0.2% Triton X-100 in PBS. Subsequently the coverslips were washed three times in wash buffer (0.05% Tween-20 in PBS) for 5 min and blocked for 20 min in 200 ml of wash buffer plus 5% normal donkey serum (Jackson ImmunoResearch Laboratories). The primary antibodies were diluted in 5% normal donkey serum in PBS and incubated with the tissues for 1 h at room temperature followed by three washes in wash buffer for 5 min each. The secondary antibodies were diluted in 5% normal donkey serum in PBS and incubated with the tissues for 1 h at room temperature followed by three washes in wash buffer for 5 min each. Afterwards, the 8 mm coverslips were mounted on microscopy glass slides using ProLong Glass Antifade (Life Technologies), cured overnight at room temperature and then sealed with clear nail polish. Coverslips in coverslip bottom dishes or µ-slides were stored in PBS at 4 °C for 3D imaging.

## Image acquisition and processing

Images were acquired using the EVOS M700 automated scanning microscope (M7000 Software Revision v.2.0.2094.0), Leica SP8 confocal microscope (Leica Application Suite X v.3.7.4), Sartorius Incucyte S3 Live Cell Imaging System (software v.v2019B) or Nikon A1 confocal microscope, and processed using Fiji/ImageJ software (National Institutes of Health, NIH)[63]. Any contrast adjustments were made in individual channels and applied evenly across the whole image in that channel. Contrast and colour balance for colour images were applied evenly across the whole image. Colours in Fig. 1b,c and Extended Data Fig. 2a were produced using the look-up tables published in ref. 64. 3D reconstructions were generated using the Nikon A1 confocal microscope to generate z-stacks spanning roughly 100 μm deep into the tissues and using Imaris (Bitplane) to construct a 3D volume from the stacks. Time-lapse videos were initially processed using Fiji/ImageJ, and annotations were added using the Adobe Premiere software. For Supplementary Video 4, a video moving through the z-slices was initially recorded performing this action in the NIS Elements HC software (Nikon), and the video was subsequently cropped and processed.

## Analysis of WT cluster areas and radial expression

Tiled whole-coverslip images were cropped to a central circumscribed square (typically 9,000 μm$^2$) image for analysis. WT compartment analysis was performed using an in-house MATLAB (Mathworks) pipeline (built in MATLAB v.R2020a and run in v.R2022b). In brief, WT compartments were detected programmatically by thresholding the nuclear dye, F-actin stain or mKate marker channel for areas of high signal intensity. Maximum individual compartment area and average cell size were defined manually. Compartments were filtered to remove those with areas close to the defined individual cell area by using the following equation:

$$\text{lower size limit} = \frac{\text{max compartment area}}{10^{\left(\frac{\text{max compartment area} - \text{single cell area}}{2}\right)}}$$

For compartment counting and WT area analysis, characteristics of each compartment were recorded using the regionprops function in MATLAB. Furthermore, the distance to the nearest WT compartment was recorded.

Analysis of marker expression inward from the compartment perimeter was performed by drawing a line a defined distance towards each compartment's centroid and recording the intensity value of each marker at each point within this range. If a line was drawn such that it would intersect the compartment edge (for example, drawing from the outer edge of a concave shape), those values were skipped. Intensity values recorded for each line were averaged to create final per-compartment expression distributions. These distributions were aligned on the basis of the point of maximum CER1 expression when present, and the lengths of the distributions were equalized to the length of the longest border recorded using the imresize function in MATLAB.

## Analysis of WT cluster features

Coverage of WT clusters was evaluated by using widefield images taken of the entire coverslip. Areas in which EGFP expression observed inside the WT clusters was greater than 50% of the cluster area were counted as being covered. Presence or absence of lumens and/or cavities was evaluated manually, and was detected by the presence of visible rings of PODXL, ZO-1 or F-actin expression within WT compartments.

ISL1 expression within the WT disc was recorded by averaging the radial expression values obtained from the protocol above.

Polarity of TBXT and CER1 domains around WT compartments was assessed in the following way: raw values of CER1 and TBXT produced by radial cluster analysis were normalized to cell density by dividing by the F-actin intensity measured at each point. The rolling average of CER1 and T values at all points within one-eighth of the disc radius in either direction away from a given point around each disc was computed for every point recorded. Positive polarity for each marker was assessed by comparing the minimum and maximum rolling average values for a difference of greater than 0.1 normalized intensity (a.u.). For islands with positivity, the radial index of the highest averaged quadrant value of CER1 was compared to the radial index of the highest average quadrant value of TBXT: a WT compartment with an average peak of TBXT between 135° and 225° away from the CER1 peak was considered to be anti-polar; a WT compartment with a peak of TBXT within 45° of the CER1 peak was considered to be syn-polar. Polarity type assignments per-island were verified manually by eye, and islands with several poles or ambiguous polarization were excluded from analysis.

## Analysis of haemogenic endothelial, haemoglobin and haematopoietic marker genes

Whole-coverslip IF images of each marker were cropped into a square, then divided into equivalent quadrants for ease of analysis. WT cluster masks were created by manual thresholding and creation of a binary image using Fiji/ImageJ. These binary images were then used to remove WT clusters from the images using the Image Calculator function applied to the individual channel images. The resulting images were evaluated by using a custom pipeline in CellProfiler[65]. Scatterplots shown were created using the mean object intensity values for each detected cell. CD43 bar graphs were produced by counts of the number of objects detected by the IdentifyPrimaryObjects function applied to the CD43 channel in isolation. The resulting images were evaluated by using a custom pipeline in CellProfiler[65]. Scatterplots shown were created using the mean object intensity values for each detected cell. CD43 bar graphs were produced by counts of the number of objects detected by the IdentifyPrimaryObjects function applied to the CD43 channel in isolation.

Haemoglobin expression was evaluated in a similar manner with the following additions: the IdentifyPrimaryObjects function was applied to both the haemoglobin ε and haemoglobin γ channels individually, then the objects were combined by using the RelateObjects function. To produce the scatterplot showing the expression of each marker, the mean object intensity values for each marker were divided by the mean object intensity value for a pan-haemoglobin stain on each coverslip. These normalized values were used for the creation of the graph. For the bar graph showing the distribution of ratios of the markers on each day, the mean object intensity value of haemoglobin ε was divided by the mean object intensity value for haemoglobin γ.

CD34$^+$ area was assessed through Fiji/ImageJ. The cropping and WT cluster masking steps above were followed. Subsequently, the threshold command was used to select areas positive for CD34, and thresholded area was assessed by using the measure command. The area of the threshold was then divided by the total cropped image area to determine the percentage CD34 coverage for each image.

Haematopoietic foci were analysed manually by eye from tilescan images of individual stained wells from a 48-well plate. Clusters of more than 20 cells expressing the marker types assessed with a clearly negative peripheral border were manually partitioned as a single focus, then were evaluated on the basis of the markers expressed within the cells of that focus. A uni-lineage focus was identified if there were fewer than three cells within the focus that did not express the given marker. Because CD43 is a progenitor and pan-haematopoietic marker, cells within the uni-lineage focus that also expressed CD43 were still counted as uni-lineage so long as there was an absence of cells that were positive only for CD43 and negative for the marker of interest within that focus. Multilineage foci (expressing two or more markers) were identified if three or more of the cells expressing each of the given markers could be counted in each area.

## Quantitative analysis of z-distribution of haematopoietic foci-related markers

Confocal z-stack images were captured near to the centre of each identified area. For each stack, the bottom was defined as the lowest z-index where an in-focus marker could be identified. The sum of the pixel intensity values from all pixels within each slice above the bottom index were summed, then divided by pixel number. The resulting values at each corresponding z-index were then averaged between all samples identified; these values were then converted to a percentage of the maximum value to produce the graphs shown.

## Enzyme-linked immunosorbent assays

Samples were assayed for AFP and APOA1 using commercially available enzyme-linked immunosorbent assay kits (abcam). Sample dilutions were optimized to attain detection in the linear range of the standard curves for each individual assay.

## RT–qPCR

Three independent cultures of embryoids per each day of collection were lysed with Trizol (Invitrogen) for RNA extraction. RNA extraction was done using Direct-zol RNA Miniprep Kit (Zymo Research). Following complementary DNA (cDNA) synthesis (Thermo Fisher), equal amounts of cDNA per sample were analysed for gene expression using QuantStudio 3 Real-Time PCR Systems (Applied Biosystems). 18S Ribosomal RNA was used as the endogenous control and the $2^{-\Delta\Delta CT}$ method was used for calculation of relative gene expression. The sequences for the used primers are shown in Supplementary Table 1.

## 10X Genomics sample preparation for next-generation sequencing

Samples were prepared as described by the 10X Genomics Cell Multiplexing Oligo Labelling for Single-Cell RNA Sequencing Protocols for cells with more than 80% viability. heX-embryoids were acquired in single-cell suspension by incubation with Accutase for 20 min at 37 °C. The cell suspension was passed through a 40 μm strainer to remove aggregates. Each sample suspension was then centrifuged at 300g for 5 min at room temperature. The supernatant was manually aspirated from each of the samples using a P1000 pipette. The cell pellets were each resuspended in 1 ml of PBS + 0.04% BSA, and the samples were centrifuged at 300g for 5 min at room temperature. The samples were then resuspended, targeting $1 \times 10^6$ cells per ml on the basis of expected densities of cells each day, and the cell suspension was passed through a 40 μm strainer again. Each cell suspension was counted using a hemocytometer, and a volume of cell suspension was removed as necessary to adjust the final cell count to $1 \times 10^6$ cells. The volumes were then adjusted to 1 ml by adding a further PBS + 0.04% BSA to replace the removed volumes.

**For multiplexing oligo labelling.** Samples were then centrifuged at 300g for 5 min at room temperature. Supernatant was aspirated manually with a P1000 pipette. Samples were resuspended in 50 μl of CellPlex Multiplexing Solution (10X Genomics), with unique multiplexing oligo solutions assigned to each sample. A maximum of 12 samples were labelled in parallel. Samples were incubated for 5 min, starting after the oligo solution was added to the last sample.

Following labelling, 1.95 ml of 1× PBS + 1% BSA was added to each sample, and the solution was mixed thoroughly. Each sample was centrifuged at 300g for 5 min at 4 °C. The supernatant was manually aspirated with a P1000 pipette, taking care to leave less than 10 μl of supernatant remaining in each sample when possible. The samples were resuspended in 2 ml of 1× PBS + 1% BSA and were mixed thoroughly to wash. This wash and centrifugation step was repeated twice more; at the last resuspension, enough wash buffer was added to reach a cell count of $1 \times 10^6$ cells per ml, assuming 50% cell loss from the count at the end of the first protocol. Labelled cell suspensions were then put on ice for transfer to the Pitt Single-Cell Core for library creation.

Following a final counting and viability analysis, cells and 10X Genomics reagents were loaded into the single-cell cassette, with a target of 25,000 single cells for analysis, accounting for predicted cell loss and doublets resulting from multiplexing as laid out in the user guide for the Chromium Single-Cell 3′ Reagent Kit (10X Genomics). After generation of gel in-bead emulsions, the cDNA library was prepared by Pitt Single-Cell Core staff following the appropriate steps determined by the 10X Genomics user guide. Libraries were sent to the UPMC Genome Centre for sequencing by a NovaSeq S4-200 for an intended read depth of 100,000 reads per cell with 150 bp paired end reads. Our downstream analysis from the sequencing data yielded between 40,000 and 150,000 mean reads per cell in different samples.

## scRNA-seq sample processing and quality control

The 10X Genomics CellRanger pipeline was used to align reads to the reference genome (GRCh38.84) appended with transgene sequences, to assign reads to individual cells and to estimate gene expression on the basis of unique molecular identifier (UMI) counts.

We used CellRanger (v.6.1.2) 'multi' pipeline to demultiplex the libraries and quantify the gene expressions from raw fastq files. We first downloaded the FASTA file and GTF file of the human reference genome (GRCh38.84) and constructed the genome index by using the mkref command available in the CellRanger tool kit. This genome index was augmented by a few exogenous sequences (sequence names) to enable alignment against the exogenous sequences. We specified the number of expected cells in the config file as 24,000 for the batch containing the day 21 scRNA-seq and 30,000 for the batch containing the day 5 and lower scRNA-seq. The multi command in the CellRanger tool kit was then run to perform demultiplexing into GEX libraries and CellPlex libraries and quantification of gene counts. Other than the number of expected cells specified above, we used the default parameters of the multi command. We used the filtered gene count output for our downstream analyses (located under the CellRanger output folder 'outs/per_sample_output/sample_name/count/sample_feature_bc_matrix').

Seurat V4 was used for downstream processing of the scRNA-seq data[66]. Single-cell data were excluded on the basis of a high mitochondrial genome transcript ratio and either high or low feature or UMI counts. Genes with UMI counts in fewer than five cells were removed from consideration. For scRNA-seq data processing and cluster analysis using Seurat, we used the following general standardized pipeline for processing of the CellRanger output: SCTransform regressing percentage mitochondrial genes, principal component analysis and clustering. Jackstraw plots and permuted P values were used to assist in determining the optimal number of principal components needed to summarize the datasets without losing a significant amount of variation. The quality of a range of clustering resolution values was assessed using enrichment of cluster marker genes (genes differentially upregulated in a given cluster relative to all other clusters) with embryo cell type-specific genes. As a quality check, principal components and resolution metrics were modulated to yield fewer or extra clusters to confirm that chosen parameters resulted in the most biologically relevant clustering. Visualization was achieved by the use of uniform manifold approximation and projection (UMAP) plots identifying cells, clusters and selected gene expression in each cell, as well as heatmaps and violin plots showing the expression level of genes by cluster. Stacked violin plots were created using the scCustomize package in R[67].

Subclustering the WT in the D3 and the endothelial cluster in the D21 dataset was achieved by subsetting the cluster of interest, then computing the distance between rows of the data matrix using Euclidean measurement (dist function in R). Hierarchical clustering was applied using this distance matrix (hclust in R), and used to create a dendrogram of the distance explained by different numbers of subclusters. The number of subclusters was selected on the basis of the highest

level of distance observed and applied to the data (clustercut in R). The cluster was then added back to the rest of the dataset and markers were determined.

## Gene set enrichment analysis

Gene set enrichment analysis was performed between cluster 5 and cluster 2 of the merged D0–D5. The differentially expressed markers in cluster 5 as compared to cluster 2 were obtained by using the FindMarkers() function in Seurat. All marker genes with a $\log_2$ fold change above 1 were used as an input to the Enrichr web server[68,69]. The top ten pathways identified in the KEGG 2021 and Reactome 2022 pathways were used. Top pathways were identified using the Combined Score, which was computed by the Enrichr site and was defined as the log of the $P$ value from the Fisher exact text multiplied by the $z$-score of the deviation from the expected rank.

## Comparison of embryoid culture clusters to yolk sac and definitive endoderm clusters

**Generating Jaccard similarity-based scores.** We compared the heX-embryoid cluster markers to yolk sac endoderm and proliferating definitive endoderm (DE(P)) with the following steps:

1. We denote the unique overlap of markers between the embryoid cluster and DE(P) cluster as $A$ and the unique overlap of markers between the embryoid cluster and yolk sac endoderm cluster as $B$ and marker genes for embryoid cluster as $C$; marker genes for DE(P) as $D$ and marker genes for yolk sac endoderm as $E$.
2. We computed the Jaccard similarity indices (JSI): JSI for embryoid/DE(P) is $j_1 = A/(C \cup D)$; JSI for embryoid/YS endoderm is $j_2 = B/(C \cup E)$. The JSI basically adjusts the overlap size by the two marker cluster sizes.
3. We then measure the significance of $j_2 - j_1$, if $j_2$ is significantly larger than $j_1$, then we may say the unique overlap of embryoid/YS endoderm is significantly greater than the unique overlap of embryoid/DE(P).
4. To compute the significance level, we performed a random sampling of three sets of genes (of the same sizes as the clusters) from the genome, which generated an empirical distribution of $j_2 - j_1$. On the basis of this empirical distribution, we then computed the empirical $P$ value for the significance and visualized the $P$ value scores by $-\log_{10}(P)$.

**Volcano plot.** We computed the expression fold changes of the yolk sac endoderm cluster with respect to the DE(P) cluster from ref. 23 using the function 'FindMarkers()' in Seurat package (v.4.0.1). The fold changes were then used to produce the volcano plots. Marker genes of each the embryoid clusters were labelled in the volcano plots.

## Statistical analysis

Statistical tests were noted in the legend of each corresponding figure panel. For the similarity quantification between lists of marker genes, we quantify the similarity between two lists of marker genes A and B by using the hypergeometric test for overrepresentation. This is equivalent to a one-tailed Fisher's exact test. Data obtained for comparison were taken from refs. 23–26.

For the statistical tests used in the bar graphs when calculating the $P$ value for Extended Data Fig. 6f, we used an unpaired, two-tailed $t$-test.

The statistical test for *HBE*/*HBG* expression used to calculate the $P$ value for the comparison of *HBE*/*HBG* expression level across days of culture in Fig. 5i was a one-way analysis of variance (ANOVA) test. Tukey's test was used as the post hoc test for multiple comparisons.

## In vitro CFU assays

Cells from day 21 heX-embryoids were dissociated into single-cell solution by application of Accutase (StemCell Technologies) for 30 min. CFU assays were performed using methylcellulose containing media (MethoCult SF H4636, StemCell Technologies) according to the manufacturer's instructions. After 14 days in culture, colonies were analysed by image acquisition using light microscopy.

## Flow cytometry analysis

To prepare a single-cell solution of heX-embryoid, the culture was treated for 45 min with Collagenase C solution (3 mg ml⁻¹ StemCell Technologies) followed by 15 min of treatment with Accuatse (Sigma). FC block solution (Thermo Fischer) was added to the samples followed by 10 min of incubation on ice. Next, the antibody mix (final dilution of 1:400) was added to the samples followed by 30 min incubation on ice. Cells were analysed using an LSR II flow cytometer (BD Bioscience) using 7-AAD (BD Pharmingen) for dead cell staining. The antibodies used were CD34-APC (Clone 581, Biolegend), KIT-BV421 (Clone 104D2, BD Bioscience), CD43-PE (Clone CD43-10G7, Biolegend), CD235ab-PE-Cy7 (Clone HIR2, Biolegend), CD33-BV605 (Clone P67.6, Biolegend), CD42b-AF700 (Clone HIP1, Biolegend), CD7-PE-Cy7 (Clone 4H9/CD7, Biolegend), CD45-Pacific Blue (Clone HI30, Biolegend), CD45-APC-Cy7 (Clone HI30, Biolegend), CD45-APC (Clone HI30, Biolegend), CD31-PE-Cy7 (Clone WM59, Biolegend), CD15-PE (Clone HI98, Biolegend), CD56-PE-Cy7 (Clone MEM-188, Biolegend) and CD49d(VLA-4)-BV605 (Clone 9F10, Biolegend). Back-gating, controls and analysis steps can be seen in Supplementary Information 1 and 2.

## Statistics and reproducibility

All experiments were repeated in at least three biological replicates with similar results, except where indicated in the legends or in the Methods.

## Reporting summary

Further information on research design is available in the Nature Portfolio Reporting Summary linked to this article.

## Data availability

The sequencing reads and single-cell expression matrices for all the scRNA-seq data are accessible with Gene Expression Omnibus accession number GSE247111. For hypergeometric analysis, we used publicly available data from Tyser et al.[23] processed the peri-implantation dataset E-MTAB-9388, Xiang et al.[24] GSE136447 and Ma et al.[26] GSE130114. Source data are provided with this paper.

## Code availability

The source code used for hypergeometric analysis is available at GitHub: https://github.com/AmirAlavi/GATA6_R.

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

**Acknowledgements** This work was supported by a fund from the Department of Pathology at the University of Pittsburgh. We acknowledge support from the Center for Biological Imaging and the University of Pittsburgh Flow Cytometry Core facilities. Characterization of haematopoietic cellular subsets was partly supported by an R01 from the National Heart Lung and Blood Institute (grant no. HL141805) to M.R.E. Development and characterization of the iPS cell lines used in these studies were partly supported by an R01 from the National Institute of Biomedical Imaging and Bioengineering (grant no. EB028532) and National Science Foundation award no. 2134999 to M.R.E. The computational analysis of published embryo

datasets was partly supported by NIH grant nos. 1R01GM122096, OT2OD026682, 1U54AG075931 and 1U24CA268108 to Z.B.-J. This work is also partly supported by the Pittsburgh Liver Research Center (grant no. NIHNIDDK P30DK120531), the Cellular Approaches to Tissue Engineering and Regenerative Medicine Training grant (grant no. T32 EB001026) and grant no. NIH 1S10OD019973-01. We thank A. Aryannejad for helping in the flow cytometry experiments, and A. Sutherland, C. Hislop and R. Porterfield for helpful discussion during this work. BioRender.com was used to create a subset of the schematics.

**Author contributions** J.H. and M.R.E. conceived the study. M.R.E., Z.B.-J., B.S. and S.K. supervised the project. J.H., M.R.E., S.K., A.A., K.K.F., J.V., R.L., D.S. and Z.B.-J. conceived the methodology for experiments. J.H., R.S., K.K.F., J.V., R.L., M.N.T. and M.R. performed heX-embryoid experiments. J.H., R.S., K.K.F., T.M., M.N.T., S.W. and D.S. performed imaging and analysis. J.H., A.A., K.K.F. and Q.S. performed the bioinformatic analysis of scRNA-seq datasets. R.L. and K.K.F. performed flow cytometry and analysis. J.H., A.A., K.K.F., J.V., R.L., D.S. and M.R.E. performed visualization of the data. M.R.E., S.K., Z.B.-J. and J.H. acquired funding for the project. J.H., K.K.F. and M.R.E. wrote the original draft of the manuscript. J.H., R.S., K.K.F., B.S.,

S.M.C.d.S.L., A.A., J.V., S.K., Z.B.-J. and M.R.E. were involved in editing and input to the final draft of the manuscript.

**Competing interests** The University of Pittsburgh has a pending patent on behalf of the inventors J.H., R.S., S.K. and M.R.E., application no. PCT/US2023/067404, covering the derivation and generation of cell and extracellular products shown in this paper. All other authors declare no competing interests.

**Additional information**
**Correspondence and requests for materials** should be addressed to Mo R. Ebrahimkhani.

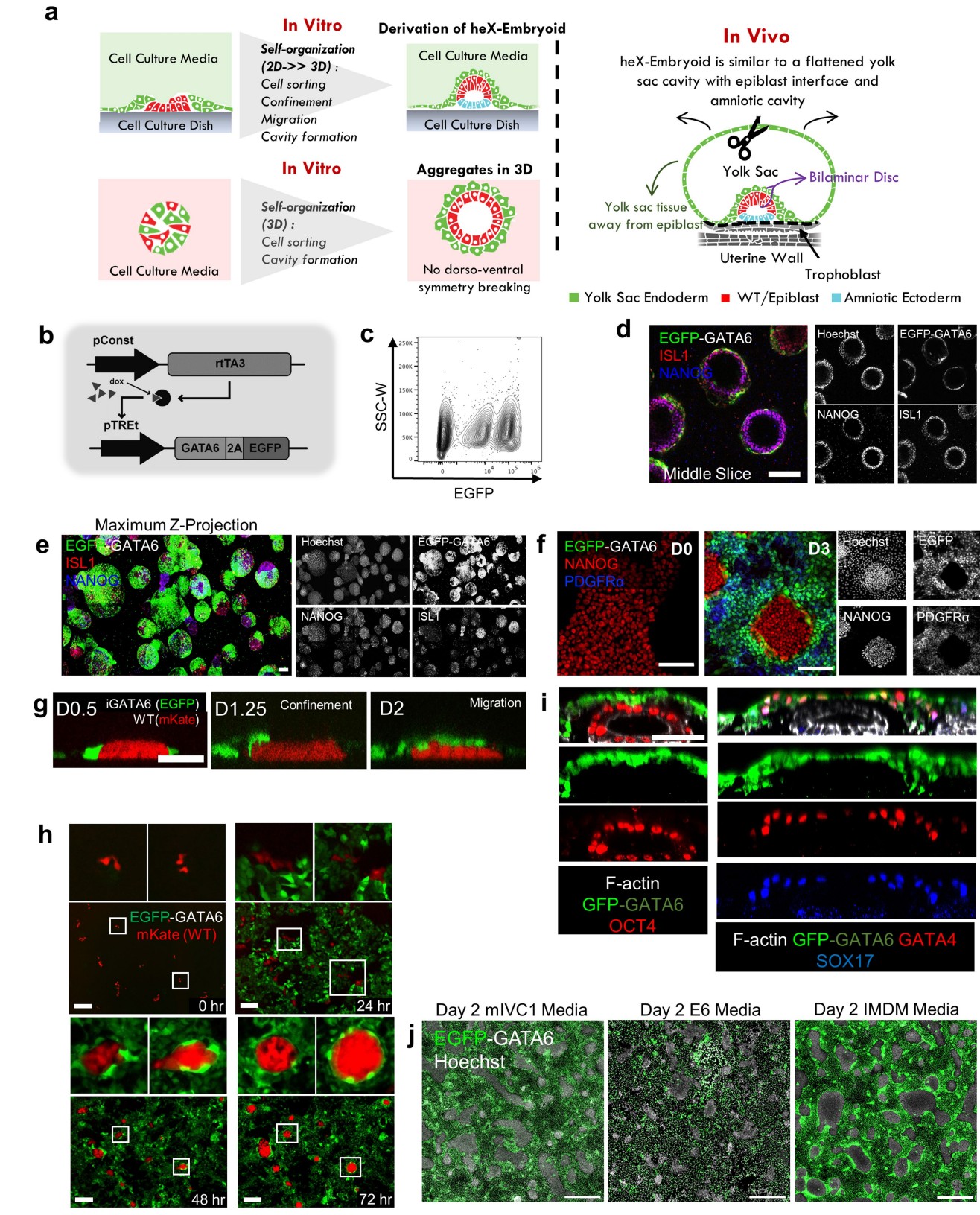

**Extended Data Fig. 1** | See next page for caption.

**Extended Data Fig. 1 | Fate acquisition, sorting, and symmetry breaking following GATA6 induction.** (**a**) Schematic depicting u-Embryoid development in 3D versus from 2D > > 3D in comparison to embryo morphology. heX-embryoid is similar to a flattened yolk sac cavity with epiblast interface and amniotic cavity. (**b**) The gene circuit used to create inducible GATA6-expressing iPSCs. pConst is a constitutively active promoter. (**c**) Heterogeneity of EGFP (GATA6) activation in iGATA6 cells, detected via flow cytometry analysis. Higher gene circuit copy numbers lead to higher expression level of EGFP and GATA6. (**d**) 3D culture of iGATA6 and WT showing co-expression of the amnion marker ISL1 and the pluripotency marker NANOG spread fully throughout the WT layer without D-V polarization. Middle slice shows the development of a central lumen. Scale bar = 100 μm. (**e**) 3D culture of iGATA6 and WT showing expression of the pluripotency marker NANOG throughout the WT layer but inconsistent ISL1 expression in notable subset of these 3D tissues. These spheres do not exhibit polarization. Scale bar = 100 μm. (**f**) Immunofluorescence images of fixed cultures demonstrating cell organization of iGATA6 (green) and WT (NANOG) hiPSCs between day 0 and day 3 after GATA6 induction. PDGFRα rises within the iGATA6 cells as they acquire a more yolk sac endoderm-like morphology. Scale bar = 100 μm. (**g**) Time lapse images showing the initial confinement of red WT cells with green iGATA6 cells, followed by migration of iGATA6 cells over the WT disc before day 2 after heX-embryoid induction. Scale bar = 50 μm. (**h**) Time-lapse images of a single position within the iGATA6/WT co-culture from day 0 to day 3 after GATA6 induction. The top cropped images correspond to positions within the white boxes in the images below. Scale bar = 200 μm. (**i**) Z-slices of two representative embryoids showing localization of OCT4, GATA4, and SOX17 within the bilaminar disc-like area of embryoid culture, as well as development of a central lumen by day 5. Scale bar = 50 μm. (**j**) Switching media away from mTeSR on Day 0 resulted in substantial cell death. Testing media changes starting on day 2 resulted in: modified IVC1 (mIVC1) causing limited iGATA6 migration and subsequent patterning; Essential 6 (E6) media causing substantially lacking disc-like morphology and poor boundary formation; IMDM media at day 2 of culture showing WT disc formation with limited iGATA6 migration. Scale bars = 500 μm.

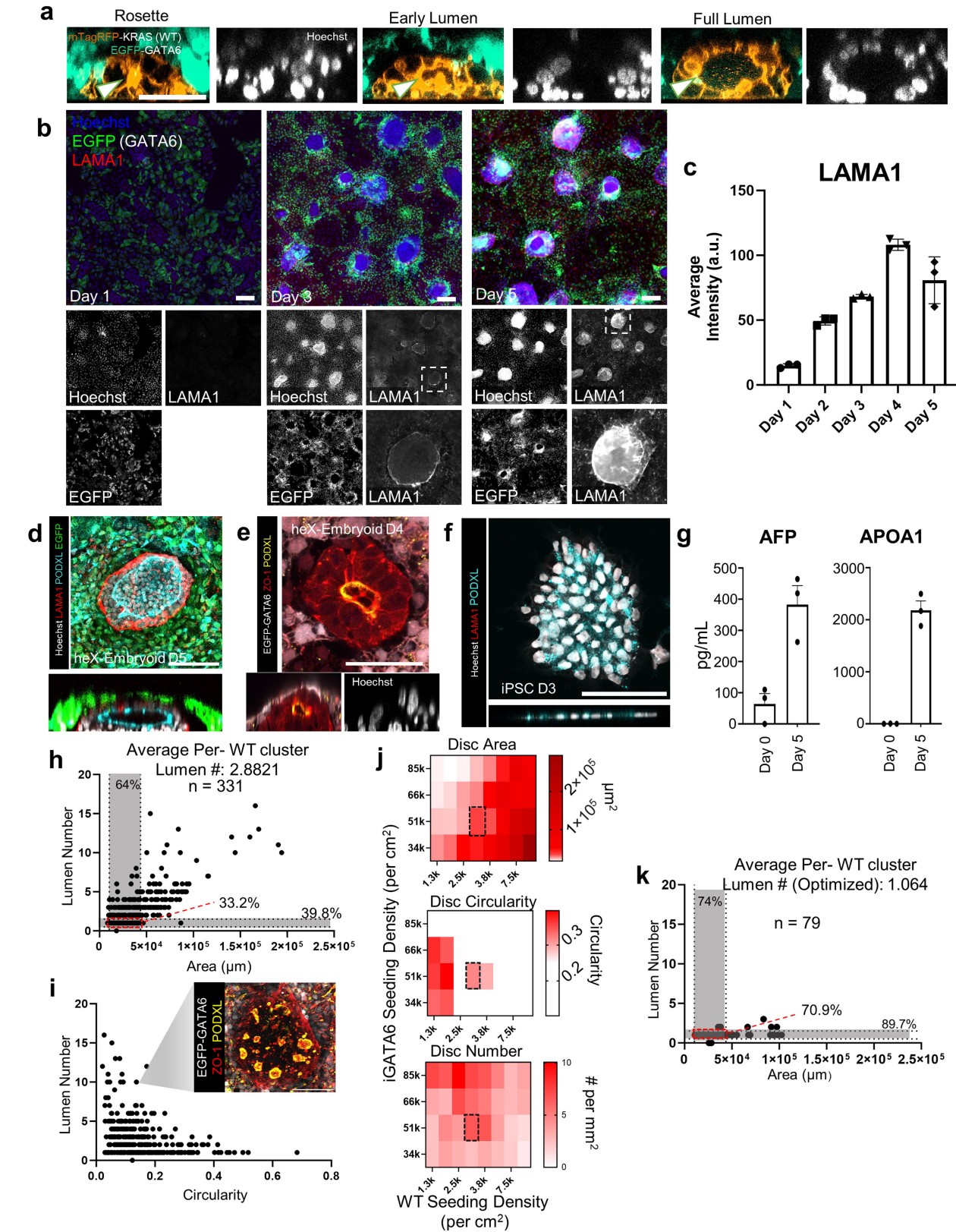

**Extended Data Fig. 2 |** See next page for caption.

**Extended Data Fig. 2 | Lumen development and optimization within WT cluster.** (**a**) Embryoids at different stages of lumenogenesis. (**b**) Immunofluorescence images showing dynamic of LAMA1 deposition on days 1, 3, and 5 of embryoid development after induction. Dotted boxes show the areas of inset in the fourth panels of the day 3 and day 5 images. (**c**) Time course graphs showing the increase in LAMA1 signal in immunofluorescence images over the first five days cultures. n = 3 randomly sampled areas in one round of experiments. Error bars represent mean ± s.e.m. (**d**) Immunofluorescence image showing the deposition of laminin around a WT cluster with a central lumen as well as polarization of PODXL. (**e**) Immunofluorescence showing horizontal and lateral slices of a representative WT cluster with polarization of PODXL and ZO-1 towards a central lumen. (**f**) A representative cluster of WT iPSCs at day 3. No laminin deposition is observable in the vicinity of the cluster. (**g**) ELISA comparing secreted AFP and APOA1 detected on D0 and D5 after GATA6 induction with Dox. n = 3 biological replicates. Error bars represent mean ± s.e.m. (**h**) Distribution of WT cluster areas versus the number of lumens observed in heX-embryoids with iGATA6 coverage. Shaded area indicates the areas of the bilaminar disc between E9 and E17 as recorded by refs. 21 and 22; 64% of discs observed have areas that fall into this range of values, and 39.8% have a single lumen. 33.2% fall into both categories. n = 331 total. (**i**) Distribution of WT cluster circularity versus the number of lumens observed in the embryoids with iGATA6 coverage, with a representative image of an embryoid with the indicated characteristics. n = 331. (**j**) Heatmaps displaying the average area, circularity, and resulting disc number of the embryoids resulting from different initial seeding densities of iGATA6 and WT cells. The dotted box shows the optimized seeding density used for the experiments. (**k**) Distribution of WT cluster areas versus the number of lumens observed in heX-embryoids with iGATA6 coverage when seeded at the optimized ratio indicated in J. Shaded area indicates the areas of the bilaminar disc between E9 and E16-19 as recorded by refs. 21 and 22; 74% of discs that fall into this area range. 89.7% of discs have a single lumen. 70.9% of discs fall into both categories. n = 79 total. Scale bars = 100 μm.

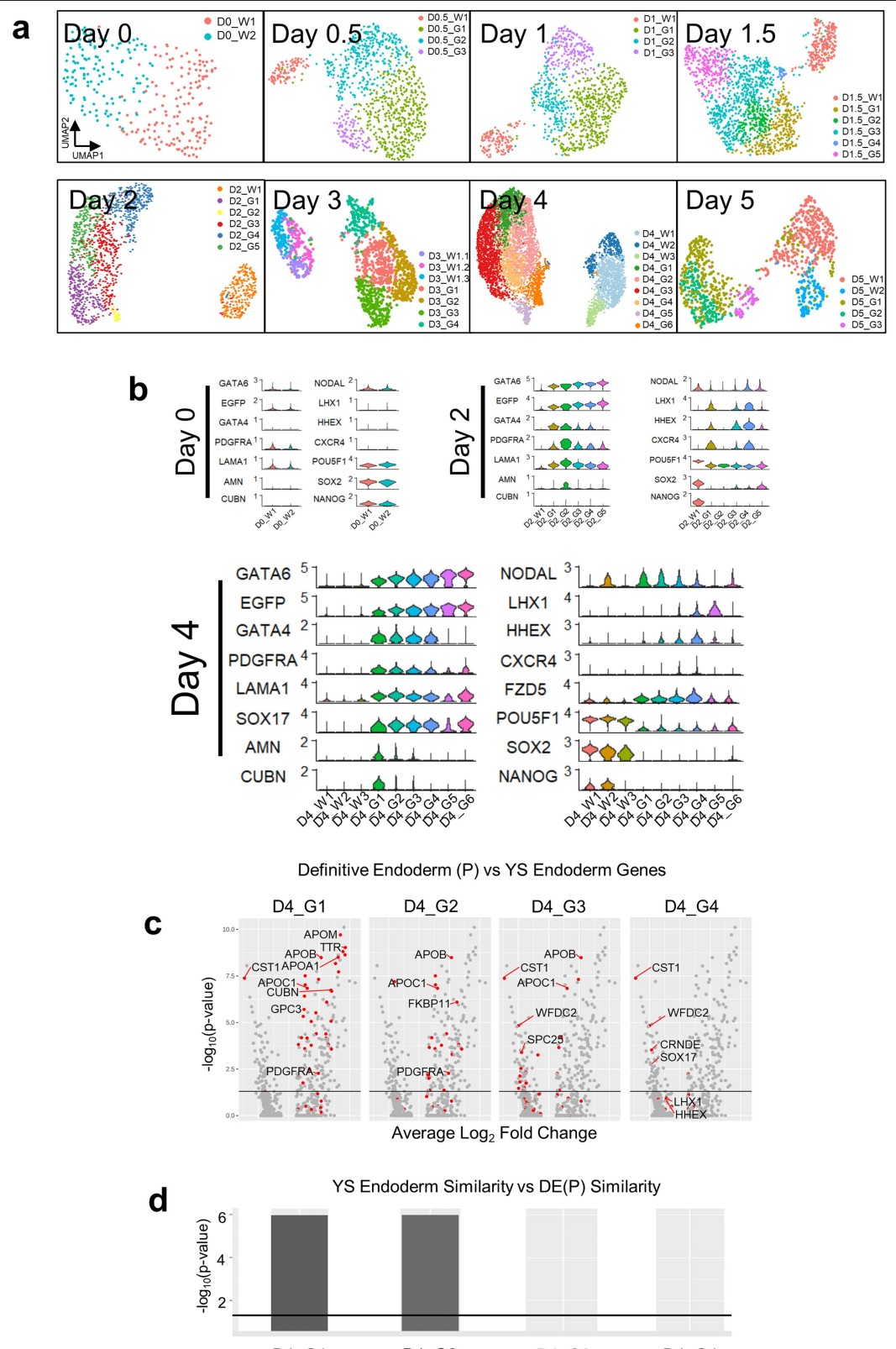

**Extended Data Fig. 3** | See next page for caption.

**Extended Data Fig. 3 | Single Cell RNA-seq analysis and clustering per day (day 0 to 5).** (**a**) Individual UMAP projections and clustering for each time point recorded through day 5. (**b**) Violin plots showing a curated set of genes in heX-embryoid day 0, day 2, and day 4 clusters. Embryoid clusters are ordered by lowest to highest GATA6 expression level. "W" clusters are clusters with putative wild-type lineage; "G" clusters are clusters with putative iGATA6 lineage. (**c**) Volcano plots showing the differentially expressed genes between the DE(P) identities (left) and YS Endoderm (right) from ref. 23. Markers highlighted in red are genes expressed in the respective day 4 embryoid cluster. A subset of notable genes is labeled. P-values were computed by the Wilcoxon rank sum test and were adjusted for multiple comparisons via Bonferroni correction. These tests were performed as two-sided tests. (**d**) Similarity to YS Endoderm versus DE(P) within the embryoid clusters with endoderm identity. Black line indicates a significance threshold of $p = 0.05$. P-values were computed from the empirical distribution (please see Methods) and were adjusted for multiple corrections by the Benjamini Hochberg (BH) method. Tests were performed as one-sided tests.

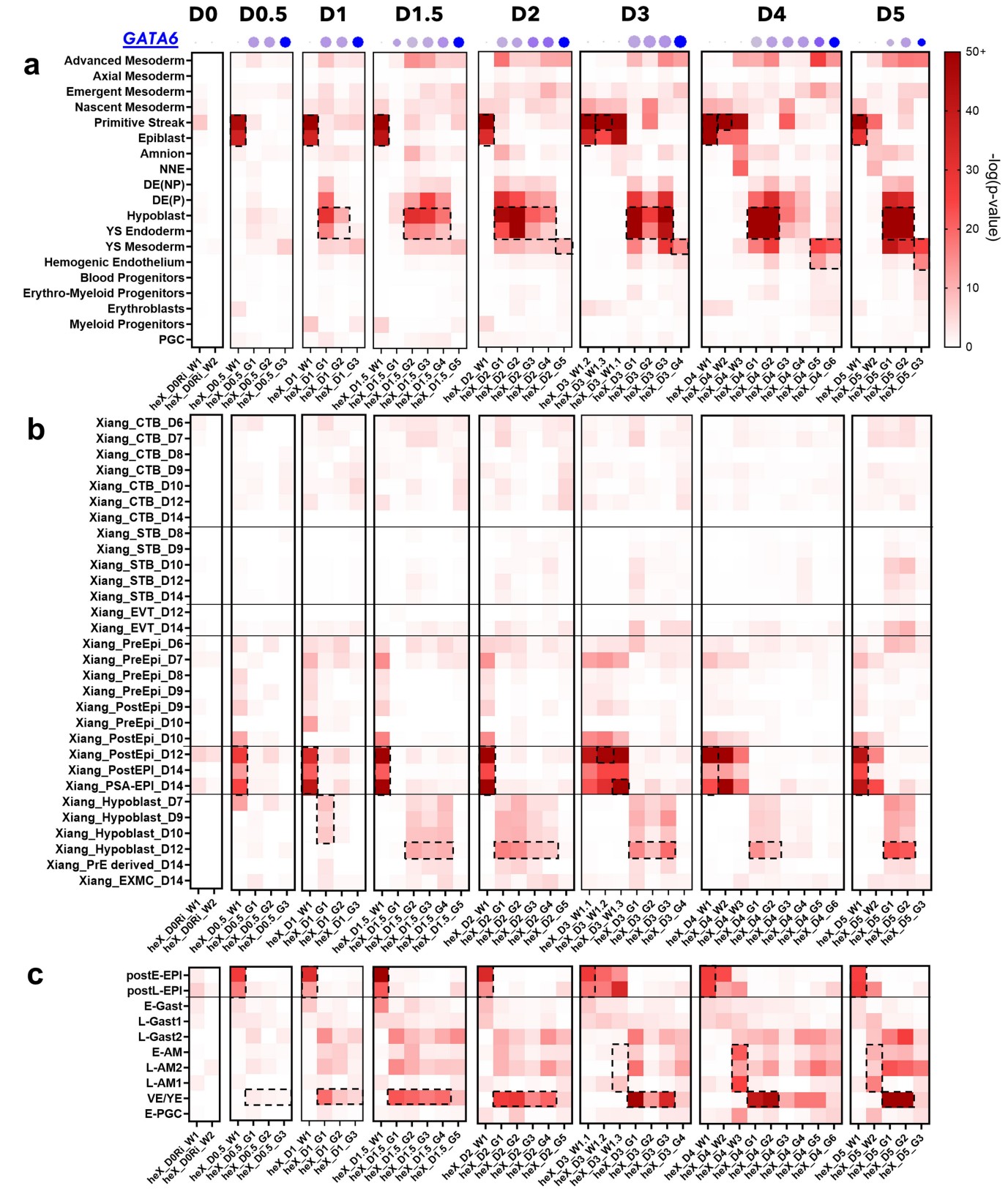

**Extended Data Fig. 4 |** See next page for caption.

**Extended Data Fig. 4 | Hypergeometric statistical comparison of heX-embryoid time points to human and NHP embryo data.** (**a**) Hypergeometric statistical comparison of each embryoid day to the annotated human embryo populations from ref. 23. Blue dots above each column indicates the relative GATA6 expression level of each population indicated per day. (**b**) Hypergeometric statistical comparison of each embryoid day to the annotated human embryo populations from refs. 24,25. Scale used is the same as shown in the panel A. (**c**) Hypergeometric statistical comparison of each embryoid day to the annotated cynomolgus embryo populations from ref. 26. Scale used is the same as shown in panel A. heX-embryoid clusters correspond to those in the individual day-by-day clustering in Extended Data Fig. 3 and are ordered from left to right on the x-axis by lowest to highest GATA6 expression level. "W" clusters are clusters with putative wild-type lineage; "G" clusters are clusters with putative iGATA6 lineage. Dotted outlines indicate fate comparisons of the most interest for each cluster. Abbreviations from other datasets: NNE = Non-neural ectoderm, DE (P) = Definitive endoderm (proliferative), DE (NP) = Definitive endoderm (not proliferative), YS = Yolk sac, PGC = Primordial germ cell, CTB = Cytotrophoblast, STB = Syncytiotrophoblast, EVT = Extravillous trophoblasts, PSA-EPI = Primitive streak anlage in the epiblast, EXMC = Extraembryonic mesoderm cells, E- = Early, L- = Late, Gast = Gastrulating cells, AM = Amnion, VE/YE = Visceral endoderm/Yolk sac endoderm. Elements of these graphs are also shown in Fig. [1f-g].

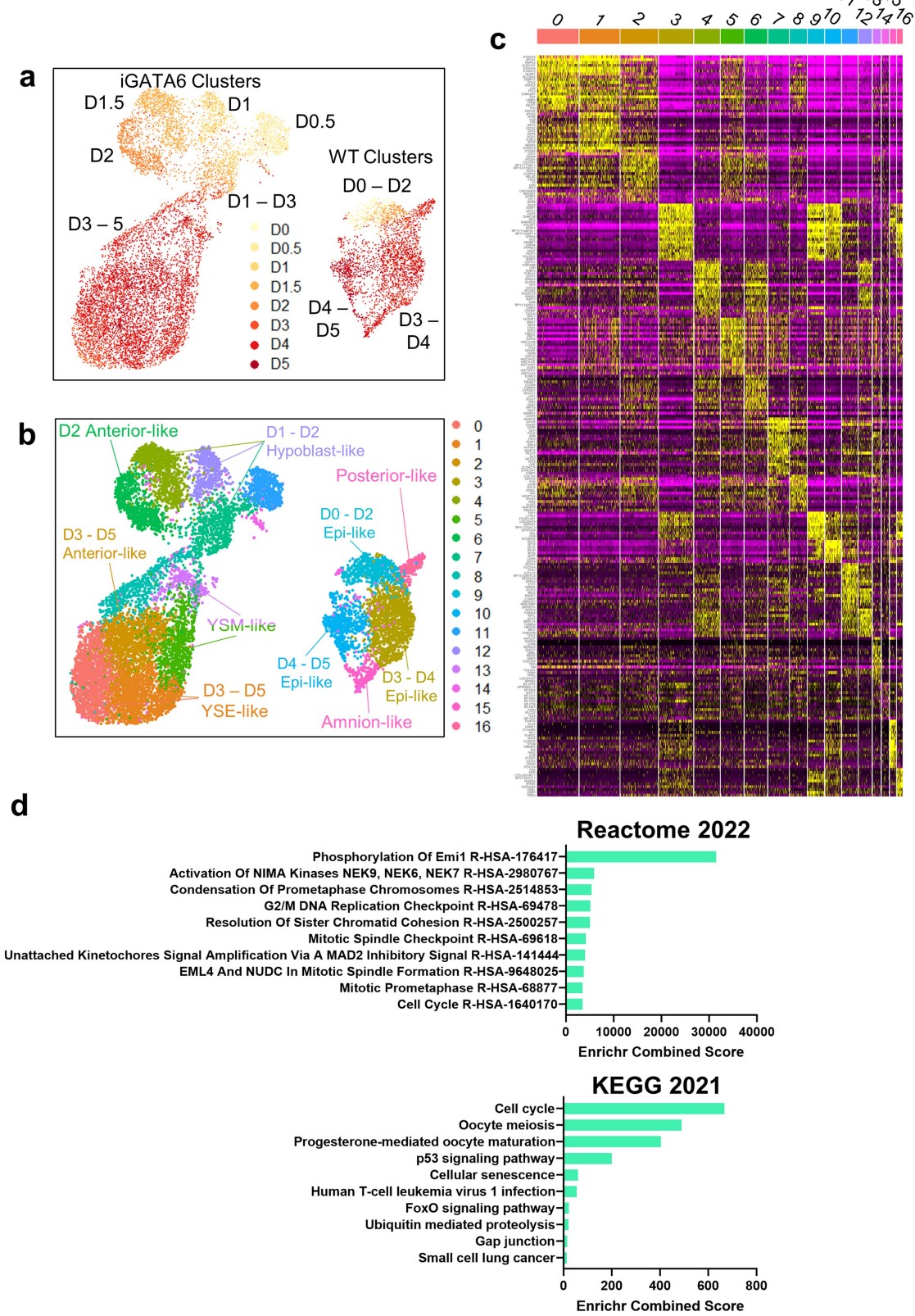

**Extended Data Fig. 5** | See next page for caption.

**Extended Data Fig. 5 | Merged clustering of RNA-seq Day 0 – Day 5 of the embryoids.** (**a**) UMAP showing the merged clusters annotated by day. (**b**) UMAP showing the merged clusters with unsupervised clustering applied. (*repeated figures*: The WT clusters are shown in Fig. 2a Cluster 14 is the amnion-like cluster shown in Fig. 2b; Cluster 6 is the D2 anterior-like cluster shown in Fig. 3a; Cluster 2 is the D3-5 anterior-like cluster shown in Fig. 3a; and Cluster 15 is the posterior-like cluster shown in Fig. 3b.). (**c**) Heatmap showing the top 20 genes corresponding to each cluster of the merged day 0 through day 5 dataset.

(**d**) Gene set enrichment analysis on the genes with greater than 2-fold upregulation in the D2 anterior-like cluster (5) as compared to the day 3–5 anterior-like cluster (2) using the indicated pathway reference datasets. All top pathways identified correspond to cell cycle, suggesting the separation observed in the UMAP is mainly attributable to differences in proliferative state. Combined score was obtained via Enrichr and was computed by taking the log of the p-value from the Fisher exact text (one-sided) and multiplying it by the z-score of the deviation from the expected rank.

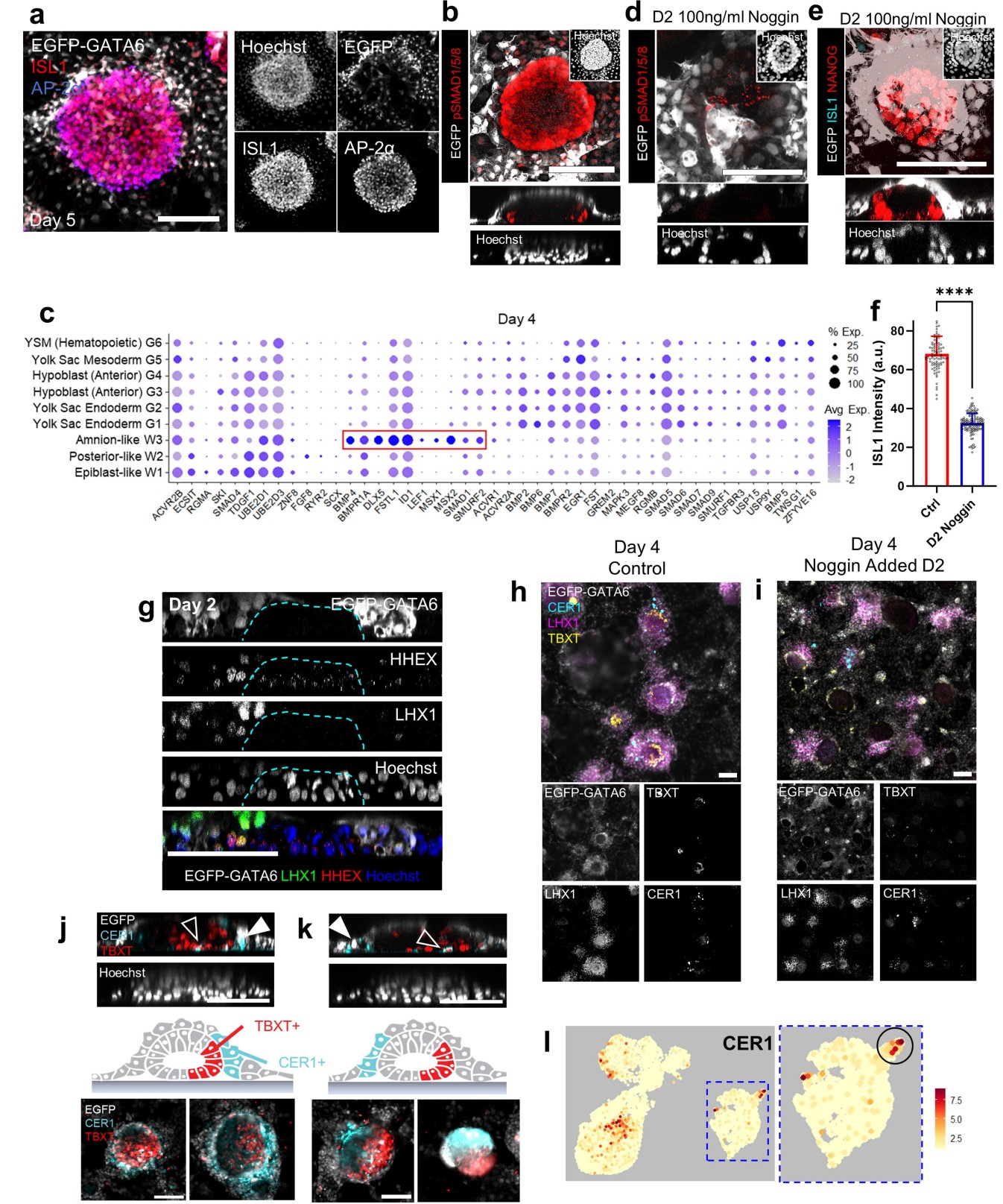

**Extended Data Fig. 6 |** See next page for caption.

**Extended Data Fig. 6 | Amnion and anteroposterior domains.** (**a**) Immunofluorescence staining for the amnion markers ISL1 and AP-2α at day 4. Top-down widefield image of a flattened coverslip. (**b**) Expression patterns of BMP4 effectors (phosphorylated SMAD1, SMAD5, and SMAD8/9) in heX-embryoid. Lower images show a lateral slice of the WT disc shown. (**c**) Dot plot of marker genes from the BMP pathway from day 4 heX-embryoid scRNA-seq. BMP4 expression and a number of associated genes (boxed in red) are highest in the amnion-like population. (**d**) Immunofluorescence staining for the BMP4 effectors (phosphorylated SMAD1, SMAD5, and SMAD8/9) in heX-embryoid at day 4 after application of the inhibitor Noggin at Day 2 of development. Lower images show a lateral slice of the WT disc shown. (**e**) Immunofluorescence staining for ISL1 and NANOG in heX-embryoid at day 4 after application of Noggin at Day 2 of development. Lower images show a lateral slice of the WT disc shown. (**f**) Bar graph showing ISL1 expression (D4) intensities within WT clusters; control (Ctrl) heX-embryoid versus BMP4 inhibition (Noggin) conditions. n[control] = 87, n[Noggin] = 134, **** p = 1.36 × 10$^{-96}$ (C.I. = 95%), calculated via a two-tailed two-sample t-test assuming equal variances. n represents heX-embryoid structures harvested from one round of experiments. Note that a.u. intensities shown will differ from those shown in different experiments, such as those in Fig. 2i, due to differences in staining and imaging parameters at time of sampling. Error bars represent mean ± s.e.m. (**g**) Z-slice of a covered day 2 heX-embryoid structures showing a pole of cells expressing anterior hypoblast markers, matching the polarization of those markers shown in day 4 (see Fig. 3c). (**h**) Control heX-embryoids showing development of TBXT$^+$ posterior-like domains and LHX1$^+$ areas expressing CER1. (**i**) heX-embryoids with 100 ng/mL Noggin added at day 2 showing presence of LHX1$^+$ areas expressing CER1, but no expression of TBXT$^+$ domains. TBXT expression is seen in iGATA6-lineage cells at the periphery of the WT disc. (**j**) A representative WT cluster showing syn-polarity of TBXT and CER1 within the WT cluster and iGATA6 layers, respectively. Filled arrow indicates CER1-expressing cells within the iGATA6 layer; empty arrow indicates CER1 and TBXT co-expressing cells within the WT layer. Z-slices are representative slices from the center of two different WT discs. (**k**) A representative WT cluster showing anti-polarity of TBXT and CER1 within the WT cluster and iGATA6 layers, respectively. Filled arrow indicates CER1-expressing cells within the iGATA6 layer; empty arrow indicates CER1 and TBXT co-expressing cells within the WT layer. Z-slice is a representative slice from the center of a WT disc. (**l**) Merged UMAPs of all D0-D5 scRNA-seq data showing the posterior-like compartment expressing the inhibitor CER1. Dotted boxes show WT lineages. Scale Bars = 100 μm.

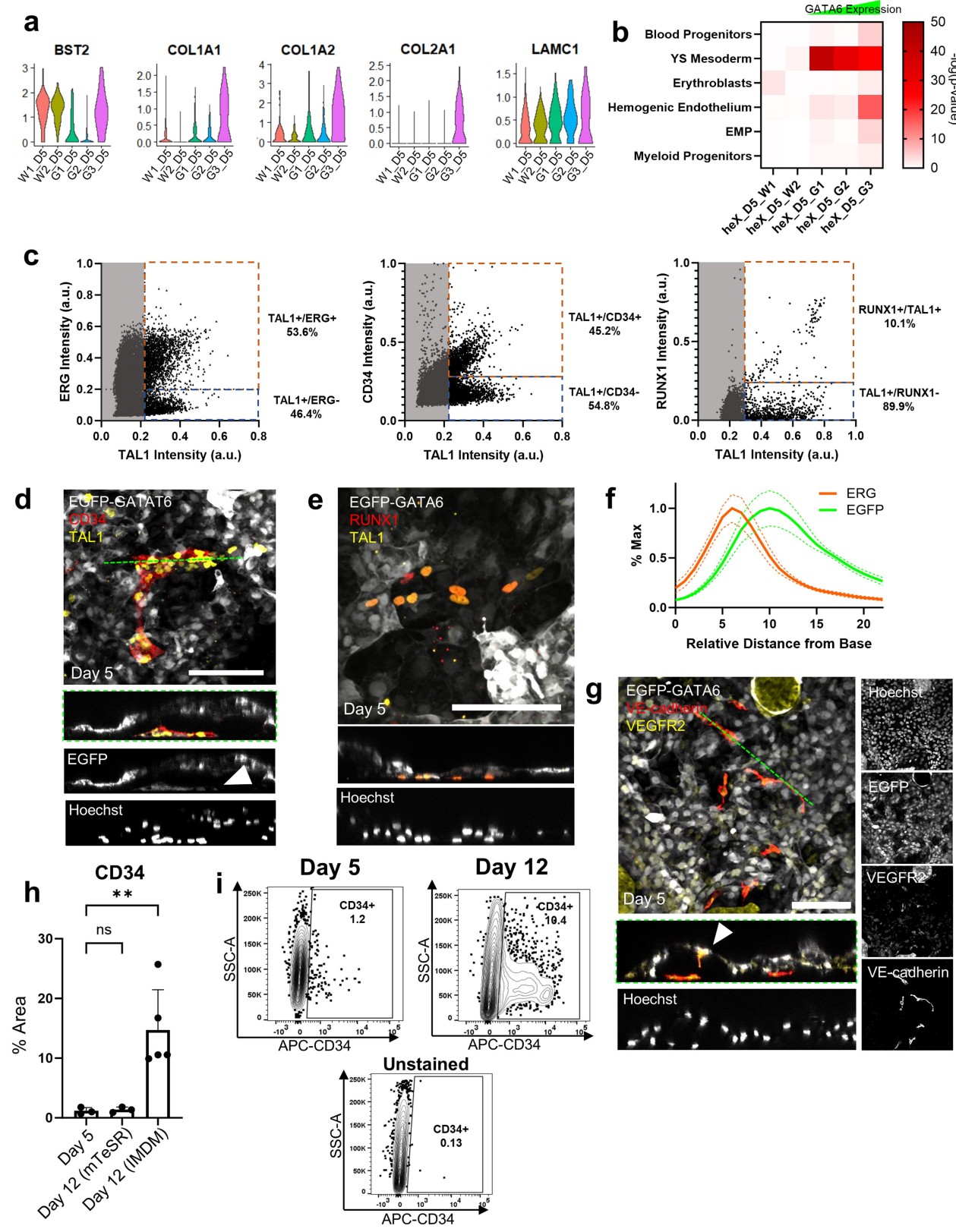

**Extended Data Fig. 7** | See next page for caption.

**Extended Data Fig. 7 | Identification of endothelial/hematopoietic populations in heX-embryoids.** (**a**) Violin plots showing the distribution of ECM genes as well as the yolk sac mesoderm marker gene *BST2* in day 5 scRNA-seq. (**b**) Heatmap showing day 5 scRNA-seq populations compared to hematopoietic populations from the human E16-19 embryo via hypergeometric statistical comparison. This heatmap shows selected results from Extended Data Fig. 4c. We performed multiple test adjustments via Benjamini Hochberg (BH) correction in all comparisons. Tests were performed as one-sided tests. (**c**) Scatterplots showing the distribution of markers obtained via image analysis of 3 independent experiments at day 5. Percentages correspond to the fraction of cells recorded with corresponding marker expression levels above the thresholds represented by the dotted lines. (**d**) IF image showing cells expressing hematopoietic markers in heX-embryoid culture. Cells expressing the hematopoietic marker TAL1 (Scl) localize between the yolk sac endoderm-like compartment and the tissue culture dish and form arrangements of spindle-shaped cells. Orthogonal slice shows the position of spindle cells against the dish. Dashed line indicates the position from which the slice was taken. Arrow indicates the position of these cells in the EGFP channel, demonstrating that they were derived from the iGATA6 population. Scale bar = 100 μm. (**e**) IF image showing RUNX1 and TAL1 expressing cells positioned against the dish. Scale bar = 100 μm. (**f**) Image analysis of z-slices from 5 areas in 3 biological replicates. The peak of ERG expression is underneath the peak of EGFP expression, representing the iGATA6 endoderm-like layer. Dotted curves represent s.e.m. calculated at each point. (**g**) IF image showing cells expressing VE-cadherin and VEGFR2 positioned against the dish. Arrow indicates likely differentiating endothelial cell from the overlaying iGATA6 layer. Scale bar = 100 μm. (**h**) Results from image analysis demonstrating the change in area of CD34$^+$ cells between day 5 and day 12 of the cultures, assessed via analysis of immunofluorescence Images. n = 3 biological replicates for D5 and D12 (mTeSR); n = 5 biological replicates for D12 (IMDM). **: P = 0.0055 (C.I. = 95%) P-value was calculated via a one-way ANOVA, using Dunnett's multiple comparisons test. Error bars represent mean ± s.e.m. (**i**) Representative flow cytometry plot on day 5 and day 12 showing expansion of the CD34$^+$ population by day 12, along with an unstained control (n = 3).

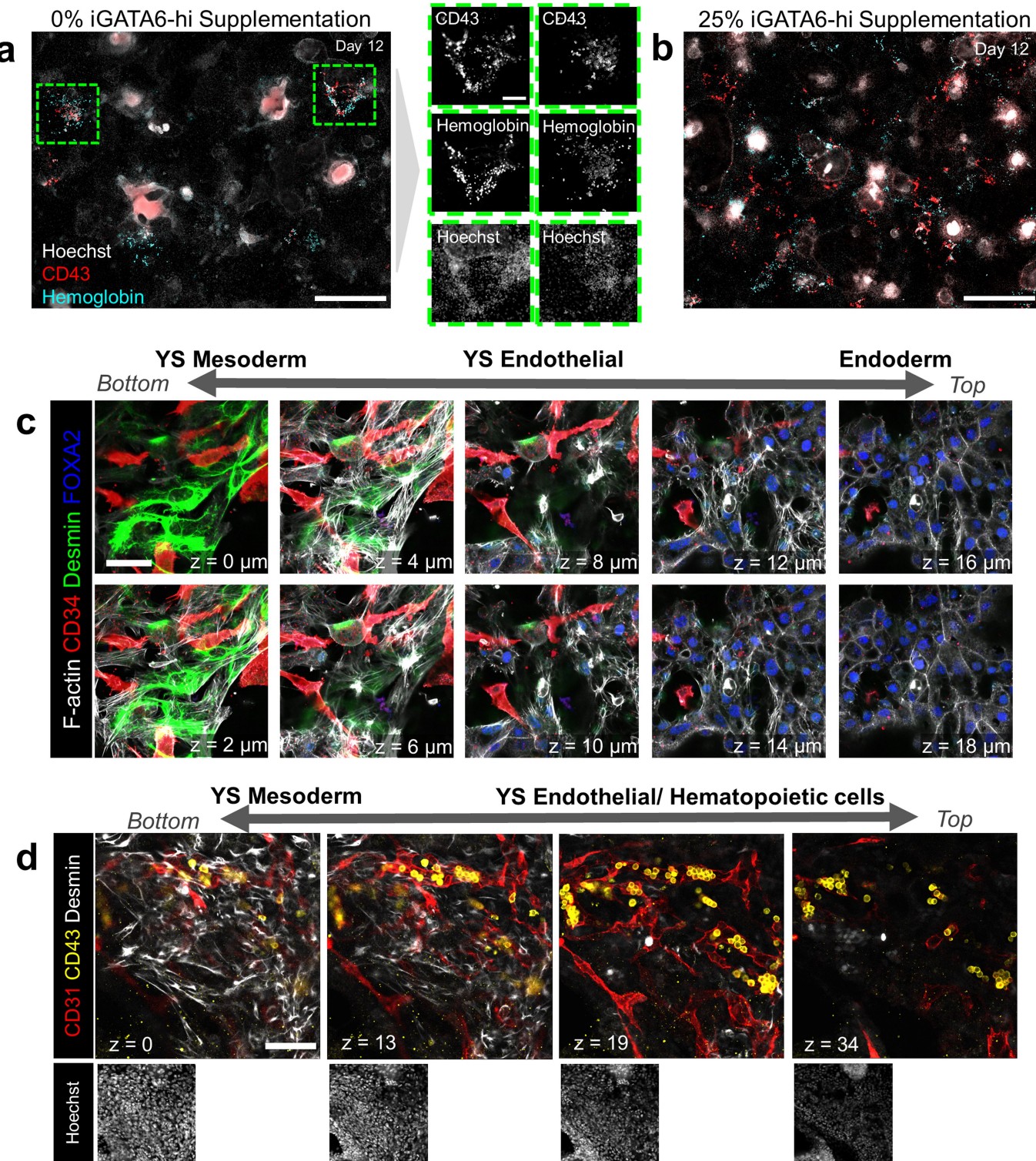

**Extended Data Fig. 8 | GATA6-hi supplementation and structure of hematopoietic foci. (a)** CD43[+] and Hemoglobin[+] cells in day 12 heX-embryoid without additional supplementation of GATA6-hi cells. Red color in high Hoechst areas is nonspecific staining. Scale bar = 500 μm. Insets show areas of CD43[+] cells. Dotted outlines on right show zoomed-in areas shown within the image. Scale bar = 100 μm. **(b)** Expression of CD43[+] and Hemoglobin[+] cells in day 12 heX-embryoid with supplementation of 25% GATA6-hi cells at initial seeding. A notable expansion of CD43[+] and Hemoglobin[+] cells is observable. Scale bar = 500 μm. A representative image of 2–4 biological replicates.

**(c)** Z-slices of a day 12 heX-embryoid at the indicated height above the bottom focal plane including the images shown in Fig. 4h. Expression of the mesodermal marker Desmin is regionalized close to the bottom of the tissue; endothelial marker CD34 is expressed near the bottom and middle of the tissue, with highest expression just above Desmin (z = 6–8); endoderm marker FOXA2 is exclusively expressed above the other tissues. Scale bar = 50 μm. **(d)** Z-slices around the hematopoietic focus shown in Fig. 4e. Desmin is observed regionalized below the endothelial-like (CD31[+]) and hematopoietic-like (CD43[+]) tissues. No stain for endoderm is shown. Scale bar = 100 μm.

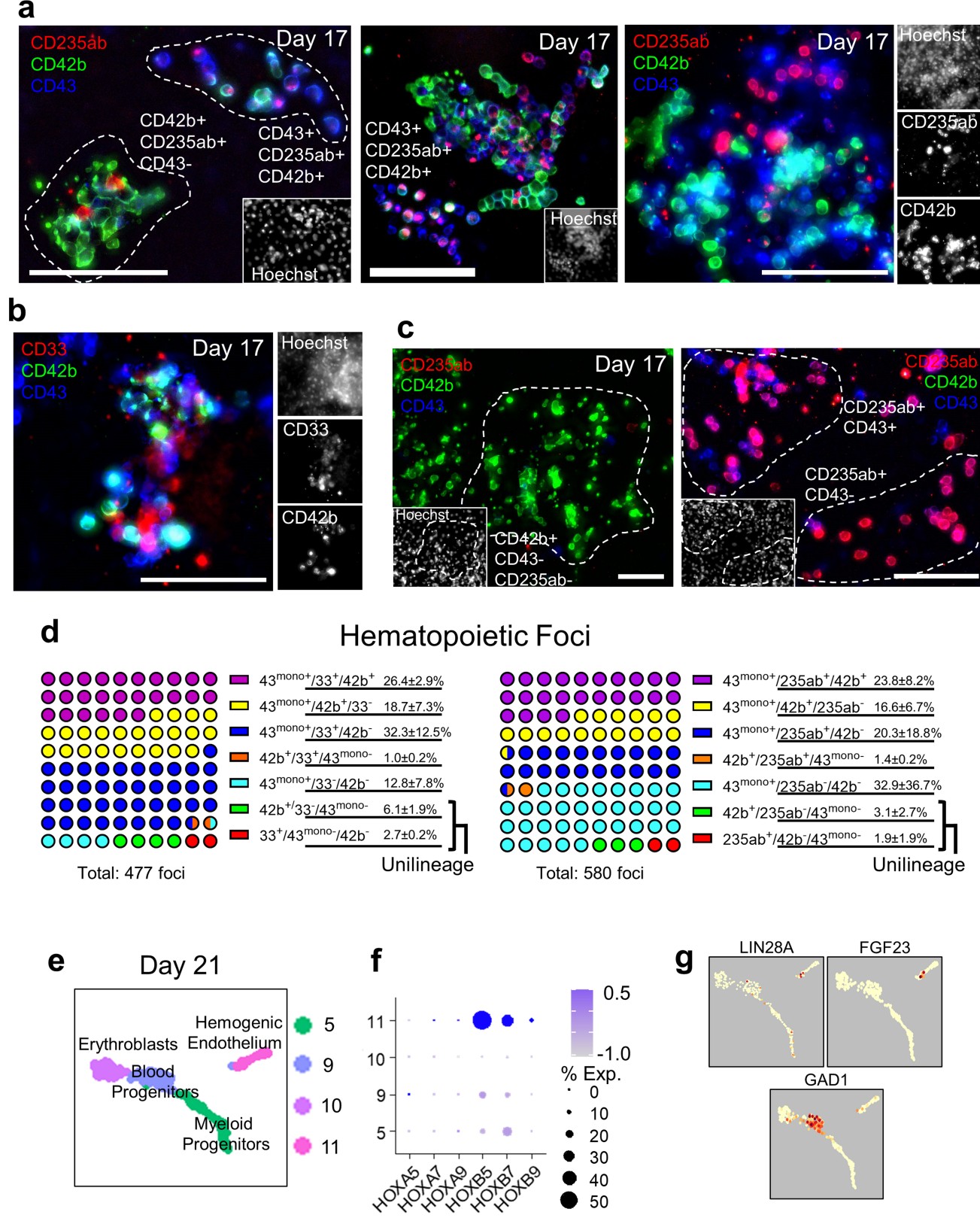

**Extended Data Fig. 9** | See next page for caption.

**Extended Data Fig. 9 | Hematopoietic foci in heX-embryoids.** (**a**) Representative images of multilineage containing erythroid- (CD235ab⁺) and megakaryocyte-like (CD42b⁺) cells from day 17 heX-embryoid Scale bars = 100 µm. (**b**) A representative hematopoietic area containing myeloid- (CD33⁺) and megakaryocyte-like (CD42b⁺) cells from day 17 heX-embryoid. Scale bars = 100 µm. (**c**) Representative images of unilineage hematopoietic foci (left, CD42b; right, CD235ab) at day 17. Scale bars = 100 µm. (**d**) Quantification of the percentages ± SD of hematopoietic areas in two-week heX-embryoids. Left panel stained for CD43, CD33 and CD42b and right panel stained for CD43, CD235ab and CD42b. CD43^mono+ indicates cells that were positive only for CD43 and negative for the other two markers stained in the sample. CD43^mono− indicates that there are no cells positive for CD43 alone, but does not exclude cells copositive for CD43 and another of the markers investigated. Quantification of colonies was performed from at least two independent samples. (**e**) UMAP from Fig. 5d showing scRNA-seq performed on day 21 heX-embryoid with annotations based on hypergeometric statistical comparisons with identities from ref. 23. The full scatterplot has been cropped to show only populations of interest with similarity to in vivo hematopoietic lineages. (**f**) A dot plot showing the expression of a subset of HOX genes within the hematopoietic clusters. HOXA genes are broadly negative, indicating a transcriptional state similar to the yolk sac tissue. (**g**) Expression of *LIN28A*, *FGF23*, and *GAD1* within the hematopoietic clusters. A subset of cells expresses these markers of early hemogenic identity within the yolk sac.

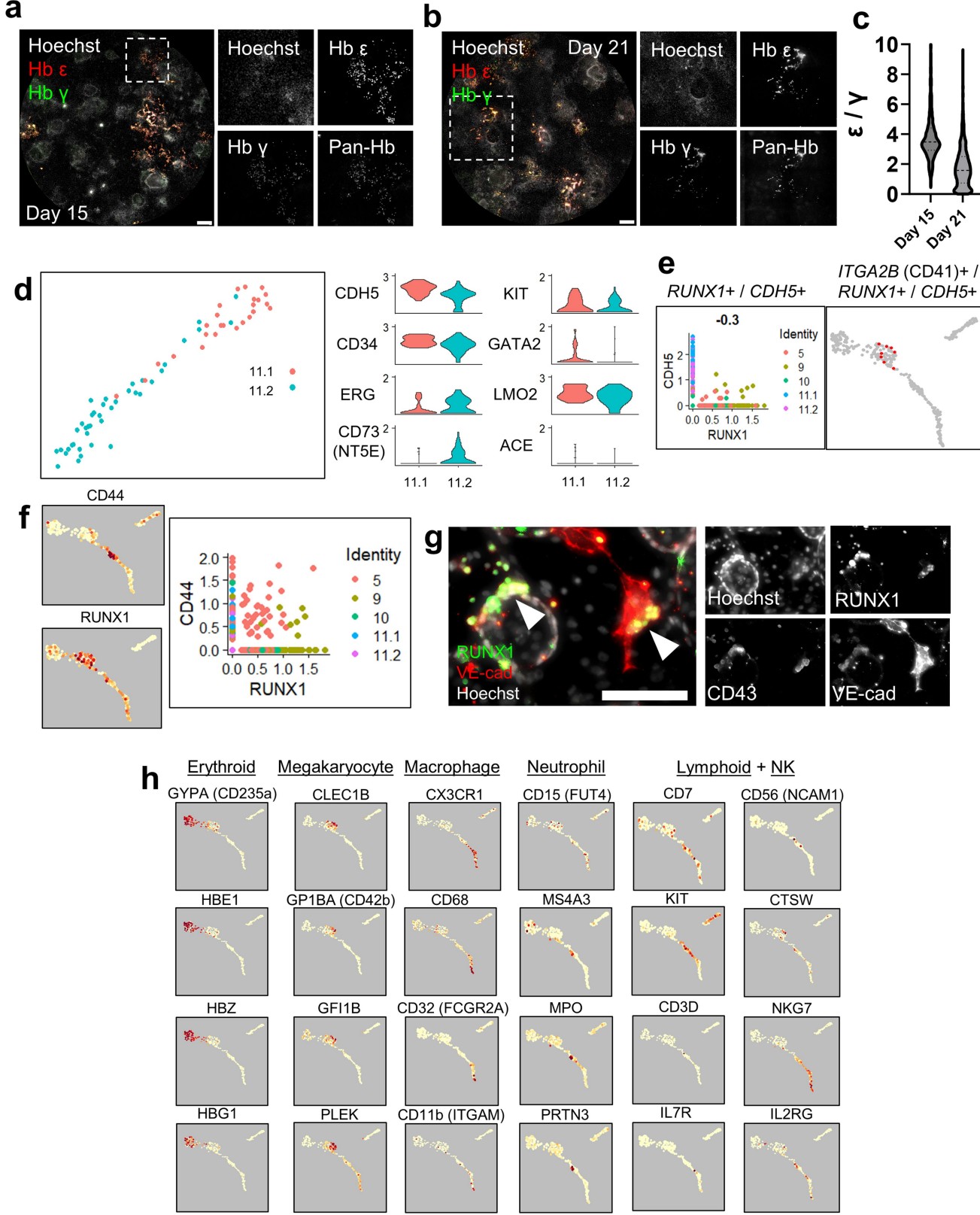

**Extended Data Fig. 10** | See next page for caption.

**Extended Data Fig. 10 | Hematopoietic cell composition in heX-embryoids.**
(**a**) Expression of Hemoglobin ε$^+$ and Hemoglobin γ$^+$ cells in day 15 heX-embryoids. At day 15, areas of Hemoglobin ε-high, Hemoglobin γ-low cells are observed, suggesting more primitive-like hematopoiesis. Scale bar = 500 μm. Dotted box outlines inset shown to the right. A representative image of 3 biological replicates. (**b**) Expression of Hemoglobin ε$^+$ and Hemoglobin γ$^+$ cells in day 21 heX-embryoid. At day 21, areas of Hemoglobin ε-high, Hemoglobin γ-high cells are observed, suggesting more definitive-like hematopoiesis. Scale bar = 500 μm. Dotted box outlines inset shown to the right. A representative image of 3 biological replicates. (**c**) Violin plot showing the distribution of the ratios of Hemoglobin ε to Hemoglobin γ at each day. n[D15] = 3,188 cells, n[D21] = 4088. Counts are from three biological replicates. Central dotted line indicates median, smaller dotted lines above and below indicate quartiles.

(**d**) Subclustering of cluster 11, the putative endothelial population, from the day 21 scRNA-seq data. Hierarchical subclustering identifies two major subpopulations within this cluster, corresponding to hemogenic (11.1) and non-hemogenic (11.2) endothelial subtypes. Marker genes for each population are shown in violin plots to the right. (**e**) Expression of hemogenic endothelial markers within the day 21 hematopoietic populations. Markers co-express mainly in the area annotated as "blood progenitors" in the hypergeometric statistical comparison. (**f**) Scatterplots showing the co-expression of hemogenic endothelial markers within the day 21 hematopoietic populations. (**g**) Immunofluorescence images of day 21 of culture shows co-expression of RUNX1 and VE-cad. Scale bar = 50 μm. (**h**) Scatterplots showing the co-expression of respective lineage markers within the day 21 hematopoietic population.

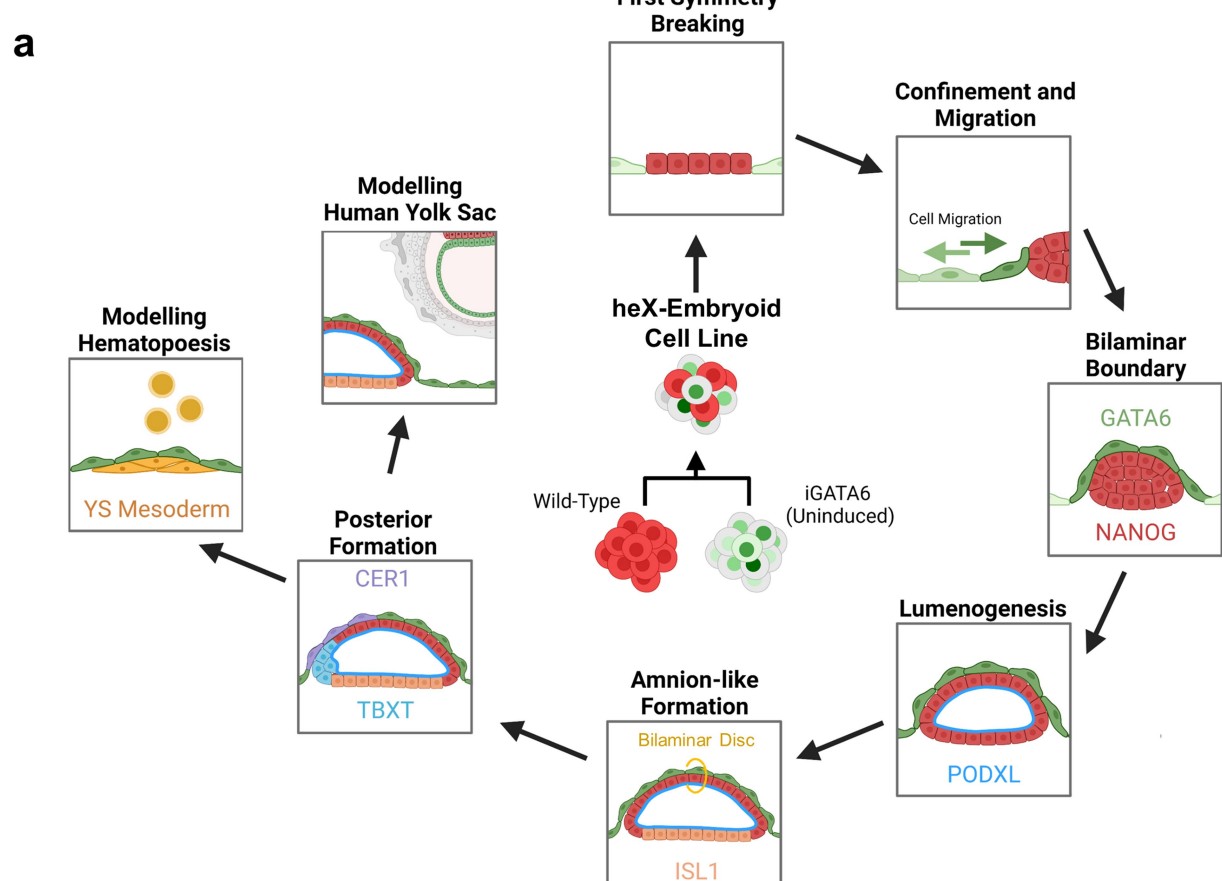

**a**

First Symmetry
Breaking

Confinement and
Migration

Cell Migration

Modelling
Human Yolk Sac

heX-Embryoid
Cell Line

Modelling
Hematopoesis

Wild-Type

iGATA6
(Uninduced)

Bilaminar
Boundary

GATA6

NANOG

YS Mesoderm

Posterior
Formation

CER1

TBXT

Amnion-like
Formation

Bilaminar Disc

ISL1

Lumenogenesis

PODXL

**b**

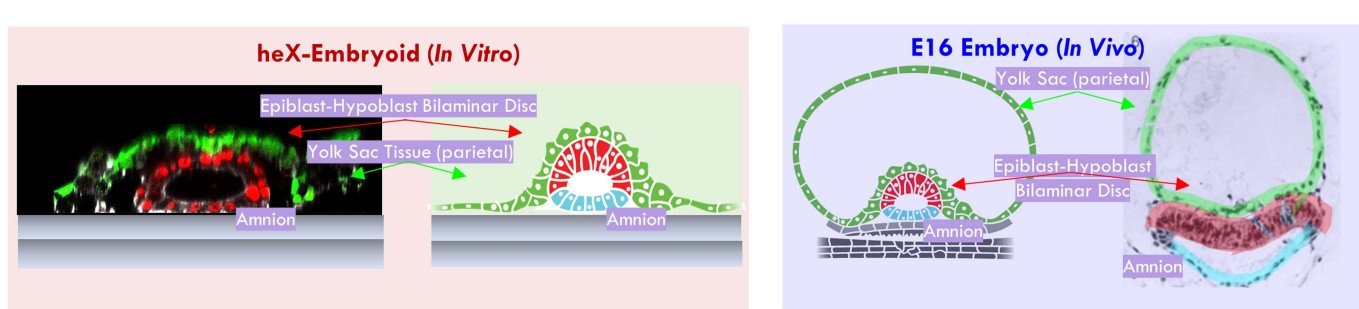

heX-Embryoid (*In Vitro*)

Epiblast-Hypoblast Bilaminar Disc

Yolk Sac Tissue (parietal)

Amnion

Amnion

E16 Embryo (*In Vivo*)

Yolk Sac (parietal)

Epiblast-Hypoblast
Bilaminar Disc

Amnion

Amnion

**Extended Data Fig. 11 | heX-embryoid formation from hiPSCs to model human early post-implantation development in vitro. (a)** From an initially mixed state, heX-embryoid cells segregate into WT clusters surrounded by iGATA6 cells. These iGATA6 cells migrate laterally to create a bilaminar boundary on top of the WT clusters. These clusters then undergo lumenogenesis, specification of amnion-like cells, and formation of anterior hypoblast-like and posterior-like pole. Within heX-embryoids, yolk sac mesoderm-like tissue specifies and hematopoietic cell specification is observed. **(b)** Schematic comparing heX-embryoid morphology in relation to the human embryo. e The fluorescence image is taken from Extended Data Fig. 1i and is stained for OCT4, GFP, and F-actin. Human E16 embryo image is taken from ref. 22. Illustrations in **a** and **b** were created using BioRender (https://biorender.com).

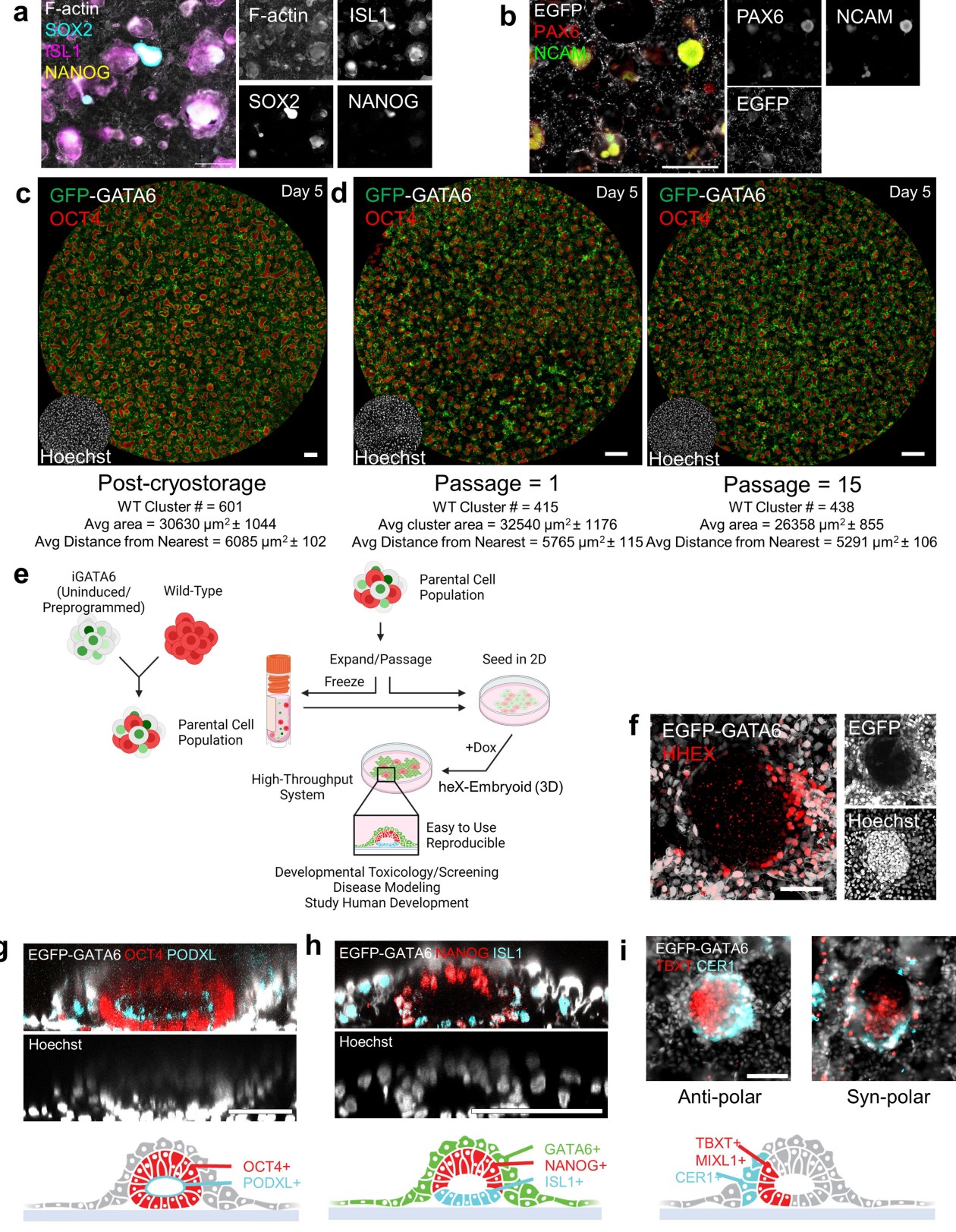

**Extended Data Fig. 12** | See next page for caption.

**Extended Data Fig. 12 | heX-embryoid development, passaging, cryostorage as well as engineering in a separate iPSC line.** (**a**) Most structures corresponding to former WT clusters at day 12 in heX-embryoid have taken on expression of ISL1 and lost expression of the pluripotency markers SOX2 and NANOG. A limited number of cells express SOX2 at the core of former WT clusters, potentially indicating the specification of an ectoderm-like fate in a small number of WT-lineage cells. Scale bar = 500 μm. (**b**) A subset of former WT clusters have taken on markers of ectoderm differentiation. Scale bar = 500 μm. (**c**) heX-embryoid morphology and characteristics following cryostorage and defrosting. Scale Bar = 500 μm. (**d**) Characteristics and morphology of cultures induced immediately after mixing (passage 1) or maintained together and passaged for two months (passage 15). Scale Bar = 500 μm. (**e**) Schematic showing the creation of the heX-embryoid parental cell line. iGATA6 cells with heterogeneous copy numbers of the inducible GATA6 circuit are mixed with wild-type. This cell combination is then maintained together or frozen prior to induction. (**f**) A portion of day 4 PGP9 iGATA6 cells expressing high levels of GATA6 (EGFP) also express the anterior endoderm marker HHEX near the edge of a WT cluster. Scale Bar = 100 μm. (**g**) A ring of PODXL expression lines the inside of a cavities formed in day 4 PGP9 WT clusters. Scale Bar = 100 μm. (**h**) ISL1[+] cells specify away from NANOG[+] cells, along the base of a cavity formed in day 4 PGP9 WT. Scale Bar = 100 μm. (**i**) TBXT[+] cells develop polar domains at the edge of day 4 PGP9 WT clusters. Representative islands show that both syn- and anti-polar arrangement of cells are observable. Scale Bar = 100 μm. n represents heX-embryoid structures from at least three biological replicates. Error shown is ± s.e.m. Illustration in **e** was created using BioRender (https://biorender.com).

# Reporting Summary

## Statistics

For all statistical analyses, confirm that the following items are present in the figure legend, table legend, main text, or Methods section.

| n/a | Confirmed | |
|---|---|---|
| ☐ | ☒ | The exact sample size (*n*) for each experimental group/condition, given as a discrete number and unit of measurement |
| ☐ | ☒ | A statement on whether measurements were taken from distinct samples or whether the same sample was measured repeatedly |
| ☐ | ☒ | The statistical test(s) used AND whether they are one- or two-sided<br>*Only common tests should be described solely by name; describe more complex techniques in the Methods section.* |
| ☒ | ☐ | A description of all covariates tested |
| ☐ | ☒ | A description of any assumptions or corrections, such as tests of normality and adjustment for multiple comparisons |
| ☐ | ☒ | A full description of the statistical parameters including central tendency (e.g. means) or other basic estimates (e.g. regression coefficient) AND variation (e.g. standard deviation) or associated estimates of uncertainty (e.g. confidence intervals) |
| ☐ | ☒ | For null hypothesis testing, the test statistic (e.g. *F*, *t*, *r*) with confidence intervals, effect sizes, degrees of freedom and *P* value noted<br>*Give P values as exact values whenever suitable.* |
| ☒ | ☐ | For Bayesian analysis, information on the choice of priors and Markov chain Monte Carlo settings |
| ☒ | ☐ | For hierarchical and complex designs, identification of the appropriate level for tests and full reporting of outcomes |
| ☒ | ☐ | Estimates of effect sizes (e.g. Cohen's *d*, Pearson's *r*), indicating how they were calculated |

*Our web collection on statistics for biologists contains articles on many of the points above.*

## Software and code

Policy information about availability of computer code

| Data collection | Images were acquired using the EVOS M700 automated scanning microscope (software version 2.0.2094.0), Leica SP8 confocal microscope (Leica Application Suite X version 3.7.4), Incucyte S3 (software version v2019B), or Nikon A1 Confocal microscope (NIS Elements AR) and processed using ImageJ software (version 1.8.0_172). Any contrast adjustments were made in individual channels and applied evenly across the whole image in that channel. Contrast and color balance for color images was applied evenly across the whole image. Information for gene set enrichment analysis was collected using the Enrichr web server (https://maayanlab.cloud/Enrichr/, updated June 8, 2023) |
|---|---|
| Data analysis | Data was analyzed using: Graphpad Prism (v9), ImageJ (version 1.8.0_172), Imaris x64 (version 9.5.0), FlowJo (version 10.7.0), CellRanger (v2.1.0), R (3.6.3), Seurat (4.3.0), SeuratObject (4.1.3), sp (1.6-0), dplyr (1.0.10), devtools (2.3.2), mudata2 (1.1.2), plyr (1.8.8), ggplot2 (3.4.1), cowplot(1.1.1), patchwork (1.1.2), data.table (1.13.6), Matrix (1.5-3), caret (6.0-86), glmGamPoi (1.10.1), sctransform (0.3.5)<br>Wild-type cluster analysis was performed using an in-house Matlab pipeline generated in Matlab version 2020a. Hemoglobin analysis was performed using a pipeline developed in CellProfiler (version 4.2.5).<br>Scripts can be found at https://github.com/AmirAlavi/GATA6_R |

For manuscripts utilizing custom algorithms or software that are central to the research but not yet described in published literature, software must be made available to editors and reviewers. We strongly encourage code deposition in a community repository (e.g. GitHub). See the Nature Portfolio guidelines for submitting code & software for further information.

## Data

Policy information about availability of data

All manuscripts must include a data availability statement. This statement should provide the following information, where applicable:

- Accession codes, unique identifiers, or web links for publicly available datasets
- A description of any restrictions on data availability
- For clinical datasets or third party data, please ensure that the statement adheres to our policy

The sequencing reads, and single-cell expression matrices for all the single-cell RNA-seq data are submitted to NIH BioProject accession number PRJNA1035788. For hypergeometric analysis, we used publicly available data from Tyser et al. processed peri-implantation dataset E-MTAB-3929, Xiang et al. GSE136447 and Ma et al. GSE130114.
The datasets generated during and/or analyzed during the current study are available from the corresponding author on reasonable request. All scripts used have been deposited at https://github.com/AmirAlavi/GATA6_R

## Research involving human participants, their data, or biological material

Policy information about studies with human participants or human data. See also policy information about sex, gender (identity/presentation), and sexual orientation and race, ethnicity and racism.

| | |
|---|---|
| Reporting on sex and gender | No reportings or findings related to sex or gender were made or presented. |
| Reporting on race, ethnicity, or other socially relevant groupings | No reportings or findings related to race, ethnicity, or other groupings were made or presented. |
| Population characteristics | No human participants were used in this data. |
| Recruitment | No participants were recruited for this data. |
| Ethics oversight | The University of Pittsburgh Stem Cell Research Oversight board approved the study. |

Note that full information on the approval of the study protocol must also be provided in the manuscript.

# Field-specific reporting

Please select the one below that is the best fit for your research. If you are not sure, read the appropriate sections before making your selection.

☒ Life sciences          ☐ Behavioural & social sciences          ☐ Ecological, evolutionary & environmental sciences

For a reference copy of the document with all sections, see nature.com/documents/nr-reporting-summary-flat.pdf

# Life sciences study design

All studies must disclose on these points even when the disclosure is negative.

| | |
|---|---|
| Sample size | All experiments were conducted with multiple biological replicates, with the exact number described in the text. We did not predetermine sample sizes via statistical methods. The number of samples used in each experiment was selected to ensure data consistency/reproducibility and was based on the available resources. Sample size for single cell RNA-Seq was determined based on our estimation of the number of cells required to capture cell types of interest detected via immunofluorescence at the developmental stage. |
| Data exclusions | No samples were excluded. For computational analysis of samples, a crop of the central 9000 pixels, excluding the edges of the sample, was used. |
| Replication | The exact numbers of replicates for methods/analyses used are indicated in the respective figure legends. All replications were successful. Each individual experimental staining was repeated in at least two dishes and/or coverslips. Additionally, a subset of experiments were validated in two cell lines (PGP1 and PGP9), and by multiple investigators involved in this work. |
| Randomization | Embryoid samples developed in each culture well for each experiment were allocated to control and experimental groups in a random manner at seeding when different conditions (i.e. pathway inhibition) were present. |
| Blinding | For hypergeormetric analysis the person who did the analysis was not aware of the groups and identity of the samples and grouping. For small molecule treatment the identity of samples are sometimes obvious due to the morphological changes, hence blinding is not practical. Media changes cannot be performed with blinding as each group should receive the appropriate media. In all other experiments we had no relevant scientific reasons to conduct blinding as the experiments were mostly descriptive. |

# Reporting for specific materials, systems and methods

We require information from authors about some types of materials, experimental systems and methods used in many studies. Here, indicate whether each material, system or method listed is relevant to your study. If you are not sure if a list item applies to your research, read the appropriate section before selecting a response.

## Materials & experimental systems

| n/a | Involved in the study |
|---|---|
| ☐ | ☒ Antibodies |
| ☐ | ☒ Eukaryotic cell lines |
| ☒ | ☐ Palaeontology and archaeology |
| ☒ | ☐ Animals and other organisms |
| ☒ | ☐ Clinical data |
| ☒ | ☐ Dual use research of concern |
| ☒ | ☐ Plants |

## Methods

| n/a | Involved in the study |
|---|---|
| ☒ | ☐ ChIP-seq |
| ☐ | ☒ Flow cytometry |
| ☒ | ☐ MRI-based neuroimaging |

## Antibodies

| Antibodies used | The following antibodies were used in this study. All primary antibodies for immunofluorescence were used at a dilution of 1:200. All secondary antibodies were used at a dilution of 1:400. All FACS antibodies were used at a dilution of 1:400. The specific clone for monoclonal antibodies is given in parentheses when applicable. |
|---|---|

GFP Abcam (B-2) ab13970
GFP Aves Labs (Polyclonal) GFP-1020
NANOG R&D (Polyclonal) AF1997
PDGFRα Cell Signaling (D13C6) 5241S
OCT4 Cell Signaling (Polyclonal) mab2750
CD34 abcam (EP373Y) ab81289
CD43 R&D (290111) MAB2038
CD31 abcam (JC/70A) AB9498-1001
CD41 abcam (EPR4330) AB134131-1001
CD33 R&D (996810) MAB11371
CD42b R&D (Polyclonal) AF4067
CD235ab Biolegend (HIR2) 306602
TAL1 Santa Cruz (Polyclonal) sc12984
ERG abcam (EPR3864) ab92513
RUNX1 Santa Cruz (A-2) sc-365644
VEGFR2 R&D (89115) MAB3571
Hemoglobin R&D (Polyclonal) G-134-C
CX3CR1 abcam (Polyclonal) AB167571-1001
Hoechst 33342 Thermo Fisher H3570
Phalloidin-iFluor 405 AB176752-1001
PODXL Invitrogen (AB_2532205) 433140
ZO-1 Invitrogen (Polyclonal) 617300
LAMA1 Sigma-Aldritch (Polyclonal) SAB4501255
ISL1 Abcam (EP4182) ab109517
Phospho-SMAD1/5/8 Cell Signaling (D5B10) 13820T
Phospho-SMAD2 Cell Signaling (D27F4) 8828S
MIXL1 Invitrogen (Polyclonal) PA564903
TBXT R&D (Polyclonal) AF2085
CER1 abcam (Polyclonal) ab184133
VE-cad R&D (Polyclonal) AF938
Desmin R&D (Polyclonal) AF3844
FOXA2 Santa Cruz (H-8) sc-271104
PAX6 abcam (AD2.38) ab78545
NCAM abcam (EP2567Y) ab75813
HHEX R&D (2018B) MAB83771
GATA6 R&D (Polyclonal) AF1700
AP-2α Invitrogen (3B5) MA1-872

FACS Analysis:
CD34-APC Biolegend (581) 343510
CD31-PE/Cy7 Biolegend (WM59) 303118
CD42b-AF700 Biolegend (HIP1) 303928
CD33-BV605 Biolegend (P67.6) 366612
CD45-APC/Cy7 Biolegend (2D1) 368516
CD45-Pacific Blue Biolegend (2D1) 368539
CD45-APC Biolegend (HI30) 304012

CD43-PE Biolegend (CD43-10G7) 343203
CD235ab-PE/Cy7 (HIR2) Biolegend 306620
CD7-PE/Cy7 Biolegend (4H9/CD7) 395609
CD15-PE Biolegend (HI98) 301905
CD49d(VLA-4)-BV605 (9F10) Biolegend 304313
CD117-BV421 BD Biosciences (104D2) 563856
CD56-PE/Cy7 Biolegend (MEM-188) 304628

Secondary Antibodies for immunofluorescence:
Alexa Fluor® 594 AffiniPure Donkey Anti-Rabbit IgG (H+L) Jackson Immunoresearch 711-585-152
Alexa Fluor® 647 AffiniPure Donkey Anti-Goat IgG (H+L) Jackson Immunoresearch 705-605-147
Alexa Fluor® 647 AffiniPure Donkey Anti-Mouse IgG (H+L) Jackson Immunoresearch 715-605-151
Alexa Fluor® 647 AffiniPure Donkey Anti-Sheep IgG (H+L) Jackson Immunoresearch 713-605-147
Alexa Fluor® 488 AffiniPure Donkey Anti-Chicken IgY (IgG) (H+L) Jackson Immunoresearch 703-545-155

| | |
|---|---|
| Validation | All antibodies were obtained from commercial sources. Detailed validation statements are available on the corresponding manufacturers' websites, as provided below: |

GFP Abcam (B-2) ab13970, referenced in 3182 publications: https://www.abcam.com/products/primary-antibodies/gfp-antibody-ab13970.html

GFP Aves Labs (Polyclonal) GFP-1020, referenced in 1782 publications: https://www.aveslabs.com/products/anti-green-fluorescent-protein-antibody-gfp

NANOG R&D (Polyclonal) AF1997, referenced in 207 publications: https://www.rndsystems.com/products/human-nanog-antibody_af1997

PDGFRα Cell Signaling (D13C6) 5241S, referenced in 60 publications: https://www.cellsignal.com/products/primary-antibodies/pdgf-receptor-a-d13c6-xp-rabbit-mab/5241

OCT4 Cell Signaling (Polyclonal) mab2750, cited in 312 publications: https://www.cellsignal.com/products/primary-antibodies/oct-4-antibody/2750

CD34 abcam (EP373Y) ab81289, referenced in 499 publications: https://www.abcam.com/products/primary-antibodies/cd34-antibody-ep373y-ab81289.html

CD43 R&D (290111) MAB2038, referenced in 1 publication: https://www.rndsystems.com/products/human-cd43-antibody-290111_mab2038

CD31 abcam (JC/70A) AB9498-1001, referenced in 210 publications: https://www.abcam.com/products/primary-antibodies/cd31-antibody-jc70a-ab9498.html

CD41 abcam (EPR4330) AB134131-1001, referenced in 20 publications: https://www.abcam.com/products/primary-antibodies/cd41-antibody-epr4330-ab134131.html

CD33 R&D (996810) MAB11371, https://www.rndsystems.com/products/human-siglec-3-cd33-antibody-996810_mab11371

CD42b R&D (Polyclonal) AF4067, referenced in 2 publications: https://www.rndsystems.com/products/human-cd42b-gpib-alpha-antibody_af4067

CD235ab Biolegend (HIR2) 306602, referenced in 7 publications: https://www.biolegend.com/en-us/products/purified-anti-human-cd235ab-antibody-743

TAL1 Santa Cruz (Polyclonal) sc12984, referenced in 12 publications: https://www.scbt.com/p/tal1-antibody-c-21

ERG abcam (EPR3864) ab92513, referenced in 201 publications: https://www.abcam.com/products/primary-antibodies/erg-antibody-epr3864-ab92513.html

RUNX1 Santa Cruz (A-2) sc-365644, referenced in 47 publications: https://www.scbt.com/p/runx1-antibody-a-2

VEGFR2 R&D (89115) MAB3571, referenced in 17 publications: https://www.rndsystems.com/products/human-vegfr2-kdr-flk-1-antibody-89115_mab3571

Hemoglobin R&D (Polyclonal) G-134-C, referenced in 2 publications: https://www.rndsystems.com/products/human-hemoglobin-antibody_g-134-c

CX3CR1 abcam (Polyclonal) AB167571-1001: https://www.abcam.com/products/primary-antibodies/cx3cr1-antibody-ab167571.html

Hoechst 33342 Thermo Fisher H3570: https://www.thermofisher.com/order/catalog/product/H3570?SID=srch-srp-H3570

Phalloidin-iFluor 405 AB176752-1001, referenced in 14 publications: https://www.abcam.com/products/chip-kits/phalloidin-ifluor-405-reagent-ab176752.html

PODXL Invitrogen (AB_2532205) 433140: https://www.thermofisher.com/antibody/product/PODXL-Antibody-Monoclonal/433140

ZO-1 Invitrogen (Polyclonal) 617300, referenced in 765 publications: https://www.thermofisher.com/antibody/product/ZO-1-

Antibody-Polyclonal/61-7300

LAMA1 Sigma-Aldritch (Polyclonal) SAB4501255: https://www.sigmaaldrich.com/US/en/product/sigma/sab4501255

ISL1 Abcam (EP4182) ab109517, referenced in 52 publications: https://www.abcam.com/products/primary-antibodies/islet-1-antibody-ep4182-neural-stem-cell-marker-ab109517.html

Phospho-SMAD1/5/8 Cell Signaling (D5B10) 13820T, referenced in 412 publications: https://www.cellsignal.com/products/primary-antibodies/phospho-smad1-ser463-465-smad5-ser463-465-smad9-ser465-467-d5b10-rabbit-mab/13820

Phospho-SMAD2 Cell Signaling (D27F4) 8828S, referenced in 676 publications: https://www.cellsignal.com/products/primary-antibodies/phospho-smad2-ser465-467-smad3-ser423-425-d27f4-rabbit-mab/8828

MIXL1 Invitrogen (Polyclonal) PA564903, referenced in 2 publications: https://www.thermofisher.com/antibody/product/MIXL1-Antibody-Polyclonal/PA5-64903

TBXT R&D (Polyclonal) AF2085, referenced in 145 publications: https://www.rndsystems.com/products/human-mouse-brachyury-antibody_af2085

CER1 abcam (Polyclonal) ab184133: https://www.abcam.com/products/primary-antibodies/cer1-antibody-ab184133.html

VE-cad R&D (Polyclonal) AF938, referenced in 17 publications: https://www.rndsystems.com/products/human-ve-cadherin-antibody_af938

Desmin R&D (Polyclonal) AF3844, referenced in 16 publications: https://www.rndsystems.com/products/human-mouse-desmin-antibody_af3844

FOXA2 Santa Cruz (H-8) sc-271104, referenced in 1 publication: https://www.scbt.com/p/hnf-3beta-antibody-h-8?requestFrom=search

PAX6 abcam (AD2.38) ab78545, referenced in 44 publications: https://www.abcam.com/products/primary-antibodies/pax6-antibody-ad238-ab78545.html

NCAM abcam (EP2567Y) ab75813 , referenced in 35 publications: https://www.abcam.com/products/primary-antibodies/ncam1-antibody-ep2567y-ab75813.html

HHEX R&D (2018B) MAB83771, referenced in 6 publications: https://www.rndsystems.com/products/human-mouse-rat-hhex-antibody-2018b_mab83771

GATA6 R&D (Polyclonal) AF1700, referenced in 61 publications: https://www.rndsystems.com/products/human-gata-6-antibody_af1700

AP-2α Invitrogen (3B5) MA1-872, referenced in 3 publications: https://www.thermofisher.com/antibody/product/AP2-alpha-Antibody-clone-3B5-Monoclonal/MA1-872

CD34-APC Biolegend (581) 343510, referenced in 18 publications: https://www.biolegend.com/en-us/products/apc-anti-human-cd34-antibody-6090

CD31-PE/Cy7 Biolegend (WM59) 303118, referenced in 18 publications: https://www.biolegend.com/en-us/products/pe-cyanine7-anti-human-cd31-antibody-6124

CD42b-AF700 Biolegend (HIP1) 303928: https://www.biolegend.com/en-us/products/alexa-fluor-700-anti-human-cd42b-antibody-14765

CD33-BV605 Biolegend (P67.6) 366612, referenced in 1 publication: https://www.biolegend.com/en-us/products/brilliant-violet-605-anti-human-cd33-antibody-12255

CD45-APC/Cy7 Biolegend (2D1) 368516, referenced in 30 publications: https://www.biolegend.com/en-us/products/apc-cyanine7-anti-human-cd45-antibody-12400

CD45-Pacific Blue Biolegend (2D1) 368539, referenced in 11 publications: https://www.biolegend.com/en-us/products/pacific-blue-anti-human-cd45-antibody-14908

CD45-APC Biolegend (HI30) 304012, referenced in 74 publications: https://www.biolegend.com/en-us/products/apc-anti-human-cd45-antibody-705

CD43-PE Biolegend (CD43-10G7) 343203, referenced in 4 publications: https://www.biolegend.com/en-us/products/pe-anti-human-cd43-antibody-6011

CD235ab-PE/Cy7 (HIR2) Biolegend 306620, referenced in 7 publications: https://www.biolegend.com/en-us/products/purified-anti-human-cd235ab-antibody-743

CD7-PE/Cy7 Biolegend (4H9/CD7) 395609: https://www.biolegend.com/en-us/products/pe-cyanine7-anti-human-cd7-antibody-21684

CD15-PE Biolegend (HI98) 301905, referenced in 27 publications: https://www.biolegend.com/en-us/products/pe-anti-human-cd15-

ssea-1-antibody-713

CD49d(VLA-4)-BV605 (9F10) Biolegend 304313, referenced in 16 publications: https://www.biolegend.com/en-us/products/pe-cyanine7-anti-human-cd49d-antibody-6776

CD117-BV421 BD Biosciences (104D2) 563856: https://www.bdbiosciences.com/en-us/products/reagents/flow-cytometry-reagents/research-reagents/single-color-antibodies-ruo/bv421-mouse-anti-human-cd117.563856

CD56-PE/Cy7 Biolegend (MEM-188) 304628, referenced in 7 publications: https://www.biolegend.com/en-us/products/pe-cyanine7-anti-human-cd56-ncam-antibody-4584

## Eukaryotic cell lines

Policy information about cell lines and Sex and Gender in Research

| Cell line source(s) | PGP1 and PGP9 parental cell lines were supplied through an MTA from the Weiss lab at MIT, and can be obtained from Coriell. |
|---|---|
| Authentication | PGP1 and PGP9 hiPSC lines were authenticated in-house via immunostaining for pluripotency markers (OCT4, NANOG, SOX2) and successful differentiation to the three germ layers. Engineered cell lines were assessed for functionality via detection of expression of GATA6 and EGFP. |
| Mycoplasma contamination | All cell lines tested negative for mycoplasma contamination. |
| Commonly misidentified lines (See ICLAC register) | No commonly misidentified cell lines listed by ICLAC were used in this work. |

## Flow Cytometry

### Plots

Confirm that:

☒ The axis labels state the marker and fluorochrome used (e.g. CD4-FITC).

☒ The axis scales are clearly visible. Include numbers along axes only for bottom left plot of group (a 'group' is an analysis of identical markers).

☒ All plots are contour plots with outliers or pseudocolor plots.

☒ A numerical value for number of cells or percentage (with statistics) is provided.

### Methodology

| Sample preparation | Cultures were treated for 45 minutes with Collagenase C solution (3 mg/ml Stem Cell Technologies) followed by 15 minutes treatment with Accuatse (Sigma) and filtered through 40μm filters (Thermo Fisher) to make the single cell suspension. FC block solution (Thermo Fischer) was added to the samples followed by 10 minutes incubation on ice. Next, the antibody mix (1:400) was added to the samples followed by 30 minutes incubation on ice. Cells were analyzed using an LSR II flow cytometer (BD Bioscience) using 7-AAD (BD Pharmingen) for dead cell staining. |
|---|---|
| Instrument | Cells were analyzed using an LSR II or Fortessa flow cytometer (BD Biosciences) |
| Software | FACSDiva (BD Biosciences) software suite was used for collection. FlowJo software (version 10.7.0) was used for flow cytometry analysis. |
| Cell population abundance | Abundance of distinct cell populations of interest was determined using appropriate negative controls. |
| Gating strategy | Standard gating settings were used. Cell debris was excluded via an SSC-A vs FSC-A gate; aggregates were excluded by comparing FSC-A and FSC-H; dead cells were gated out using the 7-AAD stain to identify positive cells. The final gating is available in Supplementary Information 1. |

☒ Tick this box to confirm that a figure exemplifying the gating strategy is provided in the Supplementary Information.

