## [Peer Review File · Nature]

Manuscript Title: Modeling Post-Implantation Human Development to Yolk Sac Blood Emergence

Reviewer Comments & Author Rebuttals

Reviewer Reports on the Initial Version:

Referees' comments:

Referee #1 (Remarks to the Author):

Hislop et al.

In this study Hislop et al. report a new methodology for modeling early human development that involves recapitulating interactions between pluripotent cells and extraembryonic cells via an engineering approach. Cells overexpressing GATA6 are combined with human pluripotent stem cells, and grown in standard 2D cultures. The authors report the formation of cell populations with gene expression consistent with extraembryonic endoderm and mesoderm, early hematopoiesis, amnion and primitive streak. In some instances there is evidence that the cell types are in appropriate anatomical configurations.

Although the authors present some intriguing data, in a number of instances the temporal sequence of events, and the cell identifications, are not fully supportive of the claim that the model represents normal human development. Many of the phenomena described have already been reported in conventional human pluripotent stem cell culture, and the study fails to show how exactly the engineered GATA6+ cells contribute to the subsequent differentiation and morphogenetic events. The most interesting claim is that of formation of blood islands, but the evidence to support this interpretation seems rather thin. The authors need to provide more information on the reproducibility of their observations, in particular, how often structures with the arrangements of cells and tissue depicted were formed.

Specific comments

1. To refer to the iDiscoid as a synthetic biology construct is a bit of a stretch. Overexpression of master regulatory transcription factors, as done here with GATA6, has been described in the field for years and is not considered by most as synthetic biology. The relevant gene circuits are not really regulated in any sense, nor are they particularly well described.
2. Ext data Figure 1-were expression levels of GATA6 regulated and if so, how?
3. Page 2 L 80-if the authors begin with conventional hPSC, which correspond to post-implantation cells, how is it that the model initially mimics the D6 inner cell mass.
4. Page 2 L 80-were the GATA6 cells characterized further? Data in Ext Data 2 is not sufficient to indicate that these cells are primitive endoderm
5. What is an appropriate control for experiments in Figure 1 and elsewhere? Cells expressing many of these genes could be found in conventional embryoid bodies, or overgrown adherent cultures. In fact previous studies have described appearance of extraembryonic endoderm like cells around the

periphery of stem cell colonies grown under standard conditions (e.g. DOI: 10.1016/j.cell.2019.03.013 and others).

6. P3 L86 figure 1C does not show segregation of the two cell types. Supplemental 2c is confusing in this respect. The GATA6 only cells could just be at any early stage of extraembryonic specification. Since they are in close contact with pluripotent cells their identity is important.

7. P3 L 97-why is the comparison with E16-19 embryo appropriate here? The timing is confusing here, since E16-19 is well into gastrulation, and the authors' model should be at an early post-implantation stage.

8. Page 3 L104-it is not clear that Cluster 0 contains PS-like cells, and it would be helpful to know what stage of epiblast it corresponds to.

9. P4 L 142-AFP is not a good marker for early erythropoiesis, and neither is APOA1. Both are expressed in yolk sac endoderm

10. P4 L145-better characterization of "endothelial" cells is required.

11. P4 L 147-what proportion of cells were double positive for RUNX and TAL1 as in C and how often were they found in the indicated arrangement? The introduction points to inconsistency in previously published human embryo models, a valid point, but throughout this study there is no quantitative indication of how reproducible the structures shown are.

12. Figure 2D-CD71 cells require further characterization. Human yolk sac hematopoiesis has been extensively studied in vitro, much more convincing evidence is required here to support the conclusions that blood islands have formed. Percentages of cells expressing specific markers are very low (Ext data Figure 5).

13. P5 L 171-cavitation and amniotic cavity like morphogenesis have been described by others. The structures appear upside down in Figure 3 e-g. What additional characterization of amnion was carried out? How does the timing here relate to an E16-19 human embryo?

14. Page 5 L185-is trophectoderm absent from the model?

15. Figure 3-again at what frequency were structures like those in E-G observed? How common were the structures with multiple cavities shown in C observed and what are they?

16. Extended data figure 9-is BMP active in this system in the pg/ml range?

17. Page 6 L229-others have shown this distribution of SMAD signaling in conventional hPSC cultures

18. P7 L245-the role of Nodal signaling in amnion induction is not clear from these experiments. Where are data on the other components of the receptor system? The inhibitor used is not specific for Nodal.

19. Figure 4-these figures show segregation of T+ cells, but do not convincingly demonstrate formation of a primitive streak.

20. Page 7 L265 onwards-it is not clear what the authors are trying to say regarding CER1. Cer1 is a BMP and WNT antagonist expressed in the anterior domain of the visceral endoderm. Why it should be expressed near epiblast destined to give rise to T+ cells in the authors' model is unclear. There is no direct evidence presented to show that CER1 acts as a Nodal antagonist in this system.

Referee #2 (Remarks to the Author):

Modelling Human Post-Implantation Development via Extra-Embryonic Niche Engineering

The iDiscoïd model consists of a mixed population of wildtype and GATA6 inducible human ESCs. This simple approach resulted in a surprising degree of complexity and led to organized structures seemingly comprising major cell types of the human periimplantation embryo, including putative amnion-like, primitive streak-like, pluripotent cells, visceral endoderm-like cells, and yolk sac cells. Overall, the paper is well presented with clear schematics.

Despite interesting observations being reported, the validity of such a model to recapitulate genuine events of human development is questionable. The “model” consists of a carpet of millions of cells, which is hardly corresponding to the small periimplantation in the womb. Although such a 2D culture system has practical conveniences, it will show many artefacts – which can be readily seen in the data. Most strikingly, the anterior visceral endoderm (CER1 positive regions) arise adjacent to gastrulating (TBXT and MIXL1 positive regions), which is at odds with chicken, mouse, rabbit, pig, and cynomolgus development. Equally the authors do not investigate alternative AVE markers, e.g. LHX1, LEFTY1, apart from HHEX. Another major issue is that in iDiscoïds epiblast lumenogenesis occurs via multiple lumen, which has not been overserved in either rhesus or human preimplantation embryos of the Carnegie collection. The role of CXCR4 in migration might be interesting, but it is unclear how this relates to the embryo in vivo.

The model may provide a useful platform to dissect the cross-talk between embryonic and extraembryonic lineages. It would be great if it were better defined in terms of developmental time and signalling requirements. The sorting behaviour of iGATA6 and WT cells would be expected to happen in preimplantation embryos – note that this happens before and not during or after implantation (Kuijk, Development, 2012). – so why has this not been done with naïve hESCs rather than primed? How does mTESR affect the sorting and what are the signalling requirements for this behaviour? How do seeding ratios affect results?

Different iGATA6 level seem to induce a variety of embryo-related lineages. This is potentially interesting, but the resulting cell types have not been further characterized (e.g. through FACS sorting, 3D culture etc). The erythroid lineage differentiation is also interesting and partially supported, however it is unclear what insights the iDiscoïd would provide as they lack the proper architecture and efficiency to study this process. The data also suggests substantial levels of spontaneous differentiation, in particular as blood islands occur specifically at the interface of extraembryonic mesoderm and yolk sac endoderm (but not between ExMes and other lineages). The monolayer approach prevents this architecture from forming.

Inhibition of SB43 seems to abolish the entire embryonic compartment. It is unclear how this may relate to amnion formation as it appears to act upstream.

Referee #3 (Remarks to the Author):

I was asked to conduct an ethics review of the manuscript “Modelling Human Post-Implantation Development via Extra-Embryonic Niche Engineering.” In this paper, the researcher developed an embryoïd (called an iDiscoïd) from induced pluripotent stem cell lines. No embryos were used

(directly or to develop an embryonic stem cell line), therefore no IRB approval was obtained nor warranted. The iDiscoid is non-integrated stem cell-based embryo model, therefore in the 2021 guidelines it does not require specialized review (just reporting to a SCRO). However, this research was likely conducted over several months which occurred prior to the release of the 2021 ISSCR guidelines (May 26, 2021). The shift between the two guidelines allowed for longer culturing of the iDiscoid. It is questionable whether this research occurred before or after May 2021 (when a 14-day limit for embryo model research was lifted), but I will give the authors the benefit of the doubt, especially since the researchers obtain approval from the work by a SCRO committee. However, I recommend that both guidelines (2016 and 2021) be referenced in the ethics session with a note that the authors followed both, unless the SCRO approval occurred after May 26, 2021. Other than this minor revision, I have no issues with how the research was conducted.

Referee #4 (Remarks to the Author):

In this study, Hislop and colleagues generated an in vitro model of the human post-implantation embryo using a mixed population of wildtype and inducible GATA6 hiPSCs, termed “iDiscoids”. As well as having a bilaminar disc morphology similar to the human embryo, the authors observed several extraembryonic populations including some potential for haematopoietic differentiation, and initial evidence of an antero-posterior axis with polarised TBXT expression. Current in vitro models of human embryonic development don’t yet model the early post-implantation stage and most lack evidence of early gastrulation, so this is an appreciable advance over existing techniques.

Overall, this is a well-written manuscript that is likely to be of interest across the field of human developmental biology. However, there are several concerns that should be addressed before publication, that are highlighted below:

Major Concerns

1. Considering that formation of the iDiscoid bilaminar disk is a dynamic process, and that the authors observe transient cell states during lumenogenesis and TBXT/MIXL1 symmetry breaking, scRNAseq data from multiple timepoints would, in my opinion, be necessary to support their full claims. This is particularly important since the authors state that the TBXT+ cells are evident at Day 4 and decrease by Day 5, so are therefore not adequately captured by their transcriptomic analysis. Since this observation is the primary advance on existing embryo-like systems, this would be an important stage to characterise and assess.
2. The authors claim a posterior-localised CER1/TBXT polarisation, as outlined in Extended Figure 10. This is a highly surprising observation, since such polarisation has not been observed in non-human in vivo embryos, nor in current in vitro human systems. As such, the authors need to provide additional evidence to support this hypothesis. Firstly, statistical evidence should be provided as to the frequency of this observation and degree of spatial bias, for instance using an angular distribution plot. In addition, evidence should be provided to support their signalling hypothesis for this localisation (in Extended Figure 10). For instance, their BMP4/NODAL inhibitor experiments are not replicated with TBXT/MIXL1 stainings, which could potentially support/disprove such a

mechanism.

3. The authors suggest that there is a temporal progression in iDiscoïd morphology from 'Early' to 'Intermediate' to 'Late' stages (eg Figure 3a), and while the overall distribution of these observed morphologies changes over the timecourse (Day 3-5, Figure 3d) they do not ever show the same iDiscoïd transforming over time from one to the other. As such, this claim is not adequately supported by their data.

4. More generally, given the high throughput potential of this system, more extensive image quantifications are needed throughout the manuscript. What is the variability in area/size/morphology of each iDiscoïd throughout an entire well? How does this affect the patterning observed – does scaling occur? How does it affect the polarisation of CER1/TBXT?

5. The authors state that the WT and iGATA6 cells undergo cell sorting, leading to their distinct domains. Although the supplementary video shows the dynamics of this process to some degree, it is difficult to assess whether cell migration or differential proliferation might be mediating some of this patterning. Since this is a crucial process in early embryonic development, the authors should clarify if the cells are truly sorting, perhaps by cell tracking.

6. The authors stress the transient nature of their structures, including the short-term expression of TBXT on Day 4, and then stop describing their structures after Day 5. What happens after this point, and why is the experiment stopped here? If cell death and/or patterning disruption is occurring, the authors should state and comment on this.

7. It is surprising that in the scRNAseq dataset (e.g. Figure 1d), the epiblast/WT cluster is a single population, whereas Figure 3E and Figure 4 at least two clearly distinct populations are observed by immunofluorescence. Why is this the case?

Minor Concerns

1. Labels to indicate the day that images were taken are needed throughout figures and/or in the legends.

2. Extended data figure 4: UMAPs of iDiscoïds and Human embryo need to be shown separately as well as overlaid. As it stands, it is incredibly difficult to observe the true crossover between the two populations.

3. Extended data figure 9C: Top left image is very blurry, which is concerning. A better-quality image is needed. Also, vertical slices should be shown for all markers including pSMAD1/5/8 and pSMAD2.

4. Figure 1A: the schematic suggests that the flat disc on D2-3 becomes a multi-layered structure on D4-5, but confocal images show only the top view. A cross-section should also be included here to support their schematic.

5. Figure 2B: It is unclear whether the region of interest includes WT cells, since the authors describe the localisation of spindle shaped, CD34/TAL1 expressing cells underneath an iGATA6 endoderm layer. The morphology of iGATA6 cells look like they are sitting on top of a disc shape which contains WT cells. This needs to be clarified by inclusion of a DNA stain or epiblast/pluripotency marker.

6. Extended data figure 13C and E: it looks like there aren't any EGFP-GATA6 cells around the top of the bilaminar disc. This needs to be commented on/clarified.

7. Line 145 and Figure 2B: The authors state that clusters are CD34+/ERG+ implying that these cells are co-expressing these markers, however the figures do not show a co-stain. How have the authors validated this?

8. Extended data figure 2: how many clusters (ie replicates) were measured? What is the significant

difference and variability between clusters in each of these expression levels? It is surprising that the peak of PDGFR α expression is less than one cell diameter from the GATA6 expression. More generally, I find the antibody stain for PDGFR α less than convincing.

9. Line 189: the authors state that iGATA6 cells deposit the laminin ECM. However, there is no quantification/example that shows negative laminin staining if iGATA6 cells aren't present. How have the authors concluded this?

10. Extended data figure 6E: Why is there no HOECHST staining on top of the iDiscoid, despite claiming that the laminin deposition occurs from iGATA6 cells?

11. Extended data figure 5A: the side projection does not show the same rounded discoid morphology. The authors should clarify or comment on this.

12. Extended data figure 2E in legend: currently states "specific subpopulation of cluster 4", but the data shown is for cluster 5, so the legend needs to be corrected.

13. Extended data figure 6A: What do the (1), (2), (3) coloured dots on the right side of the figure mean?

14. Extended data figure 6C: It looks like the cluster labels are incorrect and not in order. This needs to be corrected.

15. To generate their structures, the authors seem to use a ratio of 4:1 GATA6 to WT cells ('Generation of iDiscoid' section) and 10:1 ("10x Genomics Samples Preparation for Next-generation sequencing" section). How does modulation of this ratio affect the outcome? Why are different ratios used? Additionally, in the absence of Dox induction, is this ratio maintained over iPSC culture time/passages?

16. Lines 512, 513: What is the rationale behind adding BMP4 or NODAL inhibitors on different days (day 3 vs day 2) and why were they not added on the same day for direct comparison?

17. Lines 484, 485, 498: the authors should clarify if 10 mM Y-27632 was used, or the standard 10 μ M concentration.

18. Lines 499 and 545: the authors describe considerably different concentrations of Doxycycline used for generation of the iDiscoids. Shouldn't the same concentration of Doxycycline be used throughout the study?

19. On line 56, the authors state that current in vitro models of human embryogenesis have 'low efficiency and throughput [and] limited scalability'. I don't think this is a fair assessment, as most human embryo-like models currently described use methods such as microwells and other techniques that have fairly high throughput.

20. The authors should cite <https://doi.org/10.1038/s41467-017-00236-w> (the PASE model system based on human pluripotent stem cells, which contains an amniotic sac), and inducible GATA6 mouse studies (<https://doi.org/10.1016/j.bpj.2018.11.011>) and (<https://doi.org/10.1242/dev.127530>) on which this work appears to be built.

03/09/2023

Nature Manuscript # 2022-01-20991C-Z, point-by-point response.

Summary of the changes

To our Editor and the Reviewers,

Thank you for your time and for providing us with constructive feedback for improving our manuscript. During the past year, we have done substantial work to address the concerns raised about the initial manuscript.

To briefly reintroduce our study; upon implantation into the uterine wall, the developing human embryo is profoundly remodeled, undergoing key morphological changes that are critical for the success of pregnancy. However, the principles governing this period remain a mystery due to limited accessibility to healthy natural human embryo samples and **an absence of post-implantation models with both embryonic and extra-embryonic tissues**. Additionally, current engineered models of human development often show limited reproducibility and scalability. They also exhibit limitations that hamper their use across labs, including technical complexity, challenges regarding finding a common media that can support multiple fates, or in some cases use of supra-physiological levels of growth factors, and low efficiency.

We present ***iDiscoid***, a ***genetically engineered embryo model system***, that overcomes several of these limitations and mimics key facets of ***human post-implantation embryogenesis***. We built *iDiscoids* via engineering genetic circuits in human induced pluripotent stem cells (hiPSCs). The *iDiscoids* exhibit reciprocal co-development of embryonic and extra-embryonic tissues, formation of human bilaminar disc-like structures containing epiblast and hypoblast, lumenogenesis, development of the amniotic ectoderm, anterior hypoblast-like population, and a posterior axis, as well as hematopoiesis within a yolk sac. Many of the associated features are pre-programmed into hiPSCs via an inducible genetic circuit, which sets them apart from existing human models. As such, they are highly scalable and can be frozen, shipped, and thawed on demand for use across labs using conventional 2D plates, making this technology widely accessible to researchers. *iDiscoids* exhibit a structurally relevant and transcriptomically comparable reconstruction of the *in vivo* events of human embryogenesis at an early stage when the embryo *in vivo* is at its most inaccessible.

In this version, we have performed significant revisions to provide compelling data addressing the concerns raised by our reviewers. Our reviewers initially had raised questions or concerns primarily in respect to the aspects below, and we have made sure all are completely addressed.

- a. Number of lumens developed in iDiscoids in the context of physiological relevance to human.*
- b. Whether iDiscoid is a 2D system rather than a 3D platform.*
- c. Timeline of the system dynamics, developmental age in the context of human development*
- d. Presence of blood island structures and relevance of structures shown*
- e. CER1/TBXT dynamics and limited similarity to in vivo*
- f. Quantification of the structures developed in the study.*

To this end, in summary, we have provided revisions below:

Lumenogenesis: We have optimized *iDiscoid* production methodology and show control over disc size, the number of developing *iDiscoids*, and their subsequent lumen formation by modulating cell ratios and initial seeded cell number. **These *iDiscoids* recapitulate reported lumenogenesis similar**

to *in vivo* embryos via initiation and expansion from a single lumen. Please see Fig. 1C and Supplementary Fig. 3C-D.

3D Self-organization of iDiscoid bilaminar disc: We capture lateral views of iDiscoid development live over a period of time using time lapse microscopy that shows 3D organization of iDiscoid and development of an amniotic-like cavity and bilaminar disc-like structures within a single iDiscoid over time (see Fig 1C).

Developmental staging with detailed characterizations:

We have done extensive characterization of tissue development via single cell analysis at seven time points from induction of iDiscoid through the development of the structures presented, as well as one for long-term follow-up to examine development within the iDiscoid yolk sac-like tissues. We subsequently performed careful comparative analyses of iDiscoid development to transcriptomes of the E16-19 human embryo, E6-E14 human embryo samples, and E18-20 cynomolgus embryos.

Blood island development: Intriguingly, our extended follow up has revealed that within the yolk sac-like cells, away from the bilaminar disc structures, 3D self-organized structures emerge with structural similarities to bona fide blood islands, with endothelial cells and blood progenitors positioned between opposing yolk sac endoderm and mesoderm layers.

Posterior axis and anterior hypoblast domains: Additionally, we have taken advantage of recent spatial data in human and non-human primates to perform careful analysis of TBXT expression in the posterior and CER1⁺ anterior hypoblast cells in those datasets. We show that the dynamics observed in iDiscoid demonstrate relevant patterning dynamics similar to CS6-CS8 anterior and posterior populations.

Quantitative profiling: We also report numbers and percentages for each one of the observed phenotypes.

Summary of quantitative profiling data:

- We characterized the features of WT clusters within iDiscoid culture, including disc area, disc circularity, and number of lumens that occurred in each individual disc. We also quantified the number of lumens observed within the disc and, following optimization of the iDiscoid ratio, showed a substantial (33.2% → 70.9%) increase in the consistency of single lumen iDiscoids in a physiologically relevant size range (Supplementary Fig. 4).
- We have seen amnion development very consistently within iDiscoids, with 98.8% of WT clusters expressing the amnion marker ISL1 following day 3 of development (this is measured in 88 iDiscoids and is consistent across 3 independent experiments). iDiscoids analyzed show that all but one of the observed WT clusters (n=87) have ISL1 expression above the expression levels in the negative control (Fig. 2I).
- We have reported the frequency of anterior hypoblast-like domains and posterior-like regions and also assessed the positioning of these poles within iDiscoids. Development of a TBXT-expressing polarized posterior domain was observed in 30.1±6.3% (215 out of 746) of iDiscoids, verified across 4 independent experiments (Fig. 3E). Out of 396 iDiscoids assessed, we identified 42.4±4.1% (169) with polar CER1-expressing domains on day 4 across 3 different experiments (Fig. 3F), showing a similar efficiency to existing *ex vivo* human blastocyst cultures (Molè et al, Nature Commun. 2021).
- Via confocal microscopy, we examined 5 total areas containing TAL1⁺ cells (hemogenic endothelial-like) across 3 independent samples and verified the consistent arrangement of these cells underneath a layer of iGATA6 cells at day 5. We report the specification of RUNX1⁺ cells in 10.1% of analyzed TAL1 cells.
- We reported numbers of CD43⁺ cell development in yolk sac (after day 5) as a surrogate for the establishment of blood islands (Fig. 4H) and show supplementation with an inducible “high” GATA6 cell population can boost this phenotype, reflecting that we can exert control over final cellular

composition (a topic for future generation of iDiscoids by design). We also show flow cytometry data in tandem with scRNA-seq for profiling CD43⁺ cells (Supplementary Fig. 14D-G).

- We investigated 9 total areas across 3 independent iDiscoid samples to verify the cellular organization of yolk sac endoderm, yolk sac mesoderm, and endothelial-like cells in a blood island-like orientation (Fig. 4H-J). We show an intricate self-organization mimicking what has been reported *in vivo* (Ross C, Boroviak TE. Nat Commun. 2020).

We believe we have addressed the previous comments and concerns. iDiscoid enables further study of key events during post-implantation development, including emergence of a hematopoietic niche in a complex yolk sac model, formation of an amniotic cavity, specification of an amnion-like fate, and development of anterior hypoblast population in parallel with a TBXT⁺ posterior domain to form an A/P axis. To our knowledge, these characteristics, and especially in an integrated format, have not been reported so far in stem cell-based models of human embryogenesis.

Please find our point-by-point response below. Hopefully our reviewers will also find the revised manuscript satisfactory, significant, and innovative.

Referee(R) #1 (Remarks to the Author):

Hislop et al.

In this study Hislop et al. report a new methodology for modeling early human development that involves recapitulating interactions between pluripotent cells and extraembryonic cells via an engineering approach. Cells overexpressing GATA6 are combined with human pluripotent stem cells, and grown in standard 2D cultures. The authors report the formation of cell populations with gene expression consistent with extraembryonic endoderm and mesoderm, early hematopoiesis, amnion and primitive streak. In some instances, there is evidence that the cell types are in appropriate anatomical configurations.

Although the authors present some intriguing data, in a number of instances the temporal sequence of events, and the cell identifications, are not fully supportive of the claim that the model represents normal human development. Many of the phenomena described have already been reported in conventional human pluripotent stem cell culture, and the study fails to show how exactly the engineered GATA6⁺ cells contribute to the subsequent differentiation and morphogenetic events. The most interesting claim is that of formation of blood islands, but the evidence to support this interpretation seems rather thin. The authors need to provide more information on the reproducibility of their observations, in particular, how often structures with the arrangements of cells and tissue depicted were formed.

Response:

We thank R1 for the assessment of our initial manuscript and the feedback provided.

To address R1's concerns we performed several new studies:

- We did timeseries scRNA-seq in iDiscoid culture, as well as comparison with *ex vivo* embryos showing sequential post-implantation fate decisions in line with human embryogenesis development.
- We also provided appropriate controls (including control single cell analysis) to show differences with conventional human pluripotent stem cell cultures, clearly differentiating the platform from behavior observed in those cultures.
- We show GATA6-expressing cells contribute to different subset of cells in yolk sac based on initial GATA6 dosage. They also participate in the formation of CER1⁺ anterior hypoblast cells. We also show interaction of iGATA6 cells with WT cells result in formation of WT clusters, lumenogenesis,

and cavity formation. Cultures in equivalent conditions without the addition of Dox to induce GATA6 expression fail to show any of these events (see Reviewer Fig. A, below).

- We have strengthened the data for blood island formation. Based on the comments, we have shown clear organization of hemogenic endothelial-like structures at day 5, as well as quantification of marker expressions TAL1 and ERG at this time point. We also show the development of a subset of CD43⁺/Hb⁺ cells within the iGATA6 layer by day 12. We have also characterized the structure of these areas and confirmed that there is an arrangement of mesodermal, endothelial, and endodermal cells that has structural similarity to *in vivo* blood islands.
- In this version of manuscript, we also show measurements related to the reproducibility of data and how often iDiscooids show the reported features. In **Supplementary Figure 4** we show specific metrics related to iDiscooid WT cluster size and shape as it relates to lumen formation and report the lumen numbers after and before optimizations; in **Figure 21** we show the intensities of WT clusters expressing ISL1 in comparison to a negative control condition per tested iDiscooid; in **Figure 3E-H** we quantify the occurrence and characteristics of the anterior and posterior axis formation within iDiscooids; and in **Figure 4** and **Supplementary Figure 13** we have investigated hematopoietic features and also measured metrics related to day 5 hemogenic endothelial-like cell positioning. For more details please see below.

Reviewer Figure A: Comparison of D5 iDiscooid to D5 iPSC culture. D5 iPSCs fail to show formation of 3D bilaminar disc-like structures or exit from pluripotency in iDiscooid conditions.

Specific comments

1. *To refer to the iDiscooid as a synthetic biology construct is a bit of a stretch. Overexpression of master regulatory transcription factors, as done here with GATA6, has been described in the field for years and is not considered by most as synthetic biology. The relevant gene circuits are not really regulated in any sense, nor are they particularly well described.*

Response: Thank you for this comment. We have removed the claim regarding iDiscooid being a synthetic biology construct. One can control the temporal on/off regime as well as the expression level of GATA6 based on Doxycycline addition or removal. We have taken advantage of this feature by removing Dox beyond day 5 in our extended follow ups to facilitate hematopoietic differentiation (detailed in Fig. 4 and Supplementary Figs. 13, 14, and 16). However, we agree that the inducible expression of a transgene is a simple form of a gene circuit. While our plan initially was to build more advanced genetic circuits, our initial test exhibited interesting phenotypes on its own, which resulted in our further characterizations and set the stage for employing more complex circuits. We have described the inducible circuit in the methods.

2. *Ext data Figure 1-were expression levels of GATA6 regulated and if so, how?*

Response: In this study, we have kept the dosage of Doxycycline the same across each day of treatment (mTeSR-1 with 1 µg/mL doxycycline to induce expression of the GATA6 transgene) for 5 days. In the case of extended follow ups, we remove Dox and change the media to IMDM (without Dox) from the sixth day and on.

Additional notes: In our pilot tests, we did experiments supplementing different levels of Dox and did not observe a significant change in culture behavior through day 5, except at very low Dox levels, where circuit function is fragile. The GATA6 expression shows heterogeneity across the hiPSC population

(i.e., as a function of circuit copy number) (see Reviewer Fig. B). We demonstrated the ability to control proportions of GATA6 cells expressing a specific GATA6 level upon induction, where we show that replacement of up to 25% of the heterogeneous iGATA6 mixture with a high-GATA6-expressing cells increases the number of hematopoietic CD43⁺ cells that develop within the culture (by up to roughly fourfold) (see Fig. 4F-G). The data confirmed a correlation between the high-GATA6-expressing cell population (supplemented at day 0) and hematopoietic features (assessed at day 12) in the study, further emphasizing of the utility of the platform to examine cellular trajectories associated with described phenotypes.

Reviewer Figure B:
GATA6 and GFP
expression heterogeneity
 Crops from Supplementary Fig. 1B and 7D showing expression heterogeneity within different iDiscoid cells at day 4 of culture

3. Page 2 L 80-if the authors begin with conventional hPSC, which correspond to post-implantation cells, how is it that the model initially mimics the D6 inner cell mass.

Response:

Thank you for R1's important comment. We agree that our initial description of the system might have fallen short by linking the initial events to preimplantation D6 inner cell mass. Following the comment, we have done back-to-back comparisons to study the similarity of each stage of iDiscoid development to human embryo samples. We show the developmental stage of iDiscoids as early as 36 hours show strongest similarity to day 12 post-implantation human embryos from Xiang et al. Nature 2020 (Supplementary Fig. 6B, see Reviewer Fig. C below). Thus, we have adjusted the text accordingly, removing references to preimplantation similarity and focusing our observations on post-implantation-like events. The iDiscoid model from day 2 starts by mimicking the cell types of the D12 post-implantation human embryo but continues to show orchestrated maturation and development of endoderm- and mesoderm-associated fates within yolk sac, epiblast, amnion, and formation of the posterior axis (Supplementary Fig. 6).

Reviewer Figure C: An excerpt of Supplementary Fig. 6 in the manuscript showing hypergeometric statistical comparisons of iDiscoids to Xiang et al. 2020 human embryo data.

4. Page 2 L 80-were the GATA6 cells characterized further? Data in Ext Data 2 is not sufficient to indicate that these cells are primitive endoderm

Response:

We performed single cell analysis at different timepoints during the development of iDiscoid. We analyzed the iGATA6 cells further and show composition of cell types with respect to primitive endoderm and its derivatives yolk sac endoderm and yolk sac mesoderm in Fig. 1F-G and Supplementary Fig. 6. We have shown a subset of markers (i.e., PDGFR α , GATA4) within the iGATA6 layer in Supplementary Fig. 2A, D. We show ELISAs for AFP and APOA1 proteins in Supplementary Fig. 3F to show that the secretion of yolk sac-related soluble factors increases after the induction of iDiscoid culture.

5. What is an appropriate control for experiments in Figure 1 and elsewhere? Cells expressing many of these genes could be found in conventional embryoid bodies, or overgrown adherent cultures. In fact, previous studies have described appearance of extraembryonic endoderm like cells around the periphery of stem cell colonies grown under standard conditions (e.g. DOI: 10.1016/j.cell.2019.03.013 and others).

Response:

We respectfully have to disagree with our reviewer on this point. The self-organized morphological features, specific clusters with defined *in vivo*-like fates with high frequency and reproducibility observed in our iDiscoid will not occur in the overgrown adherent cultures or even embryoid bodies. Our single cell analysis derived from hundreds of iDiscoids shows comprehensive analysis of cell types present in these cultures (Fig. 1F-G) that specifically relate to a defined window of time in post-implantation embryogenesis. Self-organization and tissue boundary formation Fig. 1B-D cannot be seen without addition of Dox (Reviewer Fig. A (above)) or at different seeding densities.

[Text Redacted]

[Text Redacted]

Extraembryonic endoderm-like cells around the periphery of stem cell colonies : It is correct that a previous work (Nakanishi et al. Cell 2019) described appearance of extraembryonic endoderm-like cells around the periphery of stem cell colonies grown under standard conditions. However, what is reported in this paper is a small population of cells at the edge of PSC colonies and differs substantially from the characteristics of iDiscooids we report here, including the acquisition of cell types beyond primitive endoderm identities, the formation of 3D morphology, and reciprocal co-development of anterior, posterior, dorsal, and ventral axes in addition to hematopoietic characteristics. We do feel that there is significant evidence to clearly conclude that iDiscooids possess characteristics that would not be anticipated in overgrown cultures.

6. P3 L86 figure 1C does not show segregation of the two cell types. Supplemental 2c is confusing in this respect. The GATA6 only cells could just be at any early stage of extraembryonic specification. Since they are in close contact with pluripotent cells their identity is important.

Response:

We thank R1 for the comment.

- We have adjusted the language and removed the claim about cell sorting/segregation and simply mention : “WT cells are organized into clusters confined by iGata6 cells”.
- In this version of manuscript, we used scRNA-seq to characterize each cell population and show that within this system, iGATA6 cells primarily mimic post-implantation human extraembryonic cell derivatives from E12 onward (see Supplementary Fig. 6). We also show staining for cell types: We demonstrate that these iGATA6 express key markers of extraembryonic endoderm fate in scRNA-seq (Supplementary Fig. 5) and via immunofluorescence (Supplementary Fig. 2A, 2D)
- We agree that the initial Supplementary Fig. 2 was confusing. In light of our new additions, we decided to remove this figure, as it is not adding any relevant data to the current study.

7. P3 L 97-why is the comparison with E16-19 embryo appropriate here? The timing is confusing here, since E16-19 is well into gastrulation, and the authors' model should be at an early post-implantation stage.

Response:

We thank R1 for her/his feedback. We initially chose the Tyser et al. dataset because it is the only source of human single cell data that we are aware of that has cell types corresponding to amnion, early hematopoietic progenitors, and cell types emerging from the primitive streak in one dataset. In the current manuscript, we have additionally compared iDiscooids with human single cell datasets from Xiang et al. 2020 spanning embryos from E6 to E14 of development, as well as an additional cynomolgus E18-20 NHP dataset from Ma et al. 2019. We demonstrate that iDiscooid transcriptomes capture cell stages

Reviewer Figure E: A crop of Fig. 1 showing panels 1F-G from the manuscript.

that correspond to approximately E12-E16 human embryogenesis (Supplementary Fig. 6, Fig. 1, Reviewer Figure E).

8. Page 3 L104-it is not clear that Cluster 0 contains PS-like cells, and it would be helpful to know what stage of epiblast it corresponds to.

Response:

We thank the reviewer for the comment that was made. In this current version of manuscript, we have performed full characterization of WT cells using timeseries scRNA-seq. We show the WT cells maintain a state similar to human E12 epiblast from about 12 hours after induction until the specification of different epiblast-derived tissues, observed at around day 3 to 4 of iDisco development. We show the presence of amnion-like, epiblast-like, and PS-like cells, and characterize these cell types further. Please see Figs. 2, 3, and Supplementary Figs. 5-7 for more information.

9. P4 L 142-AFP is not a good marker for early erythropoiesis, and neither is APOA1. Both are expressed in yolk sac endoderm

Response:

We have adjusted the text. We have assayed these two markers as proteins produced by yolk sac in the media (now Supplementary Fig. 3F).

10. P4 L145-better characterization of “endothelial” cells is required.

Response:

We initially had shown spindle-shaped cells emerging around day 5 that were positive for TAL1 but also expressed ERG, and RUNX1 as well as CD34, suggesting hemogenic endothelial-like characteristics. In this version of the manuscript, we performed quantitative image analysis to understand the distribution of the markers ERG and CD34 in TAL1⁺ cells (Supplementary Fig. 13A). We also stained for VE-cadherin and VEGFR2 (KDR) on day 5 and show expression in these spindle-shaped cells (Supplementary Fig. 13E). Overall, endothelial-like cells have limited frequency at this time point, comprising only a small percentage of the overall population, as verified by flow cytometry analysis (1.13% of total cells). To better study the potential of these cells, we further followed up the cultures and show expansion of CD34⁺ cells around day 12. We measured this event using flow cytometry and image analysis (Supplementary Fig 13G). Our extended follow up shows development of blood islands formed around day 12 which also shows clear development of round CD43⁺ hematopoietic cells within vascular tubes (Fig. 4E). Late time point scRNA-seq analysis also shows genes associated with endothelial cells (e.g., CD34, CD31, KDR, VE-cad, CD93) (Supplementary Fig. 14F).

11. P4 L 147-what proportion of cells were double positive for RUNX and TAL1 as in C and how often were they found in the indicated arrangement? The introduction points to inconsistency in previously published human embryo models, a valid point, but throughout this study there is no quantitative indication of how reproducible the structures shown are.

Response:

Summary of data:

We thank R1 for critical assessment of our work.

The frequency of RUNX1⁺ cells across these TAL1⁺ endothelial-like cells has been measured and reported in Supplementary Fig. 13A. Also, as suggested by this reviewer, we examined how frequently endothelial-like cells developed under the endoderm layer. We investigated 5 total areas (using confocal microscopy) expressing ERG across 3 independent samples and verified the consistent arrangement of endothelial cells underneath a layer of iGATA6 cells at day 5. The data shows ERG⁺ cells positioned under the other EGFP⁺ cells (Reviewer Figure F).

We also report numbers and percentages for each one of the other observed phenotypes.

- We characterized the features of WT clusters within iDiscoid culture, including disc area, disc circularity, and number of lumens that occurred in each individual disc. We also quantified the number of lumens observed within the disc and, following optimization of the iDiscoid ratio, showed a substantial (33.2% → 70.9%) increase in the consistency of single lumen iDiscoids in a physiologically relevant size range.
- We have seen amnion development very consistently within iDiscoids, with 98.8% of WT clusters expressing the amnion marker ISL1 following day 3 of development (this was done in 88 iDiscoids and is consistent across 3 independent experiments). iDiscoids analyzed in Fig. 2I show that all but one of the observed WT clusters (n=87) have ISL1 expression above the expression levels in the negative control.
- We have reported the frequency of anterior hypoblast-like domains and posterior regions and also assessed the positioning of these poles within iDiscoids. Development of a TBXT-expressing polarized posterior domain was observed in $30.1 \pm 6.3\%$ (215 out of 746) of iDiscoids, verified across 4 independent experiments. Out of 396 iDiscoids assessed, we identified $42.4 \pm 4.1\%$ (169) with polar CER1-expressing domains on day 4 across 3 different experiments, showing a similar efficiency to existing *ex vivo* human blastocyst cultures (Molè *et al*, *Nature Commun.* 2021).
- We reported numbers of CD43⁺ cell development in yolk sac (after day 5) as a surrogate for the establishment of blood islands (Fig. 4F-G) and show supplementation with an inducible “high” GATA6 cell population can boost this phenotype, reflecting that we can exert control over final cellular composition (a topic for future generation of iDiscoids by design). We also show flow cytometry data for profiling CD43⁺ cells (Supplementary Fig. 14G).
- In vivo like self-organization of blood islands: We investigated 9 total areas across 3 independent iDiscoid samples to verify the cellular organization of yolk sac endoderm, yolk sac mesoderm, and endothelial-like cells in a blood island-like orientation. We show an intricate self-organization mimicking what has been reported in vivo (Ross C, Boroviak TE. *Nat Commun.* 2020).

12. *Figure 2D-CD71 cells require further characterization. Human yolk sac hematopoiesis has been extensively studied in vitro; much more convincing evidence is required here to support the conclusions that blood islands have formed. Percentages of cells expressing specific markers are very low (Ext data Figure 5).*

Response:

We had initially showed characterization of day 5 hematopoiesis using flow cytometry, and we agree that the cells at this timepoint are just emerging. Hence the observed frequencies are low. Additionally, CD71 is also expressed in proliferating cells, making it less specific. We have included other markers in the current version of the manuscript. We do appreciate in some instances the percentage of cells is not very high; however, the total number of cells in iDiscoid represent many different populations derived from both WT clusters and yolk sac cell types, so this is not unexpected.

We now provide quantitative image analysis, scRNA-seq, and flow cytometry for later timepoints for hemogenic endothelial and hematopoietic-like cells, where the subset of cells that develop are more amenable to characterization. We have also shown staining for CD43, CD235a, and hemoglobin at day

12 iDiscoid in Fig. 4F. We show a clear increase in hematopoietic foci (blood island-like structures) and maturation of structures with *in vivo*-like cell organization. (Please see Fig. 4 also in Reviewer Figure G, and Supp. Fig. 13-16).

In Supplementary Fig. 14G, we show that 9.31% of CD43⁺ hematopoietic progenitor cells are CD235a⁺/CD71⁺ erythroid-like and 11.4% are CD33⁺/CD31⁺ myeloid-like, in addition to 4.5% CD42b⁺ megakaryocyte-like cells. We have also shown single cell characterization and expression of many key markers associated with yolk sac tissues in our extended follow up (please see Supplementary Fig. 14).

13. P5 L 171-cavitation and amniotic cavity like morphogenesis have been described by others. The structures appear upside down in Figure 3 e-g. What additional characterization of amnion was carried out? How does the timing here relate to an E16-19 human embryo?

Response:

We are not sure if we understand the initial part of the comment, and if being upside down would be an issue. To clarify, the structure we develop shows an embryo-like structure that is attached to the dish

from the amnion layer, contains an amniotic cavity-like structure, and a yolk sac “cavity” that is formed out of the yolk sac cells attached to the dish and facing towards the media. The yolk sac secretes factors into the media. **iDiscoid is similar to an “unrolled” yolk sac cavity with epiblast that has been separated from its neighboring tissues (i.e., trophoblast and endometrial tissue) and attached to a culture dish (Reviewer Fig. I).**

In this version of the manuscript, we have characterized the amnion layer further via scRNA-seq, and show co-expression of markers such as AP-2 α , GATA3 and OCT4 in addition to ISL1. We also show AP-2 α expression in staining in Supplementary Fig. 8. In our new data, we show that the **amnion fate makes a clear cluster in the WT population in the scRNA-seq data**, distant from NANOG-expressing pluripotent cells (Reviewer Fig. J).

We have compared our amnion layer against human and non-human primate (NHP) datasets containing annotated amnion cells in order to address the timing. The amnion layer that matures in iDiscoid from days 3-5 has a signature most similar to human and NHP amnion just prior to E16-19. **On day 4, this tissue (labeled as iDisc_D4_W3) is highly similar to early (E-AM) to late (L-AM1)**

amnion D12-D16 from Ma H et al in Science 2019 that shows cynomolgus monkey blastocysts beyond early gastrulation. Also, it correlates with the amnion signature from Tyser et al.

14. Page 5 L185-is trophoderm absent from the model?

Response:

That is correct. Trophoderm is absent from iDiscoid. This also alleviates some of the ethical challenges associated with producing in vitro stem cell based human embryo-like models.

15. Figure 3-again at what frequency were structures like those in E-G observed? How common were the structures with multiple cavities shown in C observed and what are they?

Response:

We would like to thank R1 for her/his careful assessment which helped us to improve our iDiscoid development strategies. We have performed extensive studies on how to develop iDiscoids and provided in this version of the manuscript our key findings. We developed a quantitative MATLAB pipeline to examine the iDiscoid morphology carefully. Our initial effort had generated a variety of sizes of iDiscoids, and when analyzed in detail it showed that lumen formation is correlated with disc area (Supplementary Fig. 4A) (also in line with data from Orietti LC et al, Stem Cell Reports 2021).

To develop control over the size, number, and circularity of WT clusters, we have performed high-throughput optimization of iDiscoid creation. Our efforts reveal that modulation of the ratio and number of initial cells seeded provides a way to control the size and number of WT clusters within iDiscoids (see Supplementary Fig. 4). **As a result, by choosing appropriate conditions, we can achieve singly-lumen iDiscoids that are in a physiologically relevant size range.** When we study these cultures, we subsequently show that our predication can match the outcome and we can improve the frequency of structure and single lumen formation substantially (33.2% → 70.9%). We had initially seen multiple lumen formation in 63% of iDiscoids (207 out of 331). Subsequent to optimization of the iDiscoid seeding ratio, among 79 WT discs, 58 had a single lumen and had areas in a range of 10367 to 43715 μm^2 , equivalent approximately to the lateral area of the CS5b – CS6 human bilaminar disc (as recorded in Carnegie #8004, #7700 in Hertig and Rock, 1949 and Carnegie #7801 in Heuser et al. 1945).

16. Extended data figure 9-is BMP active in this system in the pg/ml range?

Response:

The initial measurement on day 5 showed 1.5 pg/ml of BMP4 in the media. We have evidence that the amnion-like tissue produces a significant amount of BMP4 beginning at day 4 (as evidenced by high BMP4 transcripts in the day 4 scRNA-seq; see violin plots taken from Supplementary Fig. 5B in Reviewer Figure K). We believe the BMP4 is produced into the cavity (resulting in a high concentration of BMP4 in amniotic cavity in iDiscoids) and at day 5, due to potential release from a subset of cavities

(i.e., possibly due to rupture), the BMP4 is released in the media outside of the cavities resulting in the increase in concentration in the media observed. We have confirmed that BMP4 is active in the system, both via observation of phosphorylation of the BMP-specific SMADs 1, 5, and 8/9, and observing that inhibition of BMP4 signaling via application of the inhibitor Noggin prevents the specification of ISL1-expressing cells as well as the posterior axis (please see Fig 2G-I, Supplementary Fig. 9). However, we are not confident that the ELISA data showing a pg/ml concentration of BMP4 in the media is an accurate reflection of its expression within the system (secretion directly into media). As such we have removed this data from the current version of the manuscript to avoid inaccurate information.

17. Page 6 L229-others have shown this distribution of SMAD signaling in conventional hPSC cultures

Response:

We agree that phosphorylated SMAD1/5/8 shows high expression in the border of hPSC cultures. We have verified this pattern of expression in our own iPSC controls. **However, the new data we provide shows substantially higher signal intensity within the iDiscoïd amnion cells with ring-like pattern (Reviewer Figure L).** This may show the signaling initiates from this ring-shaped domain, or that the cells receive a BMP4 signal at that position and move into the center-base of the cavity-containing WT cluster to form the amnion-like layer (which is a subject for future investigation).

Additionally, in iDiscoïd staining when we compare the BMP4 signaling in this amnion-like layer versus the pluripotent epiblast-like layer, we can observe clearly augmented BMP4 signaling in the amnion, demonstrating that this is an important signaling event for amniotic fate acquisition in iDiscoïd cultures (see Reviewer Figure K). This is shown in staining and also confirmed in single cell analysis (Fig. 2F, Supplementary Fig. 6B).

Furthermore, as mentioned, Noggin treatment also decreased ISL1 expression in iDiscoïd WT clusters, further emphasizing that the SMAD signaling observed plays a functional role in the development of the system (Fig. 2H-I).

18. P7 L245-the role of Nodal signaling in amnion induction is not clear from these experiments.

Where are data on the other components of the receptor system? The inhibitor used is not specific for Nodal.

Response:

Reviewer Figure L: pSMAD1/5/8 distributions at the edge of WT cultures and iDiscoïds

In both iDiscoïds and in WT monoculture, the concentration of the pSMAD1/5/8 signals around the edge of the WT colonies. **Both images above were imaged using the same imaging parameters.** pSMAD1/5/8 activation within iDiscoïds is much stronger than the equivalent signal in WT colonies.

Reviewer Figure K: Subset from Supplementary Figure 5B showing markers of the amnion-like population including elevated BMP4 transcripts.

[Text Redacted]

[Text Redacted]

19. *Figure 4-these figures show segregation of T+ cells, but do not convincingly demonstrate formation of a primitive streak.*

Response:

We agree and apologize for how we have explained this observation. We corrected the language to specify we only observe posterior fate formation. We do not claim actual structures with primitive streak-like morphology. We mention primitive streak only in the context where it relates to statistical comparison to primitive streak from *ex vivo* human datasets.

20. Page 7 L265 onwards-it is not clear what the authors are trying to say regarding CER1. Cer1 is a BMP and WNT antagonist expressed in the anterior domain of the visceral endoderm. Why it should be expressed near epiblast destined to give rise to T+ cells in the authors' model is unclear. There is no direct evidence presented to show that CER1 acts as a Nodal antagonist in this system.

Response:

We appreciate the point raised by our reviewer and agree that expression of CER1 near the TBXT⁺ region at first rather sounds surprising. Our initial observation in this respect did not have the quantitative analysis necessary to back up this finding. Therefore, in this version of manuscript, we have comprehensively profiled iDiscooids anterior-posterior (A/P) axis formation in large numbers and compared with *in vivo* data from published embryos in human, cynomolgus monkey and mouse. In our new analysis we do see the classical form of TBXT/ CER1 positioning with a polar configuration which is expected from several past studies. However, we also report a subset that doesn't follow this pattern and compared to *in vivo* dataset. Please see below.

Human CER1: It is the case that data on human CER1 is limited, and studies suggest species-specific differences. The sequence of Cerberus protein differs significantly among species (only 69% identity between mice and humans) (Aykul S et al 2015, PLoS ONE). The large variations in amino acid sequence may result in unique binding specificities and different functions. Aykul S et al also assayed human Cerberus binding activity and revealed that it binds strongly and inhibits Nodal signaling. Among TGFβ family ligands BMP-2 and GDF-11 could bind loosely to Cerberus but human Cerberus did not bind any other tested TGFβ family ligand with consequential affinity, including BMP-4 and Activin A (in contrast to mouse protein, see Fig 3F in the paper by Aykul S et al). Hence CER1 function in human in respect to A/P axis formation may not fully mirror past data in mouse models. More studies in human are required but this has been hampered by lack of access to human embryo or stem cell derived models with a CER1⁺ population.

Anterior-posterior (A-P) axis in iDiscooids: Following analysis of 169 of iDiscooids assessed, we observed that 60.4±5.3% (102 of 169) of iDiscooids with polar configuration in CER1 expressing cells possessed a TBXT-expressing pole (**Fig. 3F**). 42.2±3.2% (43) of these iDiscooids exhibited anti-polar configuration, where the TBXT⁺ pole occupied the opposite radial region as the CER1⁺ domain, in line with past data in mouse studies (Fig. 3F-H). In a second set of iDiscooids, we observed high expression of CER1 protein in the proximity to the TBXT⁺ domain (Supplementary Fig. 10A). Surprisingly, we also noted cells in the posterior domain that expressed both TBXT and CER1 proteins (**Supplementary Fig. 10A-B**). These iDiscooids showed CER1⁺ and TBXT⁺ poles that shared the same radial region (syn-polar). A closer look at scRNA-seq analysis also showed a population of cells in our posterior-like cell cluster that expressed CER1 (**Supplementary Fig. 10C**).

In vivo datasets: Subsequently we examined the recent scRNA-seq data from human or non-human primates. In marmoset embryo datasets (Bergmann et al. Nature 2022), we noted non-polar CER1 at CS5, followed by very slight polarization away from the posterior at CS6 (a subtle anti-polar format) (**Supplementary Fig. 11A**). In human E16-19 (CS6) (Tyser et al. Nature 2021), we observed higher CER1 transcript expression in the same caudal embryonic region as the primitive streak (**Supplementary Fig. 11B**). Lastly, pseudospacial assessment of the E18-20 cynomolgus embryo (Cui et al 2022, Cell Reports) shows CER1 and TBXT expression overlapping within the lower region of embryonic disc (hypoblast) and embryonic disc compartments (a syn-polar format) (**Supplementary Fig. 11C**). Interestingly, all three datasets show a subset of cells or regions that display co-expression of CER1 and TBXT in the posterior domain. When we examined mouse scRNA-seq datasets (Pijuan-Sala et al Nature 2018, Mittnenzweig et al Cell 2021, Peng et al. Dev Cell 2016), we could confirm Cer1

gene expression presents at the anterior area of primitive streak (APS) and is induced subsequent to *T* expression (E6.5-E7.5) (**Supplementary Fig. 11D-F**).

Conclusion: Taken together, our analyses of the available *in vivo* data suggests that the observed expression and positioning of TBXT⁺ and CER1⁺ cells in iDiscooids is more than an *in vitro* artifact and can present two stages representing development and progression of posterior domain. Anti-polar positioning of CER1 and TBXT domains reflect positioning of anterior hypoblast-like and posterior domains during early primitive streak formation around E14; syn-polar positioning and co-expression of CER1 and TBXT is reflective of a more advanced stages post-E14 where CER1 expression acts either to prevent spreading of posterior domain or represent early mesendoderm commitment derived from a primitive streak. In the current version of the manuscript, our added data clearly shows *in vitro-in vivo* correlation that was revealed only via using iDiscooids with subsequent assessment of *in vivo* datasets.

Referee #2 (Remarks to the Author):

Modelling Human Post-Implantation Development via Extra-Embryonic Niche Engineering
The iDiscooid model consists of a mixed population of wildtype and GATA6 inducible human ESCs. This simple approach resulted in a surprising degree of complexity and led to organized structures seemingly comprising major cell types of the human periimplantation embryo, including putative amnion-like, primitive streak-like, pluripotent cells, visceral endoderm-like cells, and yolk sac cells. Overall, the paper is well presented with clear schematics.

Response:

We thank our reviewer for her/his time and detailed analysis of our manuscript. After receiving R2 comments, we designed and ran a series of experiments to address R2's concerns. Through these studies, we gained new insight into iDiscooid generation, methodology and behavior *in vitro*. We believe the current manuscript has addressed the main R2 concerns to the extent that we can present iDiscooids as acceptable and useful model to study human post-implantation events with high degree of practical convenience as also highlighted by R2.

Despite interesting observations being reported, the validity of such a model to recapitulate genuine events of human development is questionable. The "model" consists of a carpet of millions of cells, which is hardly corresponding to the small periimplantation in the womb. Although such a 2D culture system has practical conveniences, it will show many artefacts – which can be readily seen in the data.

Response:

We thank the reviewer for the comment. To address the concern raised by R2 in this section, we have provided substantial new data and an optimized version of iDiscooids in the revised manuscript. We show in different instances in the revised version the utility of this model to capture developmental events both as proof of principle and as new findings not reported or modeled before.

- Our single cell RNAseq timeseries now shows iDiscooids follow a clear stepwise maturation culminating in a model with high statistical similarity to human/NHP post-implantation embryonic cell types when compared with multiple datasets. The data further shows iDiscooids match human developmental steps and transcriptomically align with post-implantation stages [approximately ~E12-E16].
- After our optimization of iDiscooid seeding ratios, we show the optimized version of iDiscooid grows within a physiological size range (between CS5b and CS6 human embryo) and exhibit lumenogenesis that is in line with human development (please see Supplementary Fig. 4). We would like to draw the attention of R2 to the fact that each iDiscooid is not millions of cells. An individual iDiscooid is in a size range of post implantation embryo, and is comprised of tens to hundreds of cells. Many iDiscooids can be developed together, connected via shared yolk sac-

like tissues (Fig 1B). This is an advantage of the system for high throughput studies, providing large sample sizes and enables understanding any mechanism of variation that might be observed across the iDiscoids.

- Our extended follow-ups enabled investigation of the emergence of robust hematopoietic cells in independent experiments, showing a high degree of reproducibility.
- We also show that amnion-like tissue specification in iDiscoid mimics the expected outcome after inhibition of BMP4 signaling, further highlighting utility of the model.
- We examined CER1/TBXT dynamics comprehensively, which provided additional insight on a dynamic that we could also confirmed in five different *in vivo* datasets.

Collectively we show a) the model mimics bona fide developmental events b) establishes a physiologically relevant size c) can shed light on new information or confirm old findings d) after optimization shows low artifact that is now captured in the new data and confided with data *in vivo*.

Additionally, reproducible and efficient formation of observed phenotypes occurs in iDiscoids, which we have elaborated on in the revised text. These include efficient and reproducible formation of posterior axis ($46.4 \pm 9.0\%$), anterior hypoblast domain ($60.4 \pm 5.3\%$), amnion development ($>98\%$) as well as hematopoietic features (please see below for additional discussion relating to hematopoietic features). We report quantitative metrics for each instance.

Most strikingly, the anterior visceral endoderm (CER1 positive regions) arises adjacent to gastrulating (TBXT and MIXL1 positive regions), which is at odds with chicken, mouse, rabbit, pig, and cynomolgus development.

Response:

We appreciate the point raised by our reviewer and agree that expression of CER1 near TBXT⁺ region at first sounds rather surprising. Our initial observation in this respect did not have the quantitative analysis necessary to back up this finding. Therefore, in this version of manuscript, we have comprehensively profiled iDiscoid anterior-posterior (A/P) axis formation in large numbers and compared with *in vivo* data from published embryos in human, cynomolgus monkey and mouse. In our new analysis we do see the classical form of TBXT/ CER1 positioning with a polar configuration which is expected from several past studies. However, we also report a subset that doesn't follow this pattern and compared it to available *in vivo* datasets. Please see below.

Anterior-posterior (A/P) axis in iDiscoids:

Following analysis of 169 of iDiscoids assessed, we observed that $60.4 \pm 5.3\%$ (102 of 169) of iDiscoids with polar configuration in CER1 expressing cells possessed a TBXT-expressing pole (**Fig. 3F**). $42.2 \pm 3.2\%$ (43) of these iDiscoids exhibited anti-polar configuration, where the TBXT⁺ pole occupied the opposite radial region as the CER1⁺ domain in line with past data in mouse studies (**Fig. 3F-H**). In a second set of iDiscoids, we observed high expression of CER1 in proximity to the TBXT⁺ domain (**Supplementary Fig. 10A**). Surprisingly, we also noted cells in posterior domain that expressed both TBXT and CER1 proteins (**Supplementary Fig. 10A-B**). These iDiscoids showed CER1⁺ and TBXT⁺ poles that shared the same radial region (syn-polar). A closer look at scRNA-seq analysis also showed a population of cells in our posterior-like cell cluster that expressed CER1 as well as NODAL (**Supplementary Fig. 10C**).

In vivo datasets: Subsequently we examined the recent scRNA-seq data from human or non-human primates. In marmoset embryo datasets (*Bergmann et al. Nature 2022*), we noted non-polar CER1 at CS5, followed by very slight polarization away from the posterior at CS6 (a subtle anti-polar format) (**Supplementary Fig. 11A**). In human E16-19 (CS6) (*Tyser et al. Nature 2021*), we observed higher CER1 transcript expression in the same caudal embryonic region as the primitive streak (**Supplementary Fig. 11B**). Lastly, pseudospacial assessment of the E18-20 cynomolgus embryo (**Cui**

et al 2022, Cell Reports) shows *CER1* and *TBXT* expression overlapping within the lower region of embryonic disc (hypoblast) and embryonic disc compartments (a syn-polar format) (**Supplementary Fig. 11C**). Interestingly, all three datasets show a subset of cells or regions that display co-expression of *CER1* and *TBXT* in the posterior domain. When we examined mouse scRNA-seq datasets (*Pijuan-Sala et al Nature 2018*, *Mitnenzweig et al Cell 2021*, *Peng et al. Dev Cell 2016*), we could confirm *Cer1* gene expression presents at the anterior primitive streak (APS) and is induced subsequent to *T* expression (E6.5-E7.5) (**Supplementary Fig. 11D-F**).

Conclusion: Taken together, our analyses of the available *in vivo* data suggests that the observed expression and positioning of *TBXT*⁺ and *CER1*⁺ cells in iDiscooids is more than an *in vitro* artifact and can present two stages representing development and progression of posterior domain. Anti-polar positioning of *CER1* and *TBXT* domains reflect positioning of anterior hypoblast-like and posterior domains during early primitive streak formation around E14; syn-polar positioning and co-expression of *CER1* and *TBXT* is reflective of a more advanced stages post-E14 where *CER1* expression acts either to prevent spreading of posterior domain or represent early mesendoderm commitment derived from a primitive streak. In the current version of the manuscript, our added data clearly shows *in vitro-in vivo* correlation that was revealed only via using iDiscooids with subsequent assessment of *in vivo* datasets.

(Additional information on human CER1: It is the case that data on human *CER1* is limited, and studies suggest species-specific differences. The sequence of Cerberus protein differs significantly among species (only 69% identity between mice and humans) (Aykul S et al 2015, PLoS ONE). The large variations in amino acid sequence may result in unique binding specificities and different functions. Aykul S et al also assayed human Cerberus binding activity and revealed that it binds strongly and inhibits Nodal signaling. Among TGFβ family ligands, BMP-2 and GDF-11 could bind loosely to Cerberus but human Cerberus did not bind any other tested TGFβ family ligand with consequential affinity, including BMP-4 and Activin A (in contrast to mouse protein, see Fig 3F in the paper by Aykul S et al). Hence CER1 function in human in respect to A-P axis formation may not fully mirror past data in mouse models. More studies in human are required but this has been hampered by lack of access to human embryo or stem cell derived models with a *CER1*⁺ population.).

Equally the authors do not investigate alternative AVE markers, e.g., LHX1, LEFTY1, apart from HHEX.

Response:

Thank you for this comment and we have included this data in Fig 3. Please see a representative figure in Reviewer Figure R for protein staining in a z-slice, and Reviewer Figure Q for single cell analysis.

Another major issue is that in iDiscooids epiblast lumenogenesis occurs via multiple lumen, which has not been overserved in either rhesus or human preimplantation embryos of the Carnegie collection.

Response:

We would like to thank R2 for this comment which helped us to improve our iDiscooid development strategies while gaining insight to parameters during cavity formation. In this version of the manuscript, we have performed extensive studies on how to develop iDiscooids and provide our key findings:

We developed a quantitative pipeline to examine the iDiscooid morphology. Our initial effort had generated a variety of sizes of iDiscooids, and when analyzed in detail, it showed that the number of lumens formed is correlated with disc area (Supplementary Fig. 4A) (also in line with data from Orietti LC et al, Stem Cell Reports 2021).

To develop control over the size, number, circularity, and lumen number of WT clusters, we have performed high throughput optimization of iDiscooid creation. Our efforts reveal that modulation of the ratio and number of initial cells seeded provide a way to control the size and number of WT clusters within iDiscooids and the number of lumens they form. **As a result, by choosing appropriate conditions, we can achieve iDiscooids that have a physiologically relevant size range and lumen number.** When we study these cultures, we subsequently show that our predication can match the outcome and we can improve the frequency of structure and single lumen formation (33.2% → 70.9%). We had initially seen multiple lumen formation in 63% of iDiscooids (207 out of 331). Subsequent to optimization of the iDiscooid seeding ratio, among 79 WT discs, 58 had a single lumen and had areas in a range of 10367 to 43715 μm^2 , equivalent approximately to the lateral area of the CS5b – CS6 human bilaminar disc (as recorded in Carnegie #8004, #7700 in Hertig and Rock, 1949 and Carnegie #7801 in Heuser et al. 1945).

The role of CXCR4 in migration might be interesting, but it is unclear how this relates to the embryo in vivo.

Response:

We agree that data on CXCR4 in controlling bilaminar disc morphology or integrity doesn't exist. We have decided to remove this part of the data and allocate a more in-depth study in future with comparable data from *in vivo* studies.

The model may provide a useful platform to dissect the cross-talk between embryonic and extraembryonic lineages. It would be great if it were better defined in terms of developmental time and signalling requirements.

Response:

We thank R2 for this assessment. We agree that iDiscooids will be a useful platform for the scientific community to decode a variety of cell-cell / tissue-tissue multilineage communications within a human embryo-like context. It enables study of communication signals within embryonic-like lineages (amnion, epiblast, and posterior domain), yolk sac communication with epiblast and posterior axis, or further insight on emergence of hematopoietic cells during this window of human development. As an example, we show BMP4 signaling and its target in Supplementary Fig 8B. Additionally, in different parts of the paper we show data to demonstrate the utility of iDiscooids to study signaling events and parameters relevant for development of cell fates (e.g., Supplementary Fig. 9 BMP4 for PS domain, Fig. 4F-G high GATA6 cells in development of CD43⁺ hematopoietic foci) or morphological features (i.e. Supplementary Fig 4A-E initial size of cell cluster to initiate cavity formation).

Our additional single cell analysis allowed careful temporal staging of iDiscoid development in respect to human and NHP development. In the current manuscript, we have compared iDiscoids with human single cell datasets from Xiang et al. 2019 spanning embryos from E6 to E14 of development, and Tyser et al. 2021 (E16-19 human embryo) as well as cynomolgus E16 NHP dataset from Ma et al 2019 that shows cynomolgus monkey blastocysts beyond early gastrulation. iDiscoids development captures ~E12-E16 human embryogenesis. Comparison to human embryos from Xiang et al. shows that by 36 hours, iGATA6 cells in iDiscoids present the strongest similarity with E12 hypoblast lineage, and this similarity strengthens through day 5 of culture (Supplementary Fig. 6B, Xiang_Hypoblast_D12). Through our analysis we have shown stepwise development and maturation of

Reviewer Figure S: iDiscoid morphology is not acquired under different media conditions

- (A) Day 5 iDiscoid morphology after switching the feeding media to modified IVC1 (mIVC1) media at day 2 of culture. While some migration of iGATA6 over the WT structures is observed the acquisition of a contiguous iGATA6 membrane necessary for lumen formation and subsequent patterning steps relevant to human embryogenesis are not observed.
- (B) Day 5 iDiscoid morphology after switching the feeding media to Essential 6 (E6) media at day 2 of culture. Both iGATA6 and WT morphology are substantially altered and bilaminar disc-like morphology is not observed.
- (C) Day 5 iDiscoid morphology after switching the feeding media to IMDM media at day 2 of culture. WT discs form, but GATA6 migration is not observed. Scale bars = 500 μ m

embryonic populations. For instance, the amnion layer in iDiscoids at day 4 (labeled as iDisc_D4_W3) becomes highly similar to (E-AM) and (L-AM1) amnions from Ma H et al in Science 2019. For more information, please see Supplementary Fig. 6.

The sorting behaviour of iGATA6 and WT cells would be expected to happen in preimplantation embryos – note that this happens before and not during or after implantation (Kuijk, Development, 2012). – so why has this not been done with naïve hESCs rather than primed?

Response:

We appreciate this comment and agree that the sorting behavior seen in early stages of iDiscoid development is a feature that would be expected during preimplantation development. Our scRNA-seq analysis demonstrates that the cell types observed during iDiscoid development primarily approximate post-implantation (D12 and later). As a result, we have focused our manuscript on post-implantation relevant events. Accordingly, we have removed discussion of the sorting of iGATA6 and WT cells from the manuscript as it applies to preimplantation events. With the focus on modeling this window of human development (post-implantation phase), we decided to not use naïve hESCs.

How does mTESR affect the sorting and what are the signalling requirements for this behaviour? How do seeding ratios affect results?

Response:

We initially had performed assessments of iDiscoïd formation in non-mTeSR media. Immediate application of other media (e.g., E6, IMDM) from day 0 resulted in substantial cell death, altered cell morphology, and failure of self-organization. In light of this, we started with mTeSR and tested different media types beginning at day 2 to enable analysis of their effects on the formation and development of the bilaminar disc state of the system. We found that in all of the other media formations tested, the bilaminar disc morphology fails to form, and the morphology of WT structures becomes significantly more irregular (please see a subset in Reviewer Figure S). This was also in line with a past study where authors favored mTeSR media for organized lumen formation (Fig 5g in Shahbazi M, Nature 2017 Dec 14). As such, based on our pilot studies, we decided to use mTeSR across the first 5 days of the study. mTeSR has around 18 components added to DMEM/F12. It contains BSA, insulin, bFGF, TGF- β , and GABA among others (Ludwig TE et al. Nat Methods 2006). Its suitability to both support a pluripotent state (Ludwig TE et al. Nat Methods 2006) and to maintain progenitor cell populations (Ng AHM et al, Nat Biotech 2021) were shown before, which were additional reasons we chose to use this media. However, as we have focused on peri/post-implantation stages of development, we respectfully believe in depth focus on key signaling paths in WT/GATA6 cell sorting behavior and how each component in mTeSR influences this event would be beyond the scope of the study. As such, we decided to focus our attention on those areas that can support the utility of iDiscoïds in studying post-implantation events in human.

We have substantially assayed seeding densities (in Supplementary Fig 4) in order to control the size, number, and circularity of WT clusters (as discussed in above response). We find that all three metrics are affected by seeding ratios, and at optimized ratios physiologically relevant disc sizes can be achieved.

Different iGATA6 level seem to induce a variety of embryo-related lineages. This is potentially interesting, but the resulting cell types have not been further characterized (e.g. through FACS sorting, 3D culture etc). The erythroid lineage differentiation is also interesting and partially supported, however it is unclear what insights the iDiscoïd would provide as they lack the proper architecture and efficiency to study this process. The data also suggests substantial levels of spontaneous differentiation, in particular as blood islands occur specifically at the interface of extraembryonic mesoderm and yolk sac endoderm (but not between ExMes and other lineages). The monolayer approach prevents this architecture from forming.

Response:

We thank R2 for her/his comment. In this version, we have provided substantially more experiments that help in both optimizing and characterizing the cell types and observations made. While further analysis of each cell type is possible, we believe our current new data is sufficient to address the concerns raised by R2, as we discuss below and in the revised text. We also would like to note that in some cases, sorting of the cells and investigation outside of their integrated environment cannot easily yield the same behavior as is observed within their niche. Hence, we have aimed to profile the system as it stands and have used cell isolation and separate validation strategies to gain additional insights when possible. In this version, we show characterization of the amnion layer, we show profiling of cells at early and late stages of iDiscoïd development using scRNA-seq (See Fig. 1, Supplementary Fig 5 and 6 and Text on line 136 to 185), optimization of lumenogenesis, hemogenic endothelial-like and blood island-like structures were profiled, and CER1/TBXT dynamics were further characterized and compared with in vivo embryo tissues.

Blood islands/ hematopoietic features: efficiency, proper architecture and positions: We agree that initial data on generation of endothelial-like cells on day 5 was limited, and could give the impression

that the system could not follow the same architecture as *in vivo* blood islands or demonstrate the efficiency needed for an effective *in vitro* model. We now show more complete characterization of the emergence of the endothelial cells at these interfaces, but more importantly, our extended follow up addresses the above concerns and clearly show robust generation of blood island-like structures with close proximity to *in vivo* (assessed quantitatively) (See data on Fig. 4I and Supplementary Fig. 13D). Notably, we also show the ability to augment this event via supplementing high GATA6 expressing cells. We demonstrate that this modulation increases the resulting CD43⁺ population. Hence, by changing the initial composition of the cultures, the outcome of the cultures can be affected in a predictable, reproducible way; however, we do not need to exert continuous control over the events within the system and initial genetic circuit to enable self-organizing behavior with a design principle. Additionally, expansion of endothelial like cells from 1.2±0.46% of the culture area on day 5 to covering 14.7±6.0% of the culture area at day 12 further support cellular quantity and composition that enables studying hematopoietic or related events (Supplementary Fig. 13F).

In the current manuscript, we demonstrate that, while percentages of TAL1⁺/CD34⁺ hemogenic endothelial-like cells are low at day 5 of iDiscoid culture, this cell type substantially expands, via both image analysis, showing the expansion of these cells from covering 1.2±0.46% of the culture area on day 5 to covering 14.7±6.0% of the culture area at day 12 (Supplementary Fig. 13F), and in flow cytometry analysis, where we demonstrate that CD34⁺ cells expand from 2.38% to 12.4% of total cells in iDiscoid culture between D5 and D12 (Supplementary Fig. 13). We demonstrate that the architecture of blood island-like areas specifies in a consistent way, measured across multiple experiments (Fig. 4I). We demonstrate that modulation of the initial iGATA6 populations through supplementation of GATA6-hi cells to the starting culture increases the resulting CD43⁺ population that emerges. Taken together, we are able to show both an efficient specification and expansion of these cell types, and reproducible outcomes related to the structures and cell fates they take on by day 12.

Monolayer approach:

We have added additional rationale to the text to demonstrate how the experience we had in our past 3D-focused efforts resulted in taking this approach for modeling the post-implantation window of human development (data is provided in Supplementary Fig. 1C-D):

When iGATA6 and WT hiPSCs were co-assembled in 3D, EGFP⁺ extraembryonic endoderm cells segregated to the outer layer, similar to what occurs during implantation period *in vivo*. However, the overall organization of these aggregates did not recapitulate or progress to the structural format of bilaminar disc and yolk sac tissue. Our initial effort to assemble iGATA6 and WT hiPSCs in 3D yielded aggregates in which the entire EGFP⁺ extraembryonic endoderm is continuously in contact with wild type epiblast-like cells and cannot provide structural format relevant to generation of both bilaminar disc with yolk sac tissue away from the epiblast cells. Importantly, in these 3D aggregates, we found that WT cells could not consistently induce symmetry breaking to amnion-like and epiblast-like domains (**Supplementary Fig. 1**). Moreover, *in vitro* reattachment of both synthetic and true human blastocysts to a dish have shown organizational instability that results in tissue organization that does not reflect the developmental stages approaching gastrulation (Kagawa et al 2022, Nature). Hence, mimicking stable co-development of embryonic and extraembryonic cell types through post-implantation embryonic stages from an initial 3D state is technically difficult to control. To address these challenges, we aimed to develop an alternative strategy to generate structures off the plate via igniting 2D-to-3D inherent self-organization of hiPSCs. The iGATA6-hiPSCs were mixed with wild-type hiPSCs (WT), and the mixed population of cells were seeded onto standard culture plates in 2D at a defined ratio and cell density. As described in the text, through geometric confinement we see clear self-organization of amniotic cavity, symmetry breaking, and an organized 3D layer of yolk sac tissue.

Inhibition of SB43 seems to abolish the entire embryonic compartment. It is unclear how this may relate to amnion formation as it appears to act upstream.

Response:

*[Redacted
Text]*

Referee #3 (Remarks to the Author):

I was asked to conduct an ethics review of the manuscript “Modelling Human Post-Implantation Development via Extra-Embryonic Niche Engineering.” In this paper, the researcher developed an embryoid (called an iDiscoïd) from induced pluripotent stem cell lines. No embryos were used (directly or to develop an embryonic stem cell line), therefore no IRB approval was obtained nor warranted. The iDiscoïd is non-integrated stem cell-based embryo model, therefore in the 2021 guidelines it does not require specialized review (just reporting to a SCRO). However, this research was likely conducted over several months which occurred prior to the release of the 2021 ISSCR guidelines (May 26, 2021). The shift between the two guidelines allowed for longer culturing of the iDiscoïd. It is questionable whether this research occurred before or after May 2021 (when a 14-day limit for embryo model research was lifted), but I will give the authors the benefit of the doubt, especially since the researchers obtain approval from the work by a SCRO committee. However, I recommend that both guidelines (2016 and 2021) be referenced in the ethics session with a note that the authors followed both, unless the SCRO approval occurred after May 26, 2021. Other than this minor revision, I have no issues with how the research was conducted.

Response:

We thank our reviewer for her/his comment and have cited both guidelines.

Referee #4 (R4) (Remarks to the Author):

In this study, Hislop and colleagues generated an in vitro model of the human post-implantation embryo using a mixed population of wildtype and inducible GATA6 hiPSCs, termed “iDiscoïds”. As well as having a bilaminar disc morphology similar to the human embryo, the authors observed several extraembryonic populations including some potential for haematopoietic differentiation, and initial evidence of an antero-posterior axis with polarised TBXT expression. Current in vitro models of human embryonic development don’t yet model the early post-implantation stage and most lack evidence of early gastrulation, so this is an appreciable advance over existing techniques.

Overall, this is a well-written manuscript that is likely to be of interest across the field of human developmental biology. However, there are several concerns that should be addressed before publication, that are highlighted below:

Major Concerns

1. Considering that formation of the iDiscoid bilaminar disk is a dynamic process, and that the authors observe transient cell states during lumenogenesis and TBXT/MIXL1 symmetry breaking, scRNAseq data from multiple timepoints would, in my opinion, be necessary to support their full claims. This is particularly important since the authors state that the TBXT⁺ cells are evident at Day 4 and decrease by Day 5, so are therefore not adequately captured by their transcriptomic analysis. Since this observation is the primary advance on existing embryo-like systems, this would be an important stage to characterise and assess.

Response:

We would like to thank R4 for this important comment. We have performed scRNA-seq from several time points across the early development of iDiscoid prior to day 5 in order to capture the emergence of cell lineages over this period. This has allowed us to make several important findings; we are able to track the similarity of each population detected in iDiscoids to its *ex vivo* human embryo counterparts. We have managed to capture and annotate posterior-like cells that initially appear at day 3 and are most prominent at day 4 (please see Fig. 3 and Supplementary Fig. 6). In addition, we detected the emergence of the anterior-like population at day 2, prior to the specification of TBXT⁺ posterior-like cells (please see Fig. 3A).

2. The authors claim a posterior-localised CER1/TBXT polarisation, as outlined in Extended Figure 10. This is a highly surprising observation, since such polarisation has not been observed in non-human *in vivo* embryos, nor in current *in vitro* human systems. As such, the authors need to provide additional evidence to support this hypothesis.

Response:

We appreciate the point raised by our reviewer and agree that expression of CER1 near the TBXT⁺ region at first rather sounds surprising. Our initial observation in this respect did not have the throughput necessary to support our conclusion. Therefore, in this version of manuscript, we have comprehensively profiled iDiscoids anterior-posterior (A/P) axis formation in large numbers and compared with *in vivo* data from published embryos in human, cynomolgus monkey, and mouse. In our new analysis we do see the classical form of TBXT/ CER1 positioning with a polar configuration which is expected from several past studies. However, we also report a subset that doesn't follow this pattern and compared to *in vivo* dataset. Please see below.

Anterior-posterior (A/P) axis in iDiscoids: Following analysis of 169 of iDiscoids assessed, we observed that 60.4±5.3% (102 of 169) of iDiscoids with polar configuration in CER1 expressing cells possessed a TBXT-expressing pole (**Fig. 3F**). 42.2±3.2% (43) of these iDiscoids exhibited anti-polar configuration, where the TBXT⁺ pole occupied the opposite radial region as the CER1⁺ domain in line with past data in mouse studies (**Fig. 3F-H**). In a second set of iDiscoids, we observed high expression of CER1 in proximity to the TBXT⁺ domain as well as within a subset of cells in posterior domains (**Supplementary Fig. 10A**). Surprisingly, we also noted some cells in the posterior domain that expressed both TBXT and CER1 proteins (**Supplementary Fig. 10A-B**). These iDiscoids showed CER1⁺ and TBXT⁺ poles that shared the same radial region (syn-polar). A closer look at the scRNA-seq analysis also confirmed a population of cells in our posterior-like cell cluster that expressed CER1 (**Supplementary Fig. 10C**).

In vivo datasets: Subsequently we examined the recent scRNA-seq data from human or non-human primates. In marmoset embryo datasets (*Bergmann et al. Nature 2022*), we noted non-polar CER1 at

CS5, followed by very slight polarization away from the posterior at CS6 (a subtle anti-polar format) (**Supplementary Fig. 11A**). In human E16-19 (CS6) (Tyser et al. Nature 2021), we observed higher CER1 transcript expression in the same caudal embryonic region as the primitive streak (**Supplementary Fig. 11B**). Lastly, pseudospacial assessment of the E18-20 cynomolgus embryo (Cui et al 2022, Cell Reports) shows CER1 and TBXT expression overlapping within the lower region of embryonic disc (hypoblast) and embryonic disc compartments (a syn-polar format) (**Supplementary Fig. 11C**). Interestingly, all three datasets show a subset of cells or regions that display co-expression of CER1 and TBXT in the posterior domain. When we examined mouse scRNA-seq datasets (Pijuan-Sala et al Nature 2018, Mittnenzweig et al Cell 2021, Peng et al. Dev Cell 2016), we could confirm Cer1 gene expression presents at the anterior area of primitive streak (APS) and is induced subsequent to T expression (E6.5-E7.5) (**Supplementary Fig. 11D-F**).

Conclusion: Taken together, our analyses of the available *in vivo* data suggests that the observed expression and positioning of TBXT⁺ and CER1⁺ cells in iDiscooids is more than an *in vitro* artifact and can present two stages representing development and progression of posterior domain. Anti-polar positioning of CER1 and TBXT domains reflect positioning of anterior hypoblast-like and posterior domains during early primitive streak formation around E14; syn-polar positioning and co-expression of CER1 and TBXT is reflective of a more advanced stages post-E14 where CER1 expression acts either to prevent spreading of posterior domain or represent early mesendoderm commitment derived from a primitive streak. In the current version of the manuscript, our added data clearly shows *in vitro-in vivo* correlation that was revealed only via using iDiscooids with subsequent assessment of *in vivo* datasets.

(Additional information on human CER1: It is the case that data on human CER1 is limited, and studies suggest species-specific differences. The sequence of Cerberus protein differs significantly among species (only 69% identity between mice and humans) (Aykul S et al 2015, PLoS ONE). The large variations in amino acid sequence may result in unique binding specificities and different functions. Aykul S et al also assayed human Cerberus binding activity and revealed that it binds strongly and inhibits Nodal signaling. Among TGFβ family ligands, BMP-2 and GDF-11 could bind loosely to Cerberus but human Cerberus did not bind any other tested TGFβ family ligand with consequential affinity, including BMP-4 and Activin A (in contrast to mouse protein, see Fig 3F in the paper by Aykul S et al). Hence CER1 function in human in respect to A/P axis formation may not fully mirror past data in mouse models. More studies in human are required but this has been hampered by lack of access to human embryo or stem cell-derived models with a CER1⁺ population.)

Firstly, statistical evidence should be provided as to the frequency of this observation and degree of spatial bias, for instance using an angular distribution plot. In addition, evidence should be provided to support their signalling hypothesis for this localisation (in Extended Figure 10). For instance, their BMP4/NODAL inhibitor experiments are not replicated with TBXT/MIXL1 stainings, which could potentially support/disprove such a mechanism.

Response:

We have provided angular distribution plots. In this version of our revised manuscript, we do not follow the same signaling hypothesis and, as explained above our in-depth analyses, have provided a new understanding of the iDiscooid A/P axis development.

[Redacted
Text]

[Redacted
Text

3. The authors suggest that there is a temporal progression in iDiscoïd morphology from 'Early' to 'Intermediate' to 'Late' stages (eg Figure 3a), and while the overall distribution of these observed morphologies changes over the timecourse (Day 3-5, Figure 3d) they do not ever show the same iDiscoïd transforming over time from one to the other. As such, this claim is not adequately supported by their data.

Response:

We agree that we had not shown the temporal progression of a single iDiscoïd through the stages shown in the previous version of the manuscript due to technical challenges. We have added data showing the growth of a single lumen from a rosette over time in a single iDiscoïd (Please see Fig. 1C, Supplementary Video 3).

Our initial staging of the iDiscoïds relied on events associated with the cluster with sizes that facilitate multi-lumen initiation (early to intermediate) and combination into a single final cavity (late). However, in order to address concerns raised by other reviewers related to the relevance of multi-lumen development within iDiscoïds to the human embryo, we performed optimization to develop control over the size, number, and circularity of WT clusters via high-throughput studies and optimization of the method and ratio of iDiscoïd creation. As a result, by choosing appropriate conditions, we have achieved iDiscoïds in a physiologically relevant size range that more consistently resulted in forming a single lumen. Due to the concerns raised regarding the physiological relevance to the embryo, we have used the optimized version with no intermediate stages that rely on multi-lumen development for achieving a single amniotic cavity-like state. Subsequently the data on multi-lumen iDiscoïds were removed.

4. More generally, given the high throughput potential of this system, more extensive image quantifications are needed throughout the manuscript. What is the variability in area/size/morphology of each iDiscoïd throughout an entire well? How does this affect the patterning observed – does scaling occur? How does it affect the polarisation of CER1/TBXT?

Response:

We thank our reviewer for this comment. We agree that the high-throughput nature of the system enables such examinations. In this version of manuscript, we have provided more extensive quantitative analysis of the cell fates and structures in the culture. We report variability in area/size and correlation with lumen formation and optimized cultures accordingly.

To that end, we characterized the features of WT clusters within iDiscoid culture, including disc area, disc circularity, and number of lumens that occurred in each individual disc. We also quantified the number of lumens observed within the disc and, following optimization of the iDiscoid ratio, showed a substantial increase in the consistency of single lumen iDiscoids. *See Supplementary Fig. 4*

We have seen amnion development very consistently within iDiscoids, with a high percentage of WT clusters express the amnion marker ISL1 beyond day 3 of development. iDiscoids analyzed in Figure 2I showing that all but one of the observed WT clusters (n=87) have ISL1 expression above the expression levels in the negative control, an observation that was confirmed in independent experiments. *See Figure 2I*

We confirmed the reproducibility of posterior-like domain emergence within iDiscoid at day 4. Development of a polarized posterior domain was observed in a total of 488 iDiscoids, verified across independent experiments. We also assayed CER1 polarity (Fig. 3F-H). We investigated 5 total areas expressing TAL1 across 3 independent samples and verified the consistent arrangement of endothelial cells underneath a layer of iGATA6 cells at day 5. We investigated 9 total areas across 3 independent iDiscoid samples to verify the regionalized localization of yolk sac endoderm, yolk sac mesoderm, and endothelial-like cells in a blood island-like orientation. *See Figure 4I, Supplementary Figure 13D*

5. The authors state that the WT and iGATA6 cells undergo cell sorting, leading to their distinct domains. Although the supplementary video shows the dynamics of this process to some degree, it is difficult to assess whether cell migration or differential proliferation might be mediating some of this patterning. Since this is a crucial process in early embryonic development, the authors should clarify if the cells are truly sorting, perhaps by cell tracking.

Response:

We have adjusted the text to make sure we do not assume migration as prerequisite for the sorting. We agree with 4that it is important to know the contribution of proliferation versus migration in the context of initial sorting of GATA6 and WT cells pertinent to early developmental events during pre-implantation. Several past studies have examined the segregation of epiblast and primitive endoderm cells *in vivo* (i.e., Yanagida et al., 2022, Cell 185, 777–793 or Gabarek et al. Development 2012). Now we show that soon after induction of GATA6, the cells show a post-implantation stage. As such, iDiscoids (days 0 to 4) most closely capture human E12-16 (Supplementary Fig 6). Hence, in the current version of our manuscript, we have focused on further assessment and characterization of developmental events that are related to this period of development. While we also agree that understanding dynamic of cell sorting perhaps in tandem with a computational model (and presumably via naïve ES cells) is interesting and important, we respectfully believe such further characterization can be a topic for our future efforts with a focus on early stages of embryonic development.

6. The authors stress the transient nature of their structures, including the short-term expression of TBXT on Day 4, and then stop describing their structures after Day 5. What happens after this point, and why is the experiment stopped here? If cell death and/or patterning disruption is occurring, the authors should state and comment on this.

Response:

We thank R4 for her/his valuable comments. The decrease in TBXT may be a result of fate differentiation of TBXT⁺ cells to other cell types (i.e., different mesodermal lineages). Our initial analysis (staining for different markers such as HAND1, or scRNA-seq analysis) did not highlight an increase in different mesodermal markers or a new cell type detectible as coming from the WT cluster. In addition, we have observed in a subset of iDiscoid culture continued beyond day 5 that mTeSR may facilitate progenitor proliferation, enlargement and disruption of the WT cavity(i.e., leakage) that may explain the

decrease in TBXT due to this event. However, the exact underlying mechanism and biological relevance remain to be understood.

Having said this, in the new version of manuscript, we decided to follow the morphogenesis process and facilitate differentiation of progenitors using a basal media after day 5 (IMDM). We report in our revised version of manuscript that the WT cell clusters appear to show ectodermal fates. However, our further follow up has enabled us to observe progression of the yolk sac development with emergence of hematopoietic characteristics and blood island-like structures. (Please see Fig. 4 and Supplementary Fig. 13-16 for more information). We decided not to focus on late WT structures (ectodermal) as we did not detect a notable observation which would need future optimization, and it warrants its own separate studies.

7. It is surprising that in the scRNAseq dataset (e.g. Figure 1d), the epiblast/WT cluster is a single population, whereas Figure 3E and Figure 4 at least two clearly distinct populations are observed by immunofluorescence. Why is this the case?

Response:

Our initial analysis of day 5 had a small population of WT cells and could not reach the necessary depth for detecting those two fates. We repeated the experiment and performed several scRNA-seq rounds at different stages of iDiscoid development, the results of which show clear formation of epiblast-like, amnion-like, and posterior axis-like cells in the putative WT population.

Minor Concerns

We appreciate the reviewer's suggestions for the improvement of the manuscript. We have noted the specific changes made under each of the minor comments below.

1. Labels to indicate the day that images were taken are needed throughout figures and/or in the legends.

Response:

We have added appropriate labels to the figures and within the legends to indicate the day shown in each of the images.

2. Extended data figure 4: UMAPs of iDiscoids and Human embryo need to be shown separately as well as overlaid. As it stands, it is incredibly difficult to observe the true crossover between the two populations.

Response:

In the current version of the manuscript, we have substantially increased the number of single cell datasets that we have presented, as well as the number of *in vivo* datasets that we are comparing our single cell clustering to. As such, due to the large quantity of data produced, for clarity of presentation and to direct focus to quantitative comparisons, we have decided to instead illustrate the results using hypergeometric statistical comparison between the single cell datasets and the *ex vivo* human and NHP datasets. We have elected to remove the initial overlaid figure from this version of manuscript.

3. Extended data figure 9C: Top left image is very blurry, which is concerning. A better-quality image is needed. Also, vertical slices should be shown for all markers including pSMAD1/5/8 and pSMAD2.

Response:

The blurriness in this top image was caused by the overlaid iGATA6 cells within this widefield image. We have replaced this image with a better-quality image of iDiscoid under inhibition conditions (please

see Fig. 2G-H). As advised, we have included vertical slices for each of the images shown related to pSMAD expression and signaling inhibition.

4. Figure 1A: the schematic suggests that the flat disc on D2-3 becomes a multi-layered structure on D4-5, but confocal images show only the top view. A cross-section should also be included here to support their schematic.

Response:

We have included cross-sectional views of a covered disc acquiring a lumen in our initial figure (now Fig 1B-C).

5. Figure 2B: It is unclear whether the region of interest includes WT cells, since the authors describe the localisation of spindle shaped, CD34/TAL1 expressing cells underneath an iGATA6 endoderm layer. The morphology of iGATA6 cells look like they are sitting on top of a disc shape which contains WT cells. This needs to be clarified by inclusion of a DNA stain or epiblast/pluripotency marker.

Response:

We appreciate that this part of the text may have been unclear. In brief, these spindle-shaped cells emerge from the iGATA6 layer and are positioned underneath the iGATA6 cells, away from the WT cluster (Figure 4C, Supplementary Figs 13B-E). We have revised the text to clarify that the spindle-shaped cells emerge within the iGATA6 layer and have provided a Hoechst stain for these images to show the absence of WT clusters underlying these structures.

6. Extended data figure 13C and E: it looks like there aren't any EGFP-GATA6 cells around the top of the bilaminar disc. This needs to be commented on/clarified.

Response:

We have adjusted the images shown (please refer to Supplementary Fig. 20).

7. Line 145 and Figure 2B: The authors state that clusters are CD34+/ERG+ implying that these cells are co-expressing these markers, however the figures do not show a co-stain. How have the authors validated this?

Response:

We would like to thank the reviewer for pointing this out. We have revised the text.

8. Extended data figure 2: how many clusters (ie replicates) were measured? What is the significant difference and variability between clusters in each of these expression levels? It is surprising that the peak of PDGFRa expression is less than one cell diameter from the GATA6 expression. More generally, I find the antibody stain for PDGFRa less than convincing.

Response:

This result was obtained via analysis of between 20-167 WT clusters per marker from iDiscoid cultures across 8 separate experimental replicates. We have reproduced this data in Reviewer Figure T including the requested standard error of these measurements.

With regards to the reviewer's point about PDGFR α staining being unconvincing, we appreciate that the staining for this marker may appear unclear. We have verified that this marker is expressed within iGATA6 cells in our scRNA-seq dataset and show PDGFR α negativity in WT cells in the figure referenced. However, in light of the reviewer's comment, we have substituted a separate set of immunostains for showing the endoderm identity of our iGATA6 cells in Supplementary Figure 2D. We had additionally shown grayscale images of each channel to make clearer the PDGFR α expression where we have shown it in the current manuscript (see Supplementary Figure 2A)

As mentioned in the previous comments (i.e., see comment #5 by R4), we have decided to focus on events that more closely correlate to post-implantation events. We took off our focus from this data and removed the previous Supplementary Figure 2 from the revised manuscript. We feel that a close examination of the sorting dynamics of iGATA6 cells prior to bilaminar disc formation and relation to PDGFR α are interesting but not fully aligned with our focus on presenting iDiscoids as a post-implantation embryo model system in this version of the manuscript.

9. Line 189: the authors state that iGATA6 cells deposit the laminin ECM. However, there is no quantification/example that shows negative laminin staining if iGATA6 cells aren't present. How have the authors concluded this?

Response:

We have included a stain showing a minimal amount of laminin deposition by WT cells alone when cultured in iDiscoid conditions. We have also added additional images quantifying the increase in laminin over iDiscoid culture (please see Supplementary Fig 3B and 3E).

10. Extended data figure 6E: Why is there no HOECHST staining on top of the iDiscoid, despite claiming that the laminin deposition occurs from iGATA6 cells?

Response:

The apparent lack of EGFP-GATA6 cells over the top of this image was a result of mechanical detachment of the layer during experimental processing. The iGATA6 layer in this image was present, but had lifted off several micrometers above the WT disc shown. For clarity, we removed this image and replaced it in Supplementary Fig. 3C with an image of a WT cluster with a visible iGATA6 layer, and we have included a cross section showing the presence of the marker.

11. Extended data figure 5A: the side projection does not show the same rounded discoid morphology. The authors should clarify or comment on this.

Response:

The cells shown in this image were additional markers for the endothelial-like cell type shown in the previous Fig. 2B-c (now shown in Fig. 4C and Supplementary Fig. 13B, 13C, 13E). As mentioned in Minor Comment 5 above, this cell type is not associated with the WT cluster and instead is just localized under the iGATA6 layer. We have adjusted the text to ensure that this is clear.

12. Extended data figure 2E in legend: currently states “specific subpopulation of cluster 4”, but the data shown is for cluster 5, so the legend needs to be corrected.

Response:

We have replaced the figure referenced with updated single cell data (please see Supplementary Fig. 5).

13. Extended data figure 6A: What do the (1), (2), (3) coloured dots on the right side of the figure mean?

Response:

These dots were originally intended to be a small schematic representing the coverage states of the WT clusters. We have removed this figure from the paper due to concerns raised by Reviewer 2 related to the relevance of CXCR4 data to *in vivo* human embryogenesis.

14. Extended data figure 6C: It looks like the cluster labels are incorrect and not in order. This needs to be corrected.

Response:

We have replaced the figure referenced with updated single cell data and altered the labeling scheme in the new data for clarity.

15. To generate their structures, the authors seem to use a ratio of 4:1 GATA6 to WT cells (‘Generation of iDiscoïd’ section) and 10:1 (‘10x Genomics Samples Preparation for Next-generation sequencing’ section). How does modulation of this ratio affect the outcome? Why are different ratios used? Additionally, in the absence of Dox induction, is this ratio maintained over iPSC culture time/passages?

Response:

We agree that this discrepancy in seeding ratios may have affected the outcome of iDiscoïds. As described, as a result of optimization of the iDiscoïd ratio, we find WT cluster size, morphology, and lumenogenesis are affected by altering the seeding ratio and density of iDiscoïds. We have adopted a new optimized seeding ratio and used this consistently for experiments performed during this revision, including for seeding of the new scRNAseq data. We show that over 15 passages, the ratio shifts slightly in favor of the iGATA6 population, with the WT population occupying roughly 15% less total area (13.5 mm² versus 11.5 mm² after 15 passages). However, the cell lines need not be kept combined, and the ratio can be seeded from proper mixing the independent cell lines to create the optimized iDiscoïd ratio.

16. Lines 512, 513: What is the rationale behind adding BMP4 or NODAL inhibitors on different days (day 3 vs day 2) and why were they not added on the same day for direct comparison?

Response:

We thank R3 for noting this. In the revised version of manuscript we have started the inhibition at day 2 of culture and removed any inconsistent data. We have chosen to remove NODAL data from the manuscript due to concerns raised by Reviewers 1 and 2.

17. Lines 484, 485, 498: the authors should clarify if 10 mM Y-27632 was used, or the standard 10 µM concentration.

Response:

We apologize for the typo that has occurred during formatting the font in the manuscript. We have adjusted this error to reflect the proper value ($10 \mu\text{M}$).

18. Lines 499 and 545: the authors describe considerably different concentrations of Doxycycline used for generation of the iDiscooids. Shouldn't the same concentration of Doxycycline be used throughout the study?

Response:

We apologize for the typo. We have corrected the text. The final concentration of Doxycycline in use is $1 \mu\text{g/ml}$.

19. On line 56, the authors state that current in vitro models of human embryogenesis have 'low efficiency and throughput [and] limited scalability'. I don't think this is a fair assessment, as most human embryo-like models currently described use methods such as microwells and other techniques that have fairly high throughput.

Response:

We thank our reviewer for bringing up this point. We agree that this would not be applicable to all the developed models. We have adjusted this sentence.

20. The authors should cite <https://doi.org/10.1038/s41467-017-00236-w> (the PASE model system based on human pluripotent stem cells, which contains an amniotic sac), and inducible GATA6 mouse studies (<https://doi.org/10.1016/j.bpj.2018.11.011>) and (<https://doi.org/10.1242/dev.127530>) on which this work appears to be built.

Response:

We thank our reviewer for this feedback. We have included and cited these papers as citations 17, 18, and 19.

Reviewer Reports on the First Revision:

Referees' comments:

Referee #1 (Remarks to the Author):

In their revision, the authors have gone to considerable lengths to address a number of critical issues raised by all referees in the first round of review. The authors provide additional data and analyses that make the characterization of this embryo model and its cellular components far more convincing than that provided in the original description. The new findings do much to strengthen the case that i-Discoids capture many important cell types and morphogenetic events in post-implantation human development in a manner that shows strong fidelity to available benchmark datasets. The study in its present form is a significant contribution to the literature in this fast moving field and it is timely. It will be of great interest to compare these findings to similar studies likely to emerge in the near future.

Referee #4 (Remarks to the Author):

- The authors should be explicit in their abstract, and also in their title, that this model does not include trophoctoderm lineages, and so is not a complete human embryo model.
- I agree with Reviewer 1 that the claim (in the abstract and elsewhere) that an iGata6 system is a 'synthetic gene circuit' is overstated. The authors should remove this statement.
- The authors should show their data from 3D mixing of the populations, and not just a schematic (EF1) so that readers can evaluate for themselves the 3D condition.
- The scRNA-seq data seems to match reasonably highly to DE as well as PrE. While this is not necessarily surprising given their very highly similar transcriptomic states, are the authors really sure that their iGata6 population is definitely hypoblast and not a variant of DE? This seems to me to be a critical point (on which the whole argument rests) so needs more attention in the text.
- The quantifications shown now are much improved from before. But I still don't have a quantitative sense of how common it is that individual lumens have an amnion populations.
- I don't understand why the 'anterior-like' hypoblast populations form 2 distinct clusters on the UMAP at different days (ED7B and Fig 3A) – doesn't this imply that the population changes dramatically over this time period (if it is even the same population at all)? I would be more convinced if you could find out what genes are differentially expressed at these 2 times/populations, and show spatially that they do both correspond to (the same population) of anterior hypoblast.
- I think it's too much of a stretch to say that the marmoset data (ED11A) is 'subtly anti-polar-like'. Sure, the data don't support a distinct polar bias, but this could be due to sampling limitations rather than a completely different biological mechanism. The same is true for the Cynomolgus monkey data

too (ED11C) where there isn't even an apparent posterior bias for TBXT, which we know is definitely posterior-biased in vivo.

- How does the observation that some cells co-express both CER1 and TBXT (ED10) affect the quantification of polarity? It could be that some of these double-positive cells are skewing the data towards 'syn-polar'?

- Noggin treatment (ED9) seems to be having an effect on the polarisation of both TBXT and CER1, while the text states that only TBXT is affected. The authors should either show quantification as proof of this, or amend their statement.

Minor points:

- I still think the authors need to show DAPI/Hoescht only channels for all of their confocal images. Otherwise, it's very hard to tell where negative cells are. Also, they need to ensure that their arrowheads are completely consistent across panels (eg Fig3C I'm fairly sure the arrow is higher in the LHX1 panel than in the CER1 panel).

- The categories in Fig3E should be renamed according to their example images: 'negative, polar, non-polar' makes much more sense than 'no posterior/polar posterior/not polar'.

- I don't agree with the term 'posterior-like axis' (L46 and L80) since an axis by definition includes polarisation along the whole structure. It's either an 'anterioposterior axis' or a posterior-like pole.

- Likewise, a 'yolk sac' is a morphological structure (a sac) which is not present here. The authors should be careful with their terminology, and whether they mean a particular cell type (by gene expression profile) or a morphological feature.

- My biggest bugbear with this manuscript is, as before, the term 'iDiscoid'. I find this an unhelpful term (the structures are not confined on discs nor are they even particularly disc-shaped) and it adds further field-specific jargon that is difficult for the general reader to interpret, complicates the understanding within the field, and leads to each new paper coming up with a new name every time. I don't think that each new protocol needs its own unique name, but if the authors really want this, then I suggest they use one that sits within the already established terminology of the field - but I leave this decision up to the Editor.

Referee #5 (Remarks to the Author):

In the manuscript entitled "Modelling Human Post-Implantation Development via Extra-Embryonic Niche Engineering", Hislop and colleagues described a cell engineering approach to model early post-implantation phases of human embryonic development. This approach, called iDiscoid, is based on the use of pluripotent stem cells and recapitulate the complex interaction between embryonic and extra-embryonic lineages.

I was specifically asked to comment about the hematopoietic section of this study.

While there is no doubt that evidence of active hematopoiesis is observed with the iDiscoids, the characterization provided does not allow to define it as yolk sac (YS)-like hematopoiesis.

To support their claims, the authors leverage on their comparison with the dataset from the manuscript of Tyser et al, but that comparison will only show that hematopoietic cells are similar to each other, compared to the rest of the embryo. Thus, this comparison is not sufficient to define the ontogeny of the hematopoietic process observed in this setting.

In fact, embryonic hematopoiesis occurs in at least 4 successive waves, two of which originate in the YS, the primitive and the so-called EMP one. As such, when describing YS-like hematopoiesis in vitro, it is quintessential to define which specific YS program is observed. This can be characterized easily as there are program-specific features as well as program-specific lineages that are derived in the YS and several reports have described protocol and markers for the analysis of the hematopoietic output.

Therefore, to determine which hematopoietic program(s) is(are) recapitulated here, the defining characteristics of YS hematopoiesis should be analysed, such as:

- analysis of HOXA cluster gene expression in the endothelial and hematopoietic progenitors. YS structures do not express HOXA genes (shown by many, recently by Calvanese et al Nature 2022) therefore the pattern of expression of HOXA genes, if any, would allow to allocate the hematopoiesis observed in the iDiscoids as YS-like. This reviewer believes that this is the case, but this has to be formally proven, something fairly easy to do since the authors already possess scRNAseq analysis.

- quantitative analysis of the globin expression pattern of the erythroid progeny of CD43. Authors have already performed scRNAseq analysis of day 21 cultures as well, so they should focus their analysis specifically to define which isoforms of hemoglobin are expressed in erythroid cells and the relative contribution in each cell. From the partial data shown in Ext Data Fig 14F it looks some cells express exclusively HBE while others express also HBG1. This hints that probably they are recapitulating the earliest YS program, i.e., primitive. But since the EMP generate erythroid progeny expressing both HBE and HBG1/2, other proofs would be needed, in particular analysis of the T-lymphoid (as described again by many, recently by Atkins et al, JEM 2021) and neutrophil (see for example Bredemeyer et al, Development 2022) potential of the emerging CD43. As primitive hematopoietic progenitors are devoid of T-lymphoid and neutrophil potential, failure to observe these lineages in vitro along with proper positive control, would allow to define the hematopoietic program observed.

Other comments:

First of all, for FACS plots, all negative controls such unstained and FMO should be provided, otherwise the plots are uninterpretable. In addition, there is a quite profound discrepancy between the immunofluorescent analysis in Ext Data Fig 14A and the FACS plot in Fig 14G, with the latter showing little if any CD235+ cells, which is puzzling as the cells were pre-gated based on CD43 expression. How do the authors explain this discrepancy? Does this reflect experimental variability? Same goes for Ext Data Fig 14B and C, where the hematopoietic cells displayed look unilineage. Do the authors observe unilineage hematopoiesis in each "hematopoietic area" or do they observe multilineage hematopoiesis as well in some area of the iDiscoid? Does this change across

experiments? I believe it would be beneficial for the overall interpretation of the data if a more quantitative analysis of the reproducibility across experiments of their hematopoietic output would be provided. If hematopoietic foci observed are characterized by unilineage output, that should be stressed in the text and in the discussion as it would hint at the existence of specific niches directing the differentiation of hematopoietic progenitors towards specific lineages. While this would represent an interesting observation that was never reported before, it is also true the process of emergence of primitive hematopoietic lineages, to the best of my knowledge, has not been studied in this fine detail in primates, and almost completely unexplored in the human setting.

Fig 4F and Ext Data Fig 15. The hemoglobin staining looks quite odd. I understand that images with different magnification are shown, but it is difficult to interpret if it the Hb signal is real staining. Can the author provide an image at higher magnification of the Hb staining, similarly to what provided for CD43 expression?

Ext Data Fig 13F. The results of the clonogenic assay performed with CD34+ cells are totally unconvincing. Pictures and quantification of the resulting CFC are needed. In particular, YS-hematopoiesis is biased towards the erythroid lineage and hence one would expect a robust presence of erythroid progeny, yet there is very little, if any, expression of CD235 (the lack of proper negative controls again hampers a proper interpretation of the data). YS-derived erythropoiesis has been extensively studied, in particular by the Elefanty, the Keller and the Slukvin groups, as such proper protocols and markers to read out erythroid output can be found in their papers. If the CFU assay with CD34+ cells does not yield many hematopoietic colonies, I suggest testing the CFC potential of the derivative CD43+ cells.

Blood islands are a characteristic proper only of primitive hematopoiesis and their structure, emergence has been described in anatomical detail. In particular, the lumen of blood islands is formed via the disappearance of mesodermal cells that are located centrally to the cells beginning to express vascular markers (reviewed in Julien et al, FEBS letters 2016).

In the absence of visual confirmation that this process occurs in the iDiscoids as well as of a characterization of the hematopoietic output, as proposed in my comments above, I would refrain to call the areas of active hematopoiesis as blood islands, and refer them as hematopoietic foci, and only discuss how they are reminiscent of blood islands.

In addition, whether blood islands in the human embryo contain bona fide hemogenic endothelial cells (HECs) is still a matter of debate and even if they do, HECs would represent a transient intermediate, rapidly differentiating from the hemangioblast (HB). HECs are instead the immediate blood precursor in the EMP program.

Of note, HB and HEC share many markers, but only HB are able to form BI-CFC in methylcellulose. Again, as the nature of hematopoietic activity cannot be determined yet, the nomenclature has to be adapted. I am aware that there is a lot of confusion in the field and that in Tyser et al a HECs signature has been described, but those cells were never functionally tested for their HB vs HEC activity. In this manuscript, in the absence of functional tests or other means to distinguish HBs from HECs, I would refrain from defining them and describe them simply as "Runx1+ cells" or cells displaying endothelial and hematopoietic markers, etc.

Other semantic issues:

TAL1 is not a HEC marker only, as it is expressed in hematopoietic committed mesoderm as well as endothelial and HE precursors. It becomes an HEC marker only in combination with Runx1.

Ext Data Fig 14, semantic is not correct. The markers listed under “hemogenic endothelium” are not HEC markers, but just standard genes express by endothelial cells, including by HECs.

GATA1 is not a megakaryocyte marker as it is also expressed, even at higher levels, in the erythroid lineage. And CD45 is not only a macrophage marker as it is pan-hematopoietic.

Author Rebuttals to First Revision:

**Nature Manuscript # 2022-01-20991C-Z
Response to Referees:**

Referee #R1

In their revision, the authors have gone to considerable lengths to address a number of critical issues raised by all referees in the first round of review. The authors provide additional data and analyses that make the characterization of this embryo model and its cellular components far more convincing than that provided in the original description. The new findings do much to strengthen the case that i-Discoids capture many important cell types and morphogenetic events in post-implantation human development in a manner that shows strong fidelity to available benchmark datasets. The study in its present form is a significant contribution to the literature in this fast-moving field and it is timely. It will be of great interest to compare these findings to similar studies likely to emerge in the near future.

Response:

We very much appreciate the reviewer's support of the current study.

Referee #R4 (Remarks to the Author):

We thank this reviewer for the critical analysis of our manuscript and helpful comments. Addressing the comments has helped us to significantly improve this work. We hope R4 will find this revision compelling and satisfactory.

Summary of Edits Made to Address Comments

- Adjustment of title and abstract to clarify absence of trophectoderm and to reflect derivation of the model from hypoblast and epiblast.
- Adjustment of text to replace references to a "synthetic gene circuit" with the term "transgene".
- Addition of a further image showing the mixing of the populations in 3D.
- Additional analysis to confirm the similarities of the endoderm in the system are stronger to extraembryonic rather than definitive endoderm.
- Clarification about the occurrence of the ISL1 population in lumen-containing iDiscoids.
- Confirmation that the day 2 anterior-like population has similar positioning as the day 4 anterior-like population, and clarification of differences suggested by the UMAP positioning.
- Adjustment and clarification of CER1-related conclusions.
- Addition of Hoechst inset images for confocal images.
- Corrections to address minor comments.

- The authors should be explicit in their abstract, and also in their title, that this model does not include trophectoderm lineages, and so is not a complete human embryo model.

Response:

We thank the reviewer for this point and fully agree that clarification of this point is much required. We have updated the abstract and the title to reflect the lack of trophectoderm in our model

system. We have also added this sentence to our main text in the first paragraph of page 5: “We did not observe a similarity to the trophoblast lineages”

- I agree with Reviewer 1 that the claim (in the abstract and elsewhere) that an iGata6 system is a ‘synthetic gene circuit’ is overstated. The authors should remove this statement.

Response:

We have adjusted the text accordingly, removing the reference to the synthetic gene circuit within the abstract and in the text. Instead, we use the term “transgene” to explain the construct. Please see page 2, lines 72 and 86 of the text.

- The authors should show their data from 3D mixing of the populations, and not just a schematic (EF1) so that readers can evaluate for themselves the 3D condition.

Response:

We appreciate the reviewer’s comment in this regard. The images of the results of 3D culture were added in **Extended Data Fig 1D** and we apologize if it was not clear. In this round, we have added an additional image covering a larger area of 3D culture (**Extended Data Fig 1E**).

- The scRNA-seq data seems to match reasonably highly to DE as well as PrE. While this is not necessarily surprising given their very highly similar transcriptomic states, are the authors really sure that their iGata6 population is definitely hypoblast and not a variant of DE? This seems to me to be a critical point (on which the whole argument rests) so needs more attention in the text.

Response:

We thank the reviewer for this point. It is true that these fates are similar and have many overlapping marker genes as well as a large subset of functions. In order to be fully sure that the iGATA6 population represents yolk sac endoderm (YSE), we have performed additional specific characterization of the populations corresponding to YSE in relation to DE. We first computed the differentially upregulated genes only between YSE and DE(P) from the Tyser et al. dataset. From this comparison, we were able to see the differential association of known extraembryonic endoderm genes with the YSE identity, including apolipoprotein genes, *PDGFRA*, *CUBN*, and *TTR*. We then compared the DEGs for each of the Day 4 clusters to the differentially expressed marker genes between YSE and DE(P). In order to determine whether a cluster was significantly similar to one identity versus the other, we computed the Jaccard similarity coefficient for each identity per-cluster.

Extended Data Figure 3C-D: TOP: Volcano plots showing the differentially expressed genes between the YS Endoderm (right) and DE(P) identities (left) from Tyser et al. Markers highlighted in red are genes expressed in the respective day 4 iDisco endoderm cluster. (D) Significance of the empirical distribution of the Jaccard similarity to YSE versus DE(P), reflecting high similarity to YSE in the GATA6-low D4_G1 and D4_G2 populations. Higher similarity to DE is observed in the GATA-medium anterior-like populations, aligning with the anterior signature’s greater similarity to DE in human (see reference 43).

In order to determine whether a cluster was significantly similar to one identity versus the other, we computed the Jaccard similarity coefficient for each identity per-cluster.

We measured the similarity to YSE versus the similarity to DE based on Jaccard similarity, as explained in materials and methods. We demonstrate that amongst the endoderm populations, the D4_G1 and D4_G2 GATA6-low populations show significantly greater similarity to YSE than to DE. In addition, we have shown the specific markers upregulated in these populations are important genes for YSE development, including the PDGFRA, TTR, and apolipoprotein genes mentioned above. We have included plots reflecting the distribution of significant genes in **Extended Data Fig. 3C**.

D4_G3 and D4_G4 show higher similarity to DE than to YSE, which, in combination with the anterior-associated marker genes shown in e.g., **Fig. 3A** or **Extended Data Fig. 3B**, reinforces the more anterior-like identity of these populations in accordance with the human anterior signature discussed by Zhu et al. (reference 43)

This section can be found in paragraphs 3-4 of page 4 of the manuscript and in **Extended Data Fig. 3C-D**.

- The quantifications shown now are much improved from before. But I still don't have a quantitative sense of how common it is that individual lumens have an amnion populations.

Response:

iDiscoids show robust development of amnion layer. We have found that 98.8% of WT clusters express the amnion marker ISL1 around day 4 in 100% of the lumen-containing WT structures at day 3, we observe the expression of ISL1, consistent across 3 independent experiments. We have added a graph reflecting this in **Fig 2F**.

On page 6, lines 244-246 we have mentioned: We found that in iDiscoids containing cavitated structures, all (100%, n = 350) expressed ISL1 at a level high above background (Fig B). We show that all (n = 87) but one of the observed WT clusters (n = 88) have ISL1 expression above the expression levels in the noggin treatment group.

- I don't understand why the 'anterior-like' hypoblast populations form 2 distinct clusters on the UMAP at different days (ED7B and Fig 3A) – doesn't this imply that the population changes dramatically over this time period (if it is even the same population at all)? I would be more convinced if you could find out what genes are differentially expressed at these 2 times/populations, and show spatially that they do both correspond to (the same population) of anterior hypoblast.

Response:

We analyzed the differentially expressed genes between the day 2 cluster and the day 4 cluster expressing anterior hypoblast genes. We performed gene set enrichment analysis on the genes that were upregulated greater than 2-fold in day 2 as compared to day 4 (86 genes). We found that the top 10 pathways corresponding to these genes all correspond to proliferation and the cell cycle. Additionally, among the top ten most upregulated genes, we identified genes such as *UBE2C* and centromere proteins (*CENPU*, *CEMPW*, *CENPE*) that are highly associated with

Figure 2F: Violin plot showing the expression of ISL1 in day 4 iDiscoids containing lumens (n = 350). Dotted line indicates negative threshold for ISL1. Dotted line indicates negative threshold for ISL1.

proliferating cells, implying that the major differences between these two populations comes down to a more highly proliferative state within the day 2 anterior.

As a follow up, we have verified that the anterior-associated proteins LHX1 and HHEX are expressed at day 2 as predicted by the scRNA-seq (scRNA-seq in Extended Data Fig. 3B). We have confirmed that the cells expressing these genes specify at a similar location on day 2 as they do on day 4 (please see **Extended Data Fig. 7A**).

- I think it's too much of a stretch to say that the marmoset data (ED11A) is 'subtly anti-polar-like'. Sure, the data don't support a distinct polar bias, but this could be due to sampling limitations rather than a completely different biological mechanism. The same is true for the Cynomolgus monkey data too (ED11C) where there isn't even an apparent posterior bias for TBXT, which we know is definitely posterior-biased in vivo.

Extended Data Figure 7A: Z-slice of a covered day 2 iDiscoïd structures showing a pole of cells expressing anterior markers, matching the polarization of those markers shown in day 4 (Fig. 3C).

Response:

We appreciate the reviewer's point regarding the potential sampling limitations within the external datasets shown. Because of this, and at the suggestion of the editor, we have withdrawn this figure and have modified the discussion relating to the polarization regimes in the iDiscoïd system to clarify that they can be also a potential experimental artifact. Please find the re-written paragraphs for this section on **page 7**.

- How does the observation that some cells co-express both CER1 and TBXT (ED10) affect the quantification of polarity? Ie could some of these double-positive cells be skewing the data towards 'syn-polar'?

Response:

Thanks to the reviewer for this comment. We agree that further quantitative analysis in a sufficient number of 3D images can shed more light into this matter. We would like to highlight **Extended Data Fig. 7D-F**, showing that these double-positive cells are present in iDiscoïds with both syn-polar and anti-polar polarization types. Additionally, in our assessment, we consistently observed that the presence of double positive cells within the TBXT-expressing pole was not strong enough or dense enough to generate a CER1 pole that would affect our radial quantification – rather, they were seen in very small groups of cells that are frequently outnumbered by the extraembryonic CER1 pole.

- Noggin treatment (ED9) seems to be having an effect on the polarisation of both TBXT and CER1, while the text states that only TBXT is affected. The authors should either show quantification as proof of this, or amend their statement.

Response:

We appreciate the reviewer for this comment. Because we more extensively discuss the polarization in later paragraphs, we have changed the message that we are presenting related to this data; instead of discussing polarization, we merely discuss the specification of these two cell

types. Accordingly, we have amended this statement to the following: “The emergence of posterior domains showed dependence to local BMP4 signaling, as a small molecule, Noggin, could eliminate TBXT⁺ poles, while we still observed the specification of CER1-expressing cells.”. Please find this change in lines 292-295.

Minor points:

- I still think the authors need to show DAPI/Hoescht only channels for all of their confocal images. Otherwise, it's very hard to tell where negative cells are. Also, they need to ensure that their arrowheads are completely consistent across panels (eg Fig3C I'm fairly sure the arrow is higher in the LHX1 panel than in the CER1 panel).

Response:

We thank the reviewer for this comment. In this version, we have gone through and added Hoechst images for every set of images for which it was possible, either as an inset to 2D max projection/widefield images or alongside Z-slices to show the areas containing cells (please see Fig. 2C as an example).

We were unable to include Hoechst for the small following sets of images. However, we believe in these limited cases the intended message is preserved and communicated properly:

- F-actin has been added in **Figure 2G-I** in order to show the presence of a lumen within the iDiscoids shown and can clearly show cell borders outside of these lumens as fiduciary markers.
- In **Figure 3C**, we have stained Phalloidin in the same channel as EGFP in order to be able to stain three markers simultaneously for a more complete assessment of the anterior signature. This combined staining was still able to show the presence of a lumen and cells within the covered iDiscoid shown.
- F-actin has been shown alongside other markers in **Figure 4H** and **Figure 5A-C** to highlight cellular morphology and can still clearly show areas lacking cells.
- In **Extended Data Figure 7B-C**, we are unable to show a Hoechst or F-actin image, as we have stained three markers to show anterior and posterior cell specification together. This being a widefield image, phalloidin was not effective alongside EGFP to show cell morphology. We feel that in this case, the EGFP is sufficient to highlight the borders of the WT areas, and will be present in all other cells outside of the WT areas.
- In **Extended Data Figure 14B**, we are unable to show a Hoechst or F-actin image, as the third channel was stained for another marker that was not shown. We feel that EGFP can highlight the tissue's morphology, and the PAX6 and NCAM staining are sufficient to show the morphology of the structures of interest that illustrate the point made in the text.

While we believe we have sufficiently addressed this comment to the level possible, we still will remove and replace these specific images at the discretion of our editor or reviewer. We have also adjusted the arrowheads to ensure consistency where they appear.

- The categories in Fig3E should be renamed according to their example images: 'negative, polar, non-polar' makes much more sense than 'no posterior/polar posterior/not polar'.

Response:

We have adjusted the figure to correct this discrepancy in terminology. Please see **updated Fig 3E**.

- I don't agree with the term 'posterior-like axis' (L46 and L80) since an axis by definition includes polarisation along the whole structure. It's either an 'anterioposterior axis' or a posterior-like pole.

Response:

We have adjusted the text at these lines and elsewhere in the document to use the term "posterior pole" when we are referring to the posterior in isolation. Please see page 6, line 279, and page 7, line 316.

- Likewise, a 'yolk sac' is a morphological structure (a sac) which is not present here. The authors should be careful with their terminology, and whether they mean a particular cell type (by gene expression profile) or a morphological feature.

Response:

We appreciate that though the iDiscoid model produces cells and morphological features that are reflective of the yolk sac, those features are not produced in a closed sac structure. We have adjusted the text accordingly (i.e., used "yolk sac tissue").

- My biggest bugbear with this manuscript is, as before, the term 'iDiscoid'. I find this an unhelpful term (the structures are not confined on discs nor are they even particularly disc-shaped) and it adds further field-specific jargon that is difficult for the general reader to interpret, complicates the understanding within the field, and leads to each new paper coming up with a new name every time. I don't think that each new protocol needs its own unique name, but if the authors really want this, then I suggest they use one that sits within the already established terminology of the field - but I leave this decision up to the Editor.

We respectfully believe that "iDiscoid" terminology is not unhelpful for readers of this manuscript: 1) the structure is inducible, hence the abbreviation "i" was included. 2) During the formation of the structures we show disc-"like" morphology in the wild type islands (see **Extended Data Figs. 1F-H** for more information). The wild-type structures are initially in the form of a flat disc of cells. Subsequently, they produce cell mass while they are fully confined by the iGATA6-engineered population. This confinement forms circular structures over the initial organization steps (please see **Supplementary Video 1**). This configuration triggers 2D-to-3D self-organization and eventual upward crawling of the iGATA6⁺ cells on the top of the WT islands and formation of the amniotic cavity. At this time, the morphology mimics a hypoblast-epiblast interface with two open spaces on each side exhibiting the bilaminar disc morphology.

If possible, we would prefer to keep this terminology, which has already been acquired and cited by others who reviewed our Biorxiv manuscript (see Moris & Sturmey. *Development*, 2023).

Extended Data Figure 1H: Disc-shaped structures after 3 days of iDiscoid development

Figure 2D: Bilaminar disc formed from WT and iGATA6 in a circular disc shape, as seen in the amnion at the base of the structure.

Reviewer # R5

We thank our reviewer for the constructive feedback related to the aspects of developmental hematopoiesis of iDiscooids. The comments helped us to examine the system and strengthen the message within this work. We hope R5 will find this revision compelling and satisfactory.

Summary Edits Made to Address Comments

- Investigation of hemoglobin ϵ and γ expression dynamics in iDiscooid using immunofluorescence staining and quantitative RT-PCR
- Investigation for hemogenic endothelium and EHT in culture using immunofluorescence staining.
- Analysis of single cell RNA-seq data for hemogenic endothelium
- Analysis of single cell RNA-seq data for identification of markers associated with yolk sac hematopoiesis
- Analysis of single cell RNA-seq data for the identification of lymphoid potential and neutrophil potential
- Flow cytometry analysis of the iDiscooid at multiple timepoints to investigate the erythroid, myeloid, megakaryocyte, lymphoid and neutrophil-like lineages in iDiscooid
- Addition of negative controls flow cytometry plots (FMO) to Supplementary Information
- Addition of quantification of colony types and images to the methylcellulose assay data

In the manuscript entitled “Modelling Human Post-Implantation Development via Extra-Embryonic Niche Engineering”, Hislop and colleagues described a cell engineering approach to model early post-implantation phases of human embryonic development. This approach, called iDiscooid, is based on the use of pluripotent stem cells and recapitulate the complex interaction between embryonic and extra-embryonic lineages.

I was specifically asked to comment about the hematopoietic section of this study. While there is no doubt that evidence of active hematopoiesis is observed with the iDiscooids, the characterization provided does not allow to define it as yolk sac (YS)-like hematopoiesis. To support their claims, the authors leverage on their comparison with the dataset from the manuscript of Tyser et al, but that comparison will only show that hematopoietic cells are similar to each other, compared to the rest of the embryo. Thus, this comparison is not sufficient to define the ontogeny of the hematopoietic process observed in this setting.

In fact, embryonic hematopoiesis occurs in at least 4 successive waves, two of which originate in the YS, the primitive and the so-called EMP one. As such, when describing YS-like hematopoiesis in vitro, it is quintessential to define which specific YS program is observed. This can be characterized easily as there are program-specific features as well as program-specific lineages that are derived in the YS and several reports have described protocol and markers for the analysis of the hematopoietic output. Therefore, to determine which hematopoietic program(s) is(are) recapitulated here, the defining characteristics of YS hematopoiesis should be analysed, such as:

- analysis of HOXA cluster gene expression in the endothelial and hematopoietic progenitors. YS structures do not express HOXA genes (shown by many, recently by Calvanese et al Nature 2022) therefore the pattern of expression of HOXA genes, if any, would allow to allocate the

hematopoiesis observed in the iDiscoids as YS-like. This reviewer believes that this is the case, but this has to be formally proven, something fairly easy to do since the authors already possess scRNAseq analysis.

Response:

We appreciate the reviewer's insight and have taken the steps outlined in their suggestions to show that the hematopoiesis observed in iDiscoid is recapitulating yolk sac hematopoiesis. We have analyzed our scRNA-seq data from the D21 time point and have confirmed the broad absence of HOXA gene expression within our hematopoietic and endothelial clusters, which we have shown in **Extended Data Fig 12B**. Our hematopoietic and endothelial clusters also show expression of embryonic genes *LIN28A*, *GAD1*, and *FGF23* that are shown to be correlated with early waves of hematopoiesis and pre-HSC hematoendothelial programs (CS10-11) (see reference 63). We have included data showing this in **Extended Data Fig 12. B-C**. Additionally, the expression of *HBE1* and *HBZ* in our erythroid cells in day 21 single cell RNA-seq data demonstrates an expression pattern unique to yolk sac erythropoiesis (see reference 61). We have added this analysis to **Extended Data Fig 12H**. Please see page 9, third paragraph for this information.

- quantitative analysis of the globin expression pattern of the erythroid progeny of CD43. Authors have already performed scRNAseq analysis of day 21 cultures as well, so they should focus their analysis specifically to define which isoforms of hemoglobin are expressed in erythroid cells and the relative contribution in each cell. From the partial data shown in Ext Data Fig 14F it looks some cells express exclusively HBE while others express also HBG1. This hints that probably they are recapitulating the earliest YS program, i.e., primitive.

Response:

At the reviewer's recommendation, we explored the expression pattern of *HBE1* and *HBG1* in our D21 scRNA-seq data. We observed the specification of cells expressing different levels of these two markers, with a slight bias towards *HBE1*-high cells within the erythroid cluster, suggesting that both primitive-like and EMP-lineage-like erythroid cells were present (**Reviewer Fig. A**).

We further explored the specification of these cells over time within iDiscoids via qPCR for the globin isoforms between day 12 and day 21. We observed a transition from a high ratio (1.5:1) of *HBE* to *HGB*

Figure 5E: qPCR of iDiscoid whole tissue on different days of culture showing a reduction in the ratio of *HBE* to *HGB* expression.

at day 12, followed by a ratio closer to 0.5:1 between the two isoforms at day 21 (**Fig. 5E**). Intriguingly, this pattern mimics the pattern of switching between these globin types between yolk sac primitive erythropoiesis and yolk sac EMP-based erythropoiesis described previously (e.g., reference 61). We

confirmed this change in RNA expression ratio was reflected in protein expression via staining for ϵ and γ hemoglobin proteins. We observed the high expression of hemoglobin ϵ and low expression of hemoglobin γ within clusters of hemoglobin-expressing cells at days 12 and 15. However, by day 21 the erythroid population shows an

Reviewer Figure A: Co-expression analysis of *HBE1* and *HBG1* in day 21 iDiscoid single cell RNA-seq data.

Figure 5K-M: Flow cytometry analysis showing the existence of CD45⁺/CD117⁺/CD7⁺/CD43⁺ lymphoid-like cells, CD31⁺/CD45⁺/CD15⁺ neutrophil-like cells and CD56⁺/VLA-4⁺/CD45⁺ natural killer-like cells.

mimics the specification of these erythropoietic subtypes in vivo. We have included this information in **Fig. D-E** and **Extended Data Fig. 10A-C** and have adjusted the text at page 8, fourth paragraph. We would like to note that there is no discrepancy between the scRNA-seq globin analysis and shown results for qPCR and IF image analysis, as we expect to observe primitive-like erythroid cells at D21 of the culture as the primitive colonies generated earlier would still be present within the organoid.

But since the EMP generate erythroid progeny expressing both HBE and HBG1/2, other proofs would be needed, in particular analysis of the T-lymphoid (as described again by many, recently by Atkins et al, JEM 2021) and neutrophil (see for example Bredemeyer et al, Development 2022) potential of the emerging CD43. As primitive hematopoietic progenitors are devoid of T-lymphoid and neutrophil potential, failure to observe these lineages in vitro along with proper positive control, would allow to define the hematopoietic program observed.

Response:

We appreciate the reviewer's comment motivating us to dig more into the iDiscoid hematopoietic program. Based on the evidence discussed above, indicating that we observed EMP-like hematopoietic activity occurring after day 12, we explored beyond this time point to determine whether we could observe lymphoid and neutrophil specification within the iDiscoid tissue. CD7 and CD117 are two major discriminative surface markers for lymphoid cells in yolk sac, according to a recently published paper yolk sac atlas (see reference 71) as well as in previous *ex vivo* and iPSC-based models (see references 73 and 76). Flow cytometry analysis on iDiscoid tissue as early as day 16 revealed a CD7⁺/CD43⁺/CD117⁺/CD45⁺ population comprising 3.51±1.43% of CD45⁺ cells, a signature suggesting the specification of lymphoid-like progenitor cells in iDiscoid. We were also able to identify a population of CD15⁺/CD31⁺/CD45⁺ neutrophil-like cells on day 21, comprising 2.37±0.98% of the CD45⁺ cells at this time point. We were also able to identify a population of VLA-4⁺/CD45⁺/CD56⁺ natural killer-like cells on day 21 (**Fig 5K-M**). We were additionally able to identify the presence of cells expressing these markers in the D21 scRNA-seq, within the lymphoid/myeloid cluster. Taken together, these observations reveal the presence of definitive phase of hematopoiesis in iDiscoid on day 21. We have added this data to **Fig 5K-M** and **Extended Data Fig 12H**. We have added this information to the main text. Please see the second paragraph of page 10.

Other comments:

increase in hemoglobin γ -high cells, which resulted in a decreased ratio of hemoglobin ϵ to hemoglobin γ . This data supports the timeline of the switch suggested by qPCR (see **Fig. 5D**, **Extended Data Fig. 10A-B**). We also performed image analysis that further supported the occurrence of a switch, from an average ratio of 4.49±0.08 ϵ/γ on day 15 to an average ratio of 1.78±0.02 on day 21 (see **Extended Data Fig. 10C**). We therefore conclude that we have two types of YS-like erythropoiesis occurring in iDiscoids, occurring in a sequence that

First of all, for FACS plots, all negative controls such unstained and FMO should be provided, otherwise the plots are uninterpretable.

Response:

We have updated and replaced figures showing flow cytometry plots to include unstained and FMO controls for the markers shown. We have also included FMO plots for the new flow cytometry plots. Please see **Supplementary Information 1** for FMO plots and back-gatings.

In addition, there is a quite profound discrepancy between the immunofluorescent analysis in Ext Data Fig 14A and the FACS plot in Fig 14G, with the latter showing little if any CD235+ cells, which is puzzling as the cells were pre-gated based on CD43 expression. How do the authors explain this discrepancy? Does this reflect experimental variability?

Response:

We appreciate the reviewer's comment, as we feel this has helped us strengthen our data substantially. We have repeated flow cytometry for this time point and have included the appropriate controls for the flow cytometry data. We have observed a larger population of CD43⁺/CD235ab⁺ cells, aligned with our expectations based on the IF staining. To assess the experimental variability, we have characterized the erythroid, myeloid and megakaryocyte lineages of Day 21 culture in 8 biological replicates in three independent experiments with flow cytometry (**Fig 5J**). Our flow cytometry analysis confirmed minimal variability between independent biological replicates in myeloid and erythroid lineage markers. We also performed additional follow-up IF for these markers, and assessed the experimental variability, via analysis of the types and composition of hematopoietic foci that are formed over multiple experiments (see **Extended Data Fig. 11**) which we have discussed in the response to the next comment. Please find the updated flow cytometry plots in **Fig 5J**.

Same goes for Ext Data Fig 14B and C, where the hematopoietic cells displayed look unilineage. Do the authors observe unilineage hematopoiesis in each "hematopoietic area" or do they observe multilineage hematopoiesis as well in some area of the iDiscoid? Does this change across experiments? I believe it would be beneficial for the overall interpretation of the data if a more quantitative analysis of the reproducibility across experiments of their hematopoietic output would be provided.

If hematopoietic foci observed are characterized by unilineage output, that should be stressed in the text and in the discussion as it would hint at the existence of specific niches directing the differentiation of hematopoietic progenitors towards specific lineages. While this would represent an interesting observation that was never reported before, it is also true the process of emergence of primitive hematopoietic lineages, to the best of my knowledge, has not been studied in this fine detail in primates, and almost completely unexplored in the human setting.

Response:

Thank you for the recommendation to explore these hematopoietic foci more deeply. Using CD42b, CD33, CD235ab, and CD43 as markers for megakaryocytes, myeloid cells, erythroid cells, and pan-hematopoietic lineages, we have analyzed the hematopoietic areas in which these cell types specify within iDiscoid cultures. We have found that while the majority of foci that exist are multilineage, with two or more of these cell types specifying side-by-side, there appear to be

a limited number of unilineage hematopoietic foci that specify within the iDiscoid tissue. Assessment based on staining for CD43, CD33, and CD42b found foci expressing CD33 or CD42b in all cells in an area in $8.8 \pm 2.1\%$ of total foci (avg 42 total); assessment based on staining for CD43, CD235ab, and CD42b of found foci expressing CD235ab or CD42b in all cells in $5.0 \pm 4.6\%$ of total foci (avg 29 total). The highest number of unilineage foci that we were able to identify were CD42b⁺, suggesting that there could be megakaryocyte-favored niches within these areas of the tissue. However, we believe deeper characterization of these niches is beyond the scope of this paper. We have shown the data reflecting the quantification of the hematopoietic niches, as well as examples of different hematopoietic foci, in **Fig 5F-G** and **Extended Data Fig 11**. We have added this information to the main text. Please see the last paragraph of page 8 and first paragraph of page 9.

Fig 4F and Ext Data Fig 15. The hemoglobin staining looks quite odd. I understand that images with different magnification are shown, but it is difficult to interpret if the Hb signal is real staining. Can the author provide an image at higher magnification of the Hb staining, similarly to what provided for CD43 expression?

Response:

We have provided additional images showing the Hb staining in **Fig 5D** and **Extended Data Fig 10A-B**. While there are small artifacts that can be seen in some places within the images, we feel confident that the Hb signal is real staining, especially in light of follow-up stains for specific hemoglobin types discussed above.

Ext Data Fig 13F. The results of the clonogenic assay performed with CD34⁺ cells are totally unconvincing. Pictures and quantification of the resulting CFC are needed. In particular, YS-hematopoiesis is biased towards the erythroid lineage and hence one would expect a robust presence of erythroid progeny, yet there is very little, if any, expression of CD235 (the lack of proper negative controls again hampers a proper interpretation of the data). YS-derived erythropoiesis has been extensively studied, in particular by the Elefanti, the Keller and the Slukvin groups, as such proper protocols and markers to read out erythroid output can be found in their papers.

If the CFU assay with CD34⁺ cells does not yield many hematopoietic colonies, I suggest testing the CFC potential of the derivative CD43⁺ cells.

Response:

We have updated the results of the clonogenic assay to show examples of the colonies that form, and have added quantification of the CFC colonies observed based on assays seeded from day 21 iDiscoid. These are included in **Fig 5H**.

Blood islands are a characteristic proper only of primitive hematopoiesis and their structure, emergence has been described in anatomical detail. In particular, the lumen of blood islands is formed via the disappearance of mesodermal cells that are located centrally to the cells beginning to express vascular markers (reviewed in Julien et al, FEBS letters 2016).

In the absence of visual confirmation that this process occurs in the iDiscoids as well as of a characterization of the hematopoietic output, as proposed in my comments above, I would refrain to call the areas of active hematopoiesis as blood islands, and refer them as hematopoietic foci, and only discuss how they are reminiscent of blood islands.

In addition, whether blood islands in the human embryo contain bona fide hemogenic endothelial cells (HECs) is still a matter of debate and even if they do, HECs would represent a transient intermediate, rapidly differentiating from the hemangioblast (HB). HECs are instead the immediate blood precursor in the EMP program.

Of note, HB and HEC share many markers, but only HB are able to form BI-CFC in methylcellulose. Again, as the nature of hematopoietic activity cannot be determined yet, the nomenclature has to be adapted. I am aware that there is a lot of confusion in the field and that in Tyser et al a HECs signature has been described, but those cells were never functionally tested for their HB vs HEC activity. In this manuscript, in the absence of functional tests or other means to distinguish HBs from HECs, I would refrain from defining them and describe them simply as “Runx1+ cells” or cells displaying endothelial and hematopoietic markers, etc.

Response:

We appreciate the insight the reviewer has provided. We have adjusted the terminology within the paper and discussed that similarity to Tyser’s hemogenic endothelium cluster does not mean that the high-GATA6 cells are HECs without functional evaluation. The adjusted text is on page 7, lines 330-331.

Other semantic issues:

TAL1 is not a HEC marker only, as it is expressed in hematopoietic committed mesoderm as well as endothelial and HE precursors. It becomes an HEC marker only in combination with Runx1.

Response:

We appreciate the reviewer pointing out this issue. We have removed the respective graph and have added a panel on hemogenic endothelium cells in **Extended Data Fig.12D-G**.

Ext Data Fig 14, semantic is not correct. The markers listed under “hemogenic endothelium” are not HEC markers, but just standard genes express by endothelial cells, including by HECs.

Response:

We appreciate this note. We have changed the way that we have shown the endothelial cluster identity in scRNA-seq, and have included an **Extended Data Fig.12D-G**. We have added a section on hemogenic endothelium in the main text. Please see page 10, lines 437-454.

GATA1 is not a megakaryocyte marker as it is also expressed, even at higher levels, in the erythroid lineage. And CD45 is not only a macrophage marker as it is pan-hematopoietic.

Response:

We thank the reviewer for the detailed analysis of our manuscript. We have updated the markers shown to reflect more specific markers for these populations. Please see **Extended Data Fig 12H**.

Referee #R2

1. Add medium optimisation experiments to the supplementary information

We have added the images showing the results of other media regimes to **Supplementary Information 1**. We have also added the following adjustment to the text at page 3: “This behavior was observed when the system was maintained on mTeSR (Supplementary Information 1).”

2. Include some of the discussion about the limitations of the model in relation to the blood differentiation, raised in response to the Reviewer’s final point, into the discussion itself

We appreciate the recommendation of the reviewer. We have added the following point to the discussion: “Bona fide blood islands are formed via the emergence of a lumen resulting from the disappearance of mesodermal cells that are within areas bordered by early endothelial cells. While we have not been able to confirm the occurrence of this process in iDiscooids, the yolk sac tissue within iDiscooids develops morphologically relevant hematopoietic foci that show in vivo like organization of extraembryonic endoderm and mesoderm layers.”

3. Optimisation of the seeding ratio of their iDiscooids was performed between the first (4:1 ratio) and second (81:5) version of the manuscript. This is needs to be explicit in the methods section, and which experiments came from which experimental setup needs to be clear. Many of the figure panels were the same between the two versions (experimental design not consistent for all their data).

We have added an explicit clarification in the methods about what figure panels shown were seeded under the 4:1 ratio versus the 81:5 ratio, and have added a point that both ratios were used. The following text has been added to page 17:

“The GATA6-engineered hiPSCs were seeded at either a ratio of 81:5 or 4:1 with rtTA expressing PGP1 hiPSCs either containing or lacking an mKate reporter gene at a total density of 54,000 cells per cm² in mTeSR-1 supplemented with 10 μM Y-27632. [...]

The following experiments were seeded at a 4:1 ratio, prior to optimization: Fig. 1B-C, Fig. 2C-D, Fig. 3D-E, Extended Data Figs. 1F, 1H, 2A-B, 2D, 2F-H, 6A, 7D-E, 8D-E and 15. The remaining experiments shown were seeded at 81:5.”

Reviewer Reports on the Second Revision:

Referees' comments:

Referee #4 (Remarks to the Author):

For the most part, the authors have addressed my concerns and I only have a few, relatively minor, suggestions remaining.

Most importantly, I still think the authors need to be explicit that their model does not contain trophoblast-like cells in the abstract. Readers shouldn't have to mine the whole introduction to find out this critical piece of information.

Although I appreciate that the authors have added pan-nuclei or pan-cellular stains in several of their figures, I would still want to see a pan-nuclei stain in Fig 1c and Fig 2g, h, l.

A plot legend describing the use of colour in Ext Data 3D is necessary to understand the figure.

Competing interests: the filing of a patent is a competing interest, so the authors shouldn't declare they have none.

Referee #5 (Remarks to the Author):

This revised version of the manuscript by Hislop et al. has been significantly modified. The title of the paper has been changed to better mirror the overall message of the paper. The authors have made significant efforts of clarification, provided control plots for FACS analysis, performed new transcriptomic analysis at single cell level.

There is one standing point that needs to be addressed verbally.

"At the reviewer's recommendation, we explored the expression pattern of HBE1 and HBG1 in our D21 scRNA-seq data. We observed the specification of cells expressing different levels of these two markers, with a slight bias towards HBE1-high cells within the erythroid cluster, suggesting that both primitive-like and EMP-lineage-like erythroid cells were present (Reviewer Fig. A)."

HBG1 (and HBG2) are not markers specific of EMP-derived erythroid lineage, but are also expressed by primitive erythroid cells (REF: PMID: 26095363; 23219550). Of note primitive erythrocytes undergo globin switching in circulation (for obvious reasons thus far shown only in the mouse model PMID: 16263786)

As such the data can be interpreted as either a primitive-to-EMP-like transition or, alternatively, a maturation of primitive erythroid cells.

This reviewer recommends to change the text of page 9 "We conclude iDiscooids show a primitive-to-

EMP-like transition within the erythroid population between two and three weeks of development” to mirror this. Same for the sentence line 405.

The only convincing data that EMP-hematopoiesis is occurring in iDiscoids is the presence of hematopoietic cells expressing NK-markers. This has to be stressed and PMID: 32197069 cited in support of this.

In addition, this reviewer strongly suggests to avoid the use of the ill-defined “pro-definitive” adjective. Better use EMP-like.

Last but not least, there is something wrong with the labeling or the text referring to fig 5K,L; plots and labels look swapped.

Fig 5H. Colonies for plate is not an adequate measure for quantification. Number of cells plated in the plate needs to be used instead for quantify a frequency (# colonies/# cells seeded). That used in fig 5H is not a CFU-E. Add scale bar to 5H pictures

Response to Referees:

We thank our reviewers' and editor's comments for helping us improve the manuscript. In this revised submission, we have addressed all the concerns.

Referee #4

- For the most part, the authors have addressed my concerns and I only have a few, relatively minor, suggestions remaining.

We appreciate the reviewer's suggestions, and we are glad that we could address the concerns.

-Most importantly, I still think the authors need to be explicit that their model does not contain trophoblast-like cells in the abstract. Readers shouldn't have to mine the whole introduction to find out this critical piece of information.

The abstract now is updated regarding the lack of trophoblast: "The extra-embryonic layer in these embryoids lacks trophoblast and exhibits advanced multilineage yolk sac tissue morphogenesis". Please see **line 41**.

-Although I appreciate that the authors have added pan-nuclei or pan-cellular stains in several of their figures, I would still want to see a pan-nuclei stain in Fig 1c and Fig 2g, h, i.

The figures mentioned are now updated, showing the nuclear marker.

Fig1C was acquired using timelapse video imaging using cell lines engineered with a reporter for the cell membrane. We purchased the line from the Allen Institute collection, and the derivation of such lines is not trivial. This timelapse is not trivial, either, as finding an appropriate confocal microscope with a proper focal point for deep imaging in our embryoid setting has been challenging. Also, we could not supplement nuclear dyes for several days in culture as it induces cellular stress due to nuclear dye binding to the DNA. To address the reviewer's comments, we fixed the samples, stained again together with a nuclear dye, and provided images from different stages of lumenogenesis. We have added these new images with the Hoechst staining in the **Extended Data Fig 2A**. We also repeated the inhibitor study, added nuclear stain, and provided the data in Extended **Data Fig 6B, D, E**. Everything was aligned with the past data.

Extended Data Fig 2A

Extended Data Fig 6

-A plot legend describing the use of colour in Ext Data 3D is necessary to understand the figure.

The color in this figure did not carry any additional information. We have changed all colors to grayscale.

-Competing interests: the filing of a patent is a competing interest, so the authors shouldn't declare they have none.

We thank the reviewer for her/his comment. We changed the status of competing interests for the authors contributing to patent applications.

Referee #5

-This revised version of the manuscript by Hislop et al. has been significantly modified. The title of the paper has been changed to better mirror the overall message of the paper. The authors have made significant efforts of clarification, provided control plots for FACS analysis, performed new transcriptomic analysis at single cell level.

We appreciate the reviewer's suggestions, and we are glad that we have addressed this reviewer's comments.

-There is one standing point that needs to be addressed verbally.

-“At the reviewer's recommendation, we explored the expression pattern of HBE1 and HBG1 in our D21 scRNA-seq data. We observed the specification of cells expressing different levels of these two markers, with a slight bias towards HBE1-high cells within the erythroid cluster, suggesting that both primitive-like and EMP-lineage-like erythroid cells were present (Reviewer Fig. A).”

-HBG1 (and HBG2) are not markers specific of EMP-derived erythroid lineage, but are also expressed by primitive erythroid cells (REF: PMID: 26095363; 23219550). Of note primitive erythrocytes undergo globin switching in circulation (for obvious reasons thus far shown only in the mouse model PMID: 16263786)

As such the data can be interpreted as either a primitive-to-EMP-like transition or, alternatively, a maturation of primitive erythroid cells.

-This reviewer recommends to change the text of page 9 “We conclude iDiscooids show a primitive-to-EMP-like transition within the erythroid population between two and three weeks of development” to mirror this. Same for the sentence line 405.

We appreciate the reviewer's comment. We have changed the conclusion and included the possibility of erythrocyte maturation: “The temporal switch in HemogeniX-embryoids' erythroid cells potentially implies a primitive-to-EMP-like transition in erythropoiesis. However, since the maturation of primitive erythrocytes also involves a globin switch, further investigation is necessary to understand the precise mechanisms at play”. Please see the conclusion of the first paragraph of page 10. Since the primitive-to-EMP transition cannot be declared, we removed the statement in “Line 405” of the previous submission.

-The only convincing data that EMP-hematopoiesis is occurring in iDiscooids is the presence of hematopoietic cells expressing NK-markers. This has to be stressed and PMID: 32197069 cited in support of this.

We have added this point and the reference in the discussion section, mentioning: “lineages derived from EMP-hematopoiesis such as NK-like cells”. Please see the second paragraph of the discussion.

-In addition, this reviewer strongly suggests to avoid the use of the ill-defined “pro-definitive” adjective. Better use EMP-like.

We have removed this term and replaced it with EMP-like program and have written: “Hence, yolk sac tissues of HemogeniX-embryoids show EMP- and LMPP-like hematopoietic programs”. Please see the conclusion of the first paragraph of page 11.

-Last but not least, there is something wrong with the labeling or the text referring to fig 5K,L; plots and labels look swapped.

We apologize for this mistake. These graphs labelling have been corrected. Please see **Fig 5K-M**.

-Fig 5H. Colonies for plate is not an adequate measure for quantification. Number of cells plated in the plate needs to be used instead for quantify a frequency (# colonies/# cells seeded). That used in fig 5H is not a CFU-E. Add scale bar to 5H pictures

We have adjusted the quantification. We have also replaced the figure showing a proper CFU-E colony. The scale bars have been added. Please see **Fig 5F**.